**TOOLS**

# Development of ultrafast camera-based single fluorescent-molecule imaging for cell biology

Takahiro K. Fujiwara[1], Shinji Takeuchi[2], Ziya Kalay[1], Yosuke Nagai[2], Taka A. Tsunoyama[3], Thomas Kalkbrenner[4], Kokoro Iwasawa[1], Ken P. Ritchie[5], Kenichi G.N. Suzuki[1,6], and Akihiro Kusumi[1,3]

**The spatial resolution of fluorescence microscopy has recently been greatly enhanced. However, improvements in temporal resolution have been limited, despite their importance for examining living cells. Here, we developed an ultrafast camera system that enables the highest time resolutions in single fluorescent-molecule imaging to date, which were photon-limited by fluorophore photophysics: 33 and 100 µs with single-molecule localization precisions of 34 and 20 nm, respectively, for Cy3, the optimal fluorophore we identified. Using theoretical frameworks developed for the analysis of single-molecule trajectories in the plasma membrane (PM), this camera successfully detected fast hop diffusion of membrane molecules in the PM, previously detectable only in the apical PM using less preferable 40-nm gold probes, thus helping to elucidate the principles governing the PM organization and molecular dynamics. Furthermore, as described in the companion paper, this camera allows simultaneous data acquisitions for PALM/dSTORM at as fast as 1 kHz, with 29/19 nm localization precisions in the 640 × 640 pixel view-field.**

## Introduction

Extensive attention has recently focused on improving the spatial resolution of fluorescence microscopy, leading to the development of various types of super-resolution methods (Shcherbakova et al., 2014; Liu et al., 2015; Nicovich et al., 2017; Sahl et al., 2017; von Diezmann et al., 2017; Baddeley and Bewersdorf, 2018; Sigal et al., 2018; Sezgin et al., 2019; Lelek et al., 2021; Liu et al., 2022). In contrast, there have been limited efforts toward enhancing the temporal resolution of fluorescence microscopy, particularly in single fluorescent-molecule imaging (SFMI) and single-molecule localization microscopy (SMLM; Wieser et al., 2007; Jones et al., 2011; Huang et al., 2013; Hiramoto-Yamaki et al., 2014; Kinoshita et al., 2017). However, improving the temporal resolution is critically important (Balzarotti et al., 2017; Eilers et al., 2018; Koyama-Honda et al., 2020; Schmidt et al., 2021) for observing living cells, and particularly for investigating the dynamic movements and interactions of molecules at the single molecule level (Baboolal et al., 2016). This was demonstrated by our detection of hop diffusion of membrane molecules in the plasma membrane (PM) by enhancing the time resolution of single-particle tracking down to 25 µs (Fujiwara et al., 2002, 2016; Murase et al., 2004). However, this method necessitated the use of large (40-nm diameter) gold particles as probes. Since fluorescent probes are much smaller (<1 nm) and are used far more broadly than gold probes, in the present study, we aimed to improve the temporal resolution of imaging single fluorescent molecules (many molecules at the same time) to levels comparable to those of gold-particle tracking.

Here, we developed an ultrahigh-speed camera system that has enabled the fastest SFMI to date. We achieved a 100-µs resolution (10-kHz frame rate) with a 20-nm localization precision for single Cy3 molecules for a frame size of 14 × 14 µm² (256 × 256 pixels), and a 33-µs resolution (30-kHz frame rate) with a 34-nm localization precision for a frame size of 7.1 × 6.2 µm² (128 × 112 pixels). These frame rates are faster than normal video rate (30 Hz) by factors of 330 and 1,000, respectively (summarized in Table 1).

The ultrafast camera that we have developed simultaneously achieves high frame rates and reasonable photon sensitivity, allowing for a good balance between the temporal resolution and the localization precision. This balance is particularly important for high-speed imaging (for observing molecular dynamics in live cells), where observations have to be made near the photon-limited conditions due to fluorophore photophysics. Furthermore, we optimized the imaging conditions and fluorescent dye selection for high-speed simultaneous tracking of many single molecules for reasonable durations.

For tracking single molecules in living cells, we favor a camera-based method because it enables observations of several

---

[1]Institute for Integrated Cell-Material Sciences (WPI-iCeMS), Kyoto University, Kyoto, Japan; [2]Photron Limited, Tokyo, Japan; [3]Membrane Cooperativity Unit, Okinawa Institute of Science and Technology Graduate University (OIST), Okinawa, Japan; [4]Carl Zeiss Microscopy GmbH, Jena, Germany; [5]Department of Physics and Astronomy, Purdue University, West Lafayette, IN, USA.; [6]Institute for Glyco-core Research (iGCORE), Gifu University, Gifu, Japan.

Correspondence to Akihiro Kusumi: akihiro.kusumi@oist.jp.

Table 1. **Summary of the camera frame rate, maximal data acquisition image size (pixels), position determination precision, and number of observable frames for various fluorescent probes for the developed camera system and other cameras**

| Frame Rate (kHz) | Maximal Data Acquisition Image Size (pixels) | | Probe and Illumination Power Density | Position Determination Precision (nm; mean ± SEM) | | Reference |
|---|---|---|---|---|---|---|
| | 1024PCI-CMOS[a] | SA1-CMOS[a] | | TIR illumination | Oblique illumination | |
| 0.002 | 128 × 128 (EM-CCD) | | Cy3[b] | 3.2 molecules lasting 10–100 frames | ND | Pertsinidis et al. (2010) |
| 0.06 | 1,024 × 1,024 | 1,024 × 1,024 | Cy3[c] 79 µW/µm$^2$ | 2.6 ± 0.099 ($n$ = 50) molecules lasting ≈1 frame | ND | This work |
| 0.065 | 512 × 512 (EM-CCD) | | JF549[d] | 23 4.5% molecules lasting >100 frames | ND | Banaz et al. (2019) |
| 0.065 | 512 × 512 (EM-CCD) | | TMR[d] | 24 6.1% molecules lasting >100 frames | ND | Banaz et al. (2019) |
| 10 | 256 × 256 | 768 × 640 | Cy3[e] 23 µW/µm$^2$ | 27 ± 1.1 ($n$ = 50) | 38 ± 1.7 ($n$ = 50) 14% molecules lasting >100 frames | This work |
| 10 | 256 × 256 | 768 × 640 | Cy3[c] 79 µW/µm$^2$ | 20 ± 0.71 ($n$ = 50) 1.6% molecules lasting >100 frames | 27 ± 1.2 ($n$ = 50) | This work |
| 30 | 128 × 112 | 512 × 256 | Cy3[c] 79 µW/µm$^2$ | 34 ± 1.0 ($n$ = 50) | 55 ± 1.7 ($n$ = 50) | This work |
| 45 | 128 × 64 | 256 × 256 | 5xCy3-Tf[f] 79 µW/µm$^2$ | 29 ± 0.73 ($n$ = 50) | 34 ± 0.96 ($n$ = 50) | This work |
| 110 | 128 × 16 | 128 × 80 | - | ND | ND | This work |
| 0.1–0.15[g] | 128 × 128 (EM-CCD; PALM) | | mEos2[h] | 11[i] | ND | Jones et al. (2011) |
| 0.6 | 2,048 × 256 (sCMOS; PALM) | | mEos3.2[j] | ≈22 | ND | Huang et al. (2013) |
| 1 | 1,024 × 1,024 PALM | 1,024 × 1,024 PALM | mEos3.2[k] | 29 ± 0.22 ($n$ = 2,940) | ND | Fujiwara et al. (2023) |
| 0.033 | 512 × 512 (EM-CCD; STORM) | | Alexa647[l] | 7.2[i] | ND | Dempsey et al. (2011) |
| 0.8 | 2,048 × 256 (sCMOS; STORM) | | Alexa 647[m] | ≈10 | ND | Lin et al. (2015) |
| 1.6 | 2,048 × 128 (sCMOS; STORM) | | Alexa 647[m] | ≈12 | ND | Lin et al. (2015) |

[a]Single-molecule localization precisions listed here were obtained with the 1024PCI-CMOS sensor, which was used in most of the experiments in the present research.

[b]Attached to the end of a short DNA duplex and immobilized on the glass surface.

[c]Covalently linked to the glass surface. In the basal PM, Cy3-DOPE observed under the same TIR illumination conditions at 79 µW/µm$^2$ with the camera operated at 10 and 30 kHz, the single-molecule localization precisions were 34 ± 5.1 and 51 ± 10 nm, respectively (Fig. S4 A). Compare these values with the data of Cy3 on the glass shown here (20 and 34 nm, respectively).

[d]Conjugated to MukB-Halo and observed in fixed *E. coli*.

[e]Covalently linked to the glass surface. When our standard laser illumination conditions (oblique illumination at a power density of 23 µW/µm$^2$ at the sample plane) were employed to observe Cy3-DOPE in the apical PM, with the camera operated at 10 kHz, the single-molecule localization precision was 49 ± 2.1 nm (Fig. S4 B, top). Compare this value with the data of Cy3 on the glass shown here (38 nm).

[f]Single transferrin (Tf) molecules bound by an average of 5.0 Cy3 molecules, immobilized on the glass surface.

[g]From this row to the bottom row, single-molecule localization precisions applicable for PALM and STORM experiments are listed, so under these observation conditions, most single molecules are photobleached within one or a few frames.

[h]Fused to clathrin light chain and observed in live BS-C-1 cells.

[i]Estimated from the reported FWHM value.

[j]Fused to clathrin light chain and observed in live HeLa cells.

[k]Fused to caveolin-1 and observed in live T24 cells.

[l]Conjugated to in vitro-prepared microtubules and immobilized on the glass surface.

[m]Covalently linked to antibodies and immobilized on the glass surface.

ND: Not Determined.

tens to hundreds of molecules simultaneously in a given field of view, like in a cell. This approach is particularly advantageous for investigating the interactions and assemblies of molecules in live cells and distinct behaviors of molecules in various regions of the cell. In addition, the highly sensitive ultrafast cameras can dramatically accelerate the data acquisitions for single-molecule localization microscopy methods, such as PALM and dSTORM, as described in the companion paper.

We examined the applicability of this camera system to observations of rapid molecular movements in the PM of live cells, by testing whether it can detect the hop diffusions of Cy3-labeled single molecules, a phospholipid and a transmembrane protein, transferrin receptor (TfR). Previously, the detection of hop diffusion of phospholipids and membrane proteins was only possible using 40-nm diameter gold probes and high-speed bright-field microscopy (single particle tracking), at a camera frame rate of 40 kHz (a time resolution of 25 µs [Fujiwara et al., 2002; Fujiwara et al., 2016; Murase et al., 2004] or 100 µs [Chai et al., 2022]). However, this could only be achieved in the apical (dorsal) PM, because the gold probes cannot enter the space between the basal (ventral) PM and the coverslip. We here report that ultrafast single-molecule imaging of fluorescently labeled molecules based on the developed ultrafast camera successfully reproduced the ultrafast gold-particle tracking data in the apical PM, while maintaining cell viability. Furthermore, when coupled with improved analysis methods developed here (Supplemental theories 1 and 2 in the Supplemental text), for the first time, we were able to directly evaluate the residency time distributions of membrane molecules within a compartment and perform a diffusion anomaly analysis over a 5 orders of magnitude time range (from 0.044 ms to 2 s). These novel findings provided further supports for the picket-fence model, based on the actin-based membrane skeleton meshes (fences) and transmembrane picket proteins anchored to and aligned along the fences (Fujiwara et al., 2002; Fujiwara et al., 2016; Morone et al., 2006).

With the development of ultrafast SFMI and tracking, we could experimentally address whether the molecular dynamics and compartmentalization in the basal PM are similar to those in the apical PM, using the fluorescently labeled phospholipid and transmembrane proteins. Our results demonstrate that both phospholipids and transmembrane proteins, TfR and EGF receptor (EGFR), undergo hop diffusion among the compartments in the bulk basal PM, with virtually the same compartment sizes (108 nm) and hop frequencies (once every 10 and 24 ms on average for the phospholipid and transferrin receptor, respectively) as those in the apical PM. These results indicate that, although the basal PM is located close to the coverslip, within 40 nm from the coverslip, and binds to it via focal adhesions in the basal PM, the basic structures and molecular dynamics in the bulk basal PM are very similar to those in the apical PM. Furthermore, in the companion paper (Fujiwara et al., 2023), we demonstrate that membrane molecules undergo hop diffusion even inside the focal adhesion, a micron-scale structure formed in/on the basal PM responsible for cellular attachment to and migration in/on the extracellular matrix.

Our new ultrafast camera system enables the data acquisition for PALM and dSTORM at accelerated rates of up to 1 kHz, which

is ≈30× faster than standard rates, with 29 and 19 nm localization precisions, respectively, for large view-fields that can include an entire live cell. These developments will be described in the companion paper (Fujiwara et al., 2023).

## Results

### Basic design of the ultrafast camera system

Our camera system is composed of a microchannel-plate image intensifier (Fig. 1, A a), a high-speed complementary metal-oxide semiconductor (CMOS) sensor (Fig. 1, A b), and an optical-fiber bundle, which couples the phosphor screen of the image intensifier to the CMOS sensor chip (Fig. 1, A c; Materials and methods). The image intensifier (Hamamatsu, V8070U-74) comprises a third-generation GaAsP photocathode with a quantum efficiency of 40% at 570 nm (Fig. 1, A a α), a three-stage microchannel plate allowing maximal gain over $10^6$, with virtually undetectable non-linear noise increase (Fig. 1, A a β), and a P46 phosphor screen with a decay time of ~0.3 µs from 90 to 10% (Fig. 1, A a γ). A CMOS sensor (the sensor developed for a Photron 1024PCI camera) with a global shutter exposure was used (Table 1).

A straight (1:1) optical-fiber bundle (Fig. 1, A c) was directly adhered to the phosphor screen of the image intensifier on one side (Fig. 1, A a γ, input side) and to the CMOS sensor on the other side (Fig. 1, A b, output side). This approach enhanced the signal reaching the CMOS sensor by a factor of 5–10, as compared with the lens coupling used for our standard camera system employed for SFMI (Suzuki et al., 2012; Kinoshita et al., 2017). The electrons generated at the CMOS sensor were transferred, amplified (Fig. 1, A d), and digitized (Fig. 1, A e) in 64 and 8 parallel paths, respectively, and subsequently transferred to the host PC.

### Basic conceptual strategy to address the large readout noise of the CMOS sensor

We opted to use a CMOS sensor instead of the more common scientific CMOS (sCMOS) sensor in fluorescence microscopy due to its higher frame rates, which can reach 10—100 kHz (10,000—100,000 frames per second), compared to sCMOS sensors. However, the readout noise of the CMOS sensor is generally much greater than that of the sCMOS sensor by factors of several 10s, making CMOS sensors rarely used in fluorescence microscopy. Therefore, to employ the CMOS sensor for SFMI, the problem of high levels of readout noise had to be solved.

Our basic conceptual strategy to address this problem was the following: we should place an amplifier before the noisy detector, so that both the input signal and noise generated at the photocathode can be amplified (they will be amplified in the same way by the amplifier) at least to the level that the amplified noise (be mindful that this must be the noise and not the signal) becomes comparable to the noise that the detector generates. For the CMOS camera (noisy detector), we placed an image intensifier (more specifically the microchannel plate; a-β in Fig. 1 A), before the CMOS sensor (b in Fig. 1 A), so that the noise (including the background) generated by the photocathode of the image intensifier (the first step of light detection; a-α in Fig. 1 A)

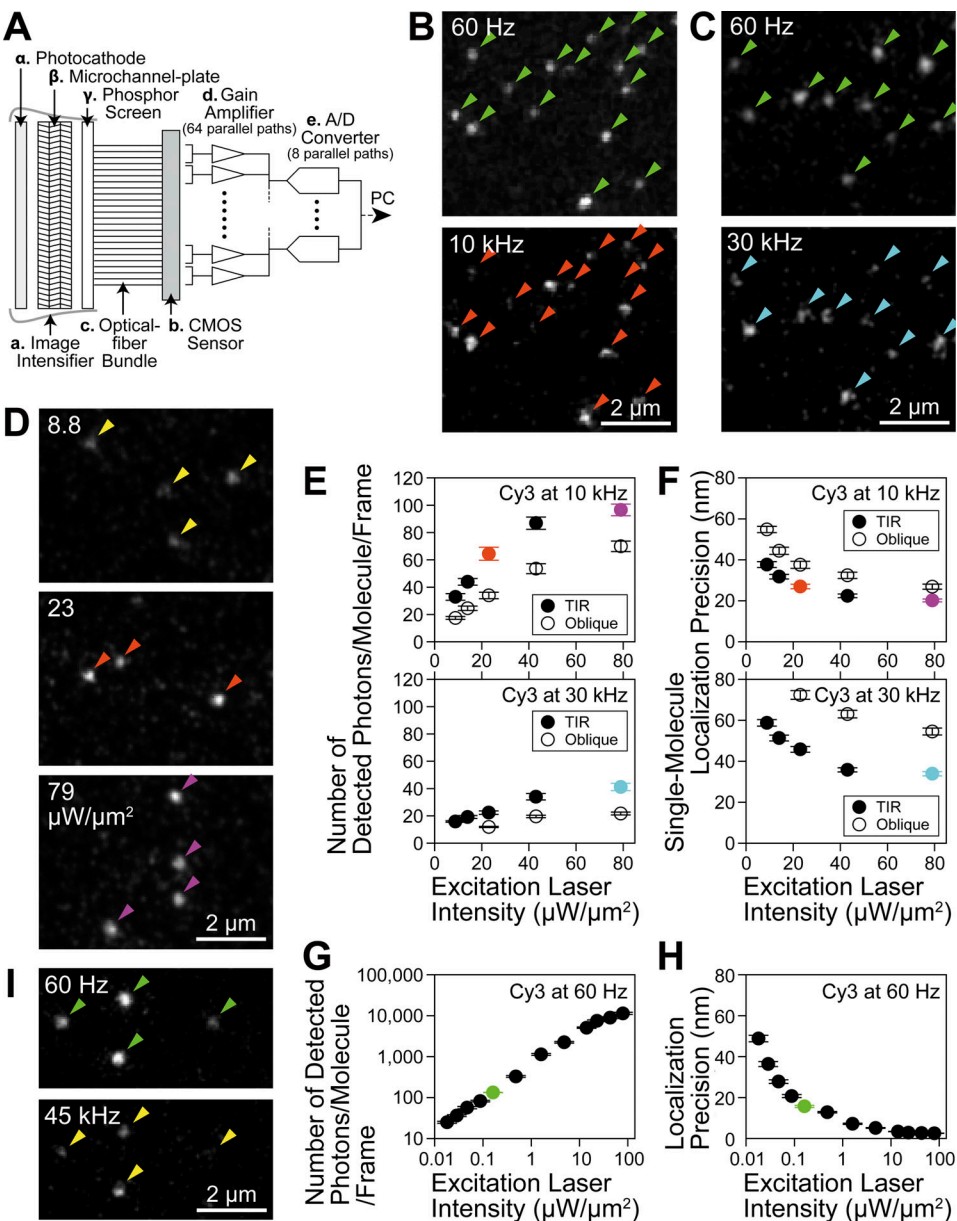

**Figure 1. Basic design of the ultrahigh-speed intensified CMOS camera system and ultrafast single fluorescent-molecule imaging and tracking (SFMI) of Cy3 molecules immobilized on coverslips, performed to establish suitable conditions for ultrafast SFMI.** To obtain the data shown here, total internal reflection (TIR) illumination was employed. The mean ± SEM and/or the median values are provided in the figures throughout this report. **(A)** Schematic diagram of the camera system. See Materials and methods. **(B and C)** All of the single Cy3 molecules detected at 60 Hz were also detectable at 10 (B) and 30 (C) kHz (the same field of view). Typical snapshots are shown. **(D)** Typical snapshots of single Cy3 molecules excited at various laser powers, observed at 10 kHz. **(E)** The number of detected photons/molecule/frame emitted from single Cy3 molecules during frame times of 0.1 ms (10 kHz) and 0.033 ms (30 kHz; n = 50 Cy3 molecules for each measured point; the same for F, G, and H), plotted against the excitation laser power density. See the caption to Fig. S3 A. **(F)** The localization errors for single Cy3 molecules imaged at 10 and 30 kHz, plotted against the excitation laser intensity. Orange, cyan, and purple data points are color-matched with the arrowheads in B, C, and D. **(G and H)** The number of detected photons/molecule/frame (G) and the localization error (H) for single Cy3 molecules imaged at 60 Hz, plotted as a function of the excitation laser power density. See the caption to Fig. S2 D. The green data points are color-matched with the arrowheads in B, C, and I. **(I)** Virtually all of the single 5xCy3-Tf molecules (3–8 molecules of Cy3 bound to Tf) were detectable at 45 kHz (compare with the 60-Hz image); typical snapshots.

is amplified at least to a level comparable to the readout noise of the CMOS sensor (signal at the photocathode is amplified by the same factor as the noise, but in the present argument, the amplification of the noise is the critical issue). Here, we explain why this basic strategy would work.

We denote the average intensities of the signal and noise + background at the photocathode of the image intensifier as $S_p$ and $N_p$, respectively, that of the readout noise as $Nr$, and the image intensifier amplification gain as $G$. Note that when using the CMOS sensor for SFMI, generally $N_p << Nr$.

The signal-to-noise (including the background intensity) ratio (S/N) of the output from the CMOS sensor can be written as a function of $G$,

$$\text{Output } S/N(G) = S_p \times G / [N_p \times G + Nr]$$
$$= S_p / [N_p + Nr/G]. \quad (1)$$

This is because both the signal and noise at the photocathode stage are amplified by a factor of $G$, neglecting the noise generated by the amplifier. Note that when there is no gain; i.e., $G = 1$,

$$\text{Output } S/N(G = 1) = S_p / [N_p + Nr] \approx S_p / Nr, \quad (2)$$

because $N_p \ll Nr$. Namely, the S/N ratio of the CMOS sensor output is dominated by the large readout noise of the CMOS sensor $Nr$, and we will not be able to detect the single-molecule signal, $S_p$, due to the large $Nr$ (i.e., $S_p/N$ ($G = 1$) <1).

With an increase in gain $G$, $Nr/G$ becomes much smaller than $N_p$ (or $Nr \ll N_p \times G$; namely, when the noise at the photocathode is amplified so that the amplified noise becomes greater than the readout noise of the CMOS sensor $Nr$), and Eq. 1 can be written as

$$\text{Output } S/N(G \gg Nr/N_p) = S_p / [N_p + \varepsilon], \quad (3)$$

where $\varepsilon$ represents the very small value of $Nr/G$. Eq. 3 shows that with an increase of the amplifier gain $G$, the S/N of the output signal from the CMOS sensor is no longer dominated by $Nr$, and thus increased close to that at the photocathode ($S_p/N_p$). Eq. 3 is consistent with the general truth that amplifier placement cannot increase the output S/N to more than the input S/N. Note that, in this argument, the key is not to compare $S_p$ with $Nr$, but to compare $N_p$ (or $N_p \times G$) with $Nr$.

Namely, when the detector itself is a large noise generator and the limiting factor for the S/N of the entire system, this problem can be minimized by placing an amplifier before the noisy detector. Eqs. 1 and 3 indicate that larger gains are preferable. However, since the amplifier itself can also generate noise and since the full-well capacity of each pixel of the CMOS sensor is limited (i.e., to make this strategy work, the dynamic range of the camera system must be quite large), there is an upper limit for the amplifier gain.

When we set the gain $G$ so that $N_p \times G = Nr$ (the noise at the photocathode $N_p$ is amplified to become comparable to the readout noise $Nr$); i.e., if we set $G = Nr/N_p$,

$$\text{Output } S/N(G = Nr/N_p) = [1/2] \times (S_p/N_p). \quad (4)$$

Namely, under these conditions, the output S/N of the CMOS sensor is half the S/N at the photocathode output. This provides a rule of thumb that the noise at the photocathode ($N_p$) should be amplified at least to a level comparable to the readout noise of the CMOS sensor ($Nr$). For the actual values of $Nr$ and $N_p$, see the subsection "Ultrahigh-speed intensified CMOS camera system: Design and operation, Basic design concept of the camera system, (b) The use of an image intensifier" in Materials and methods.

Under these conditions, the sensitivity limitation for this camera system is determined by the quantum yield of the photosensor, which is ≈0.4 at ≈580 nm (for dyes including Cy3,

Alexa555, JF549, and TMR). The typical photosensor quantum yield of the sCMOS cameras frequently used for SFMI is often ≈0.95, whereas the frame rate can go up to only ≈800 Hz for a frame size of 256 × 256 pixels (for example, Hamamatsu ORCA-Fusion BT). Therefore, in essence, we increased the speed by a factor of more than 12.5 (10 kHz) and up to 56.3 (45 kHz) by sacrificing the sensitivity by a factor of ≈2.4.

Apart from the basic concept, to determine the useful amplifier gain for experiments, we obtained the actual image readout noise, images of single photons acquired as a function of the intensifier gain, stochastic gain variations (fluctuations) of the image intensifier, and the probability of detecting single photons, as described in Fig. S1 and the subsection "Ultrahigh-speed intensified CMOS camera system: Design and operation, Basic design concept of the camera system, (b) The use of an image intensifier" in Materials and methods. Based on our findings, we selected an overall electron amplification gain of 8,100×, which provides a 90.0% probability of detecting a photon-converted electron emitted at the photocathode of the image intensifier (Fig. S1; Materials and methods).

Our developed camera system is typically operated at frame rates of 10 and 30 kHz (100- and 33-μs resolutions) with frame sizes of 256 × 256 and 128 × 112 pixels, respectively. However, the camera can be operated at faster rates with reduced image sizes. For example, it can be operated at 45 kHz or every 22 μs, with a frame size of 7 × 3.5 μm². Virtually all of the single Cy3 molecules immobilized on coverslips that were observed at 60 Hz were also detectable at frame rates of 10 and 30 kHz (Fig. 1, B–D).

## Ultrafast imaging of many single molecules simultaneously at 10 and 30 kHz

With the developed ultrafast camera system, the crucial issue for ultrafast SFMI is the number of detected photons emitted from a single fluorescent molecule during a single frame time. This is because the single-molecule localization precision is fundamentally determined by the number of detected photons/molecule during the camera's single frame time, as described by the Mortensen equation (Mortensen et al., 2010; refer to the legend for Fig. S2). We have thus assessed whether a sufficient number of photons can be emitted from single fluorescent molecules of various fluorophores during the 33 and 100 μs frame times of the developed camera system (30 and 10 kHz, respectively; neglecting a readout time of 812 ns). The plots of single-molecule localization precision vs. the number of detected photons/molecule during a single-frame time (using fluorescent molecules bound to the coverslip) were consistent with the Mortensen equation for all dye molecules examined here (Fig. S2). These plots demonstrated that the number of detected photons/molecule/frame and the single-molecule localization precision were enhanced with an increase of the excitation laser intensity, but to a limited extent (Fig. 1, E and F; and Fig. S3). This occurs because at higher excitation light intensities, the number of photons emitted from a single fluorescent molecule during a frame time is limited by the triplet bottleneck saturation (see the subsection "Estimation of the number of photons that can be emitted by a single Cy3 molecule during 0.1 ms: Triplet

bottleneck saturation" in Materials and methods). The saturation became apparent from around 23 µW/µm² for Cy3 (Fig. 1, E and F; and Fig. S3). Among the eight dyes we tested, Cy3 exhibited the lowest tendency to saturate (Fig. S2 B and Fig. S3 A). Therefore, we primarily employed Cy3, but also utilized tetramethylrhodamine (TMR) as a membrane-permeable probe.

The best single-molecule localization precision achieved with the Cy3 fluorophore at a frame integration time of 0.1 ms (10-kHz frame rate) was 20 ± 0.71 nm, using total internal reflection (TIR) laser illumination at a density of 79 µW/µm² at the sample plane (n = 50 molecules; mean ± standard error of means [SEM] are given throughout this report; Fig. 1 F; and Fig. S2, A and B; and Fig. S3, B and C). At 30 kHz, since fewer photons were emitted from a Cy3 molecule during a frame time of 33 µs, the localization precision was worse (34 ± 1.0 nm; n = 50 molecules; Fig. 1 F and Fig. S2 A and Fig. S3, B and C; and Table 1). The best single-molecule localization precision achieved with this camera system was 2.6 ± 0.099 nm, with a maximum number of detected photons/molecule/frame of 11,400 ± 700, at a frame integration time of 16.7 ms (60-Hz frame rate) in which most of the Cy3 molecules were photobleached (within a single frame; n = 50 molecules; Fig. 1, G and H; and Fig. S2, C and D; summarized in Table 1).

Taken together, we conclude that the observation frame rates of 10—30 kHz (with localization precisions of 20—34 nm, using the view-fields of 256 × 256—128 × 112 pixels [14.1 × 14.1—7.1 × 6.2 µm²]), represent the ultimate fastest frame rates employable for single-molecule tracking in living cells, using the currently available fluorescent molecules. Meanwhile, the camera itself could be operated much faster. The Photron 1024PCI CMOS sensor, which was used most extensively here, can be operated at 10 kHz (256 × 256 pixels), 45 kHz (128 × 64 pixels; or 256 × 256 pixels using the newer SA1 sensor), and 110 kHz (128 × 16 pixels; summarized and compared with the results with other cameras in Table 1; Materials and methods, "Ultrahigh-speed intensified CMOS camera system: Design and operation"). Therefore, the camera is no longer the limitation for the faster frame rates, and with the future development of fluorescent dyes with higher throughputs, we should be able to perform single-molecule tracking at even better time resolutions. In fact, single transferrin (Tf) molecules bound by an average of 5.0 Cy3 molecules (5xCy3-Tf) adsorbed on the coverslip could be imaged and tracked at 45 kHz, with localization precisions of 30 (38) and 29 (34) nm, using TIR (oblique-angle) illuminations of 43 and 79 µW/µm², respectively (Fig. S2 A; and Fig. S3, B and C; and Table 1).

### TIR and oblique illuminations for ultrafast SFMI

In this study, we employed both TIR and oblique-angle laser illuminations. TIR illumination is useful for observing single molecules in the basal (ventral, bottom) PM by suppressing the signals from the cytoplasm and the enhanced evanescent electric field for the same laser intensity. The TIR illumination provides better localization precisions than the oblique-angle illumination at the same laser intensities (27 vs. 38 nm; Fig. 1 F).

On the other hand, the oblique-angle illumination is more versatile: it can be used to illuminate the apical PM, endomembranes, and cytoplasm, which cannot be accomplished using the TIR illumination. Particularly, in this research, since we hoped to compare the present single fluorescent-molecule tracking results of the phospholipid diffusion with our previous 40-kHz single gold-probe tracking data obtained in the apical PM (Fujiwara et al., 2002; Fujiwara et al., 2016; Murase et al., 2004), we had to use the oblique-angle illumination. Therefore, as our "standard test conditions" in the present research (the end of the caption to Fig. S3), we employed the oblique-angle illumination with a laser intensity of 23 µW/µm² at the sample plane (the dye saturation starts around this laser intensity; Fig. 1, E and F; and Fig. S3), despite its worse localization precisions as compared with the TIR illumination.

### Trajectory length and cell viability under the standard conditions

Under the standard test conditions (oblique-angle illumination at 23 µW/µm²), single Cy3 molecules immobilized on the coverslip could be observed at 10 kHz with an average signal-to-noise ratio (SNR) of 2.5 ± 0.11 (Fig. 2, A and B; and Video 1), and the fractions of the trajectories with durations (uninterrupted length of the trajectories) longer than 100, 300, and 1,000 frames (after the 3-frame gap-closing, which neglects the non-detectable periods lasting for three frames or less) were 14, 2.9, and 0.31%, respectively, among all obtained trajectories (Fig. 2 C). Therefore, performing single-molecule tracking for 100—300 frames (10—30 ms at 10 kHz) is quite practical. Meanwhile, under the conditions for the best localization precision with TIR illumination of 79 µW/µm² (20 ± 0.71 nm), the fractions were 1.6, 0.13, and 0.0%, respectively (Fig. 2 C), making it difficult to obtain trajectories longer than 100 frames (for the summary and comparison with results using other cameras, see Table 1).

These results indicate that, for successful single-molecule tracking, a proper compromise between the localization precision and the trajectory length is necessary. This is one of the major differences between single-molecule tracking and SMLM. SMLM basically requires only one localization for a single molecule (observations lasting only for a single frame) before photobleaching, while single-molecule tracking needs many localizations to acquire long trajectories (essentially, longer is better) before photobleaching/photoblinking.

Under the standard test conditions, the laser illumination (oblique-angle at 23 µW/µm² of the 532-nm line) for 1 min hardly affected the cell viability, although half of the T24 cells did not survive after 10-min irradiation (Fig. 2, D and E; human epithelial T24 cells were employed throughout this research and their microscope observations were always conducted at 37°C). Since all our ultrafast measurements were completed within 5 s, we conclude that the toxic effect of the illumination laser is minimal for our observations. In a previous report (Wäldchen et al., 2015), the light toxicity was found strongly dependent on the cell type, and our result using T24 cells is similar to their result using HeLa cells. However, the direct comparison of the present result and the previous result is difficult due to the differences in the viability test method, laser wavelength (532 vs. 514 nm), and illumination durations (1 and 10 vs. 4 min; our

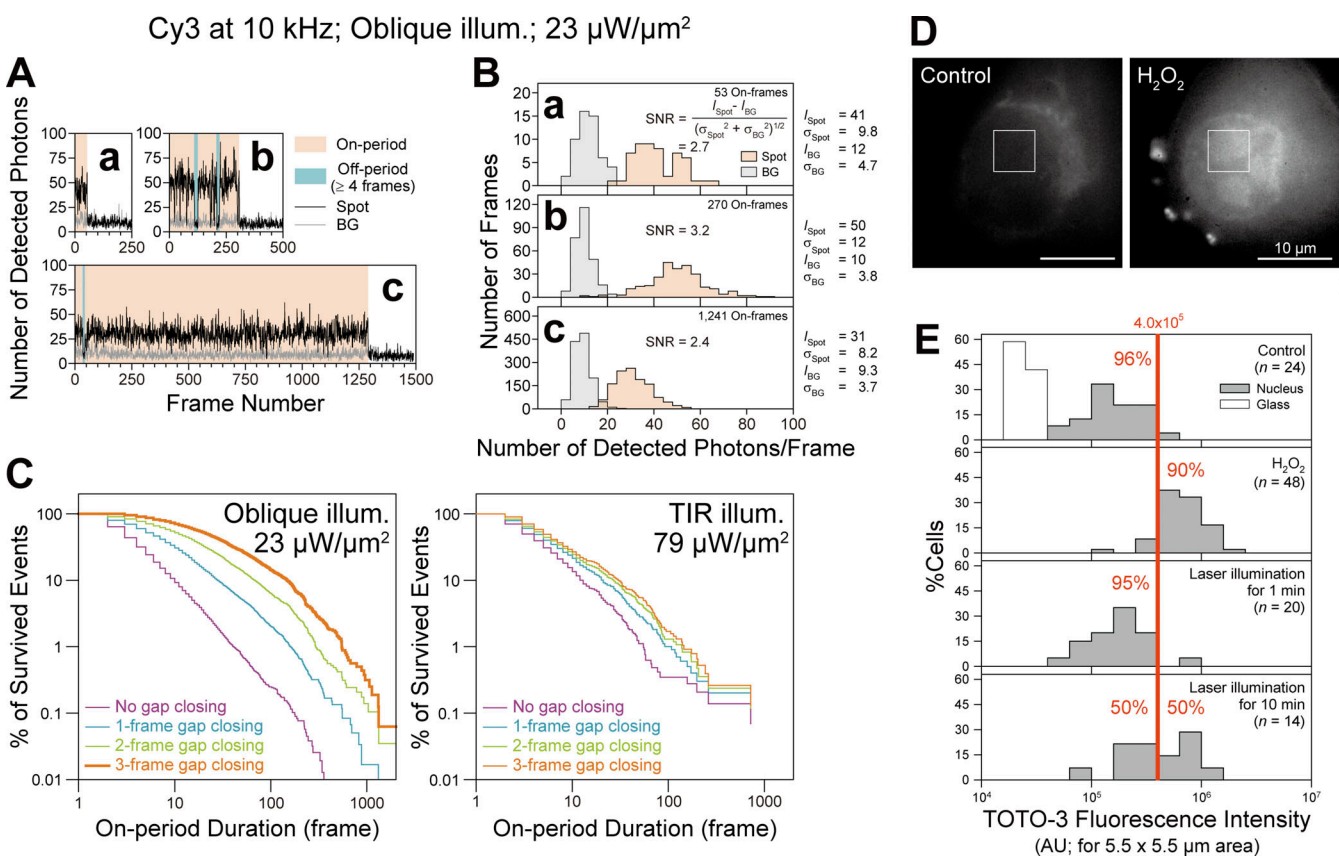

Figure 2. **Trackable durations of single Cy3 molecules under two typical laser illumination conditions and minimal photo-induced damage to cells during ultrafast SFMI. (A–C)** Single Cy3 molecules immobilized on a coverslip were observed. The trackable durations are limited by photoblinking/photobleaching, stochastic fluctuations of the signal, and photon shot noise. **(A a–c)** Typical time-dependent fluorescence signal intensities (number of detected photons/0.1-ms-frame) of three single Cy3 molecules (a, b, and c) excited by oblique-angle illumination at 23 µW/µm², and observed at 10 kHz (background signal in gray). See Materials and methods for details. **(B a–c)** The signal-to-noise ratios (SNRs) for the images of the three molecules (a, b, and c) shown in A fall in the range between 2.4 and 3.2. See Materials and methods. **(C)** The distributions of the durations of the on-periods and those after neglecting the off-periods (non-emission periods) lasting for 1, 2, or 3 frames (gap closing; see Materials and methods) of single Cy3 molecules immobilized on a coverslip and excited by oblique-angle illumination at 23 µW/µm² (standard conditions; left) or by TIR illumination at 79 µW/µm² (right), observed at 10 kHz (totals of 264 and 593 molecules, respectively). The three-frame gap closing was employed in single-molecule tracking under the standard test conditions (thick orange curve; left). **(D and E)** Photo-induced damage to the cells during ultrahigh-speed SFMI is very limited under our standard experimental conditions. Cell viability was examined by staining with 1 µM TOTO-3 iodide, which selectively stains dead cells, at 37°C for 5 min, and then observing the stained cells using epi-illumination with a 594-nm laser. **(D)** Representative images of the nuclei stained with TOTO-3 iodide. Control, a reference image of a living cell (n = 24 images). H₂O₂, a reference image of a dead cell after the treatment with 100 µM H₂O₂ at 37°C for 1 h (n = 48 images). **(E)** Histograms showing the fluorescence intensity of TOTO-3 in the 5.5 × 5.5-µm area inside the nucleus (see the square box in D; n = the number of examined cells). Top: Live cells without laser illumination (negative control). Second: Dead cells after the H₂O₂ treatment (positive control). Based on the results of the negative and positive controls (top and second boxes, respectively), a threshold fluorescence intensity of 4.0 × 10⁵ (arbitrary unit = AU) was selected to categorize the live and dead cells (96% [90%] of the negative [positive] control cells were categorized as alive [dead]). Third and Fourth: Cells were subjected to illumination under our typical 10-kHz single Cy3 molecule imaging conditions (oblique-angle 532-nm laser illumination at 23 µW/µm²) for 1 and 10 min, respectively, which are longer by factors of 12 and 120 than our longest illumination duration of 5 s/cell (10 500-ms image sequences = 5 × 10⁴ frames). We concluded that the light-induced damage to the cells is insignificant under our standard experimental conditions.

result suggests non-linear dependence of the viability on the illumination duration).

**Testing the ultrafast camera system using PM molecules undergoing hop diffusion 1: Qualitative observations**

As stated in the Introduction, we previously detected non-Brownian diffusion (hop diffusion) of phospholipid and transmembrane protein molecules in the PM, using large (40-nm diameter) gold particles as a probe (Fujiwara et al., 2002; Fujiwara et al., 2016; Murase et al., 2004; Kusumi et al., 2005; Kusumi et al., 2012a; Kusumi et al., 2012b; also see Sheetz, 1983).

This was made possible by improving the time resolution of single-particle imaging using bright-field microscopy down to 25 µs (Fujiwara et al., 2002, 2016; Murase et al., 2004). The hop diffusion is influenced by modulating the actin-based membrane-skeleton meshes (Fig. 3 A). Therefore, we concluded that the entire apical PM is compartmentalized, by the steric hindrance of actin-based membrane-skeleton meshes (fences) and the steric hindrance + hydrodynamic friction-like effects from rows of transmembrane-protein pickets anchored to and aligned along the actin fence (Fig. 3 A). In the compartmentalized PM, both transmembrane proteins and lipids undergo short-term

confined diffusion within a compartment plus occasional hop movements to an adjacent compartment, which was termed hop diffusion (Kusumi et al., 2005; Kusumi et al., 2012a; Jacobson et al., 2019).

Each gold-tagged molecule in the PM exhibited two distinct diffusion coefficients: the microscopic diffusion coefficient ($D_{micro}$), which characterizes the unhindered diffusion within a compartment, and the macroscopic diffusion coefficient ($D_{MACRO}$), which is largely determined by the compartment size and the hop frequency across intercompartmental boundaries composed of the picket-fence. $D_{MACRO}$ is significantly smaller than $D_{micro}$.

The compartment sizes were in the range of 40–230 nm depending on the cell type (≈110 nm in human epithelial T24 cell line used here), and the dwell lifetimes of membrane molecules within a compartment are in the range of several to a few 10s of milliseconds (Murase et al., 2004; Fujiwara et al., 2016). Therefore, the hop diffusion of membrane molecules in the apical PM appeared to be quite suitable for testing SFMI using the developed ultrafast camera system. Indeed, this project was originally undertaken to develop a fast single fluorescent-molecule imaging method to observe the hop diffusion of membrane molecules in the PM. We sought to examine whether we could detect the hop diffusion of L-α-dioleoylphosphatidylethanolamine (DOPE) conjugated with Cy3, Cy3-DOPE, at a time resolution of 0.1 ms (10 kHz, 333 times faster than normal video rate) and transferrin receptor (TfR) tagged with Cy3-conjugated transferrin (average dye to protein molar ratio of 0.2) at a frame rate of 6 kHz (0.167 ms resolution; 200 times faster than normal video rate; slowed from 10 kHz because the hop rate of TfR was expected to be slower than that of Cy3-DOPE). The observations were made in the apical PM using the oblique-angle illumination (Fig. 3, B–D; and Videos 2, 3, and 4), for making the direct comparisons with the previous single gold-particle tracking data. All of the cell experiments reported here were performed at 37°C using human epithelial T24 cells (previously called ECV304 cells).

Virtually all of the single-molecule images and trajectories (Fig. 3, B–D; and Videos 2, 3, and 4) gave the impression that the Cy3-DOPE and TfR molecules underwent rapid diffusion within a confined domain of ≈100 nm and occasionally moved out of this domain, became confined again at the place where it moved to, and repeated such behaviors. These typical behaviors appear to reproduce the movement of gold-tagged DOPE and TfR quite well.

**Testing the ultrafast camera system using PM molecules undergoing hop diffusion 2: Quantitative and statistical analyses**
Apart from the subjective impression, the trajectories of single fluorescent molecules were examined quantitatively and statistically, using the plot of mean-square displacement (MSD) vs. time interval ($\Delta t$; Qian et al., 1991; called the MSD-$\Delta t$ plot, which also provides single-molecule localization precisions; Fig. 4 A and Fig. S4). The results were compared with those obtained by single gold-particle imaging.

Based on the MSD-$\Delta t$ plot (Fig. 4, A a), each trajectory could be classified into a suppressed-, simple-Brownian-, or directed-

diffusion mode (Kusumi et al., 1993; see the caption to Fig. 4, B a). More than three-quarters of the Cy3-DOPE and TfR trajectories obtained in the intact apical PM at 10 and 6 kHz (0.1 ms and 0.167 ms resolutions, respectively) were categorized into the suppressed-diffusion mode (top boxes in Fig. 4, B b; also see Table 2), reproducing the results obtained by single gold-particle tracking at 40 kHz (Table 2, in which all of the statistical parameters and the statistical test results for the diffusion parameters are also summarized).

Meanwhile, in the f-actin-depleted blebbed PM, almost all of the Cy3-DOPE and TfR trajectories, obtained at 10 and 6 kHz, respectively, were classified into the simple-Brownian-diffusion mode (Fig. 4, A c and B b; and Table 2), again consistent with the data obtained by single gold-particle tracking in the blebbed PMs of NRK cells (Fujiwara et al., 2002, 2016). As concluded previously, these results indicate that the actin-based membrane skeleton is involved in inducing the suppressed diffusion of both phospholipids and transmembrane proteins (Fujiwara et al., 2002; Fujiwara et al., 2016; Murase et al., 2004; Hiramoto-Yamaki et al., 2014).

The MSD-$\Delta t$ plot obtained for each trajectory was then fitted, using an equation based on the hop-diffusion theory (Kenkre et al., 2008; called "hop-diffusion fitting"; Fig. 4 A; see "Supplemental theory 2. Hop-diffusion fitting: the function describing the MSD-$\Delta t$ plot for particles undergoing hop diffusion" in the Supplemental text). The hop-diffusion fitting provided two diffusion coefficients for each trajectory observed in the PM: $D_{micro}$ and $D_{MACRO}$. The presence of two diffusion coefficients for a single trajectory observed in the PM and only one diffusion coefficient in the actin-depleted blebbed PM (Fig. 4 C) again reproduced the observations obtained using single gold-particle imaging (Fujiwara et al., 2002; Fujiwara et al., 2016).

The hop-diffusion fitting of each trajectory obtained in the intact PM also provided the average compartment size $L$ for each trajectory (Fig. 4 D and Table 2). Importantly, the $L$s for Cy3-DOPE and TfR were quite similar, with median values of 101 and 103 nm, respectively (non-significant difference; in the following discussions and calculations, we will use a compartment size of 100 nm). This result suggests that the underlying mechanisms for confining phospholipids and transmembrane proteins are based on the same cellular structures, perhaps by the actin-based membrane-skeleton meshes (fences) and the transmembrane picket proteins bound to and aligned along the actin-fence (Fig. 3 A). The compartment sizes $L$ found here (median sizes of 101 and 103 nm for Cy3-DOPE and TfR, respectively) are quite comparable to those previously detected using gold-conjugated molecules in the same T24 cells (110 and 100 nm for gold-tagged DOPE and TfR, respectively; Table 2).

Taken together, our ultrafast SFMI based on the ultrafast camera system developed here could successfully detect the hop diffusion of membrane molecules, which was previously detectable only using gold probes and high-speed bright-field microscopy. Therefore, the ultrafast SFMI developed here is likely to be suitable to observe the fast molecular dynamics occurring in living cells.

Here, we add a remark on the $D_{micro}$s for Cy3-and gold-labeled molecules in the PM. The model of hop diffusion in the

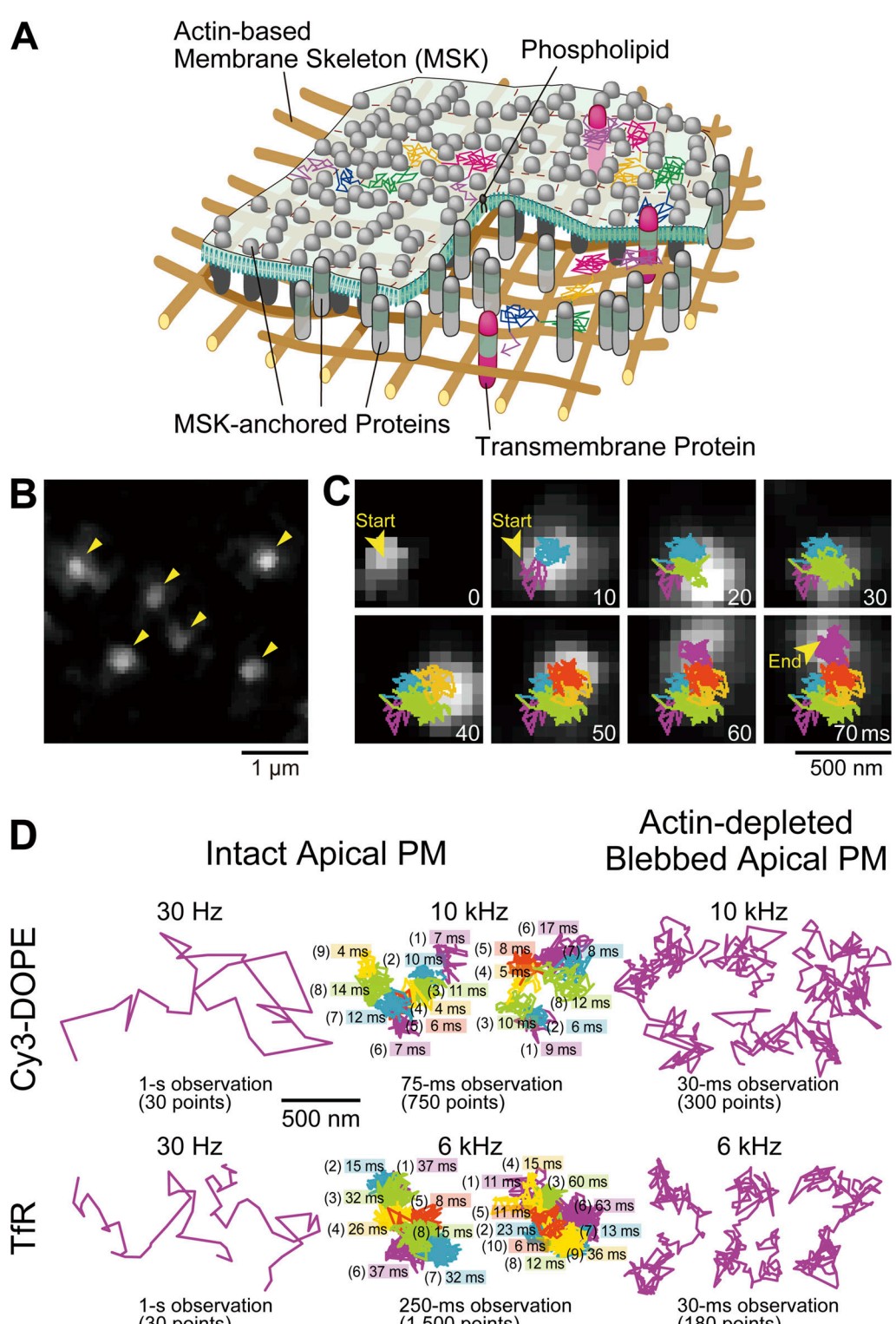

Figure 3. **Ultrafast SFMI of Cy3-labeled DOPE and TfR in the apical PM. (A)** Schematic drawing showing that membrane molecules undergo hop diffusion in the PM, which is compartmentalized by the actin-based membrane-skeleton (MSK) meshes (fences; brown mesh) and rows of transmembrane-protein pickets anchored to and aligned along the actin fence (gray molecules). **(B)** A typical snapshot of single Cy3-DOPE molecules in the apical PM observed under our standard conditions, with an integration time of 0.1 ms. **(C)** A representative image sequence of a single Cy3-DOPE molecule diffusing in the apical PM, recorded every 0.1 ms (shown every 100 image frames = every 10 ms). The colors in the trajectory indicate the diffusion in different plausible compartments (order of appearance: purple, blue, green, orange, red, and then back to purple; the same color order is used throughout this report), detected by the TILD analysis software (Fig. S5). **(D)** Typical single-molecule trajectories of Cy3-DOPE and Cy3-Tf-TfR in intact and actin-depleted blebbed apical PMs, recorded at normal video rate and enhanced rates. The order of the compartments that the molecule visited (parenthesized integers) and the residency times there, as determined by the TILD analysis, are indicated (intact PMs).

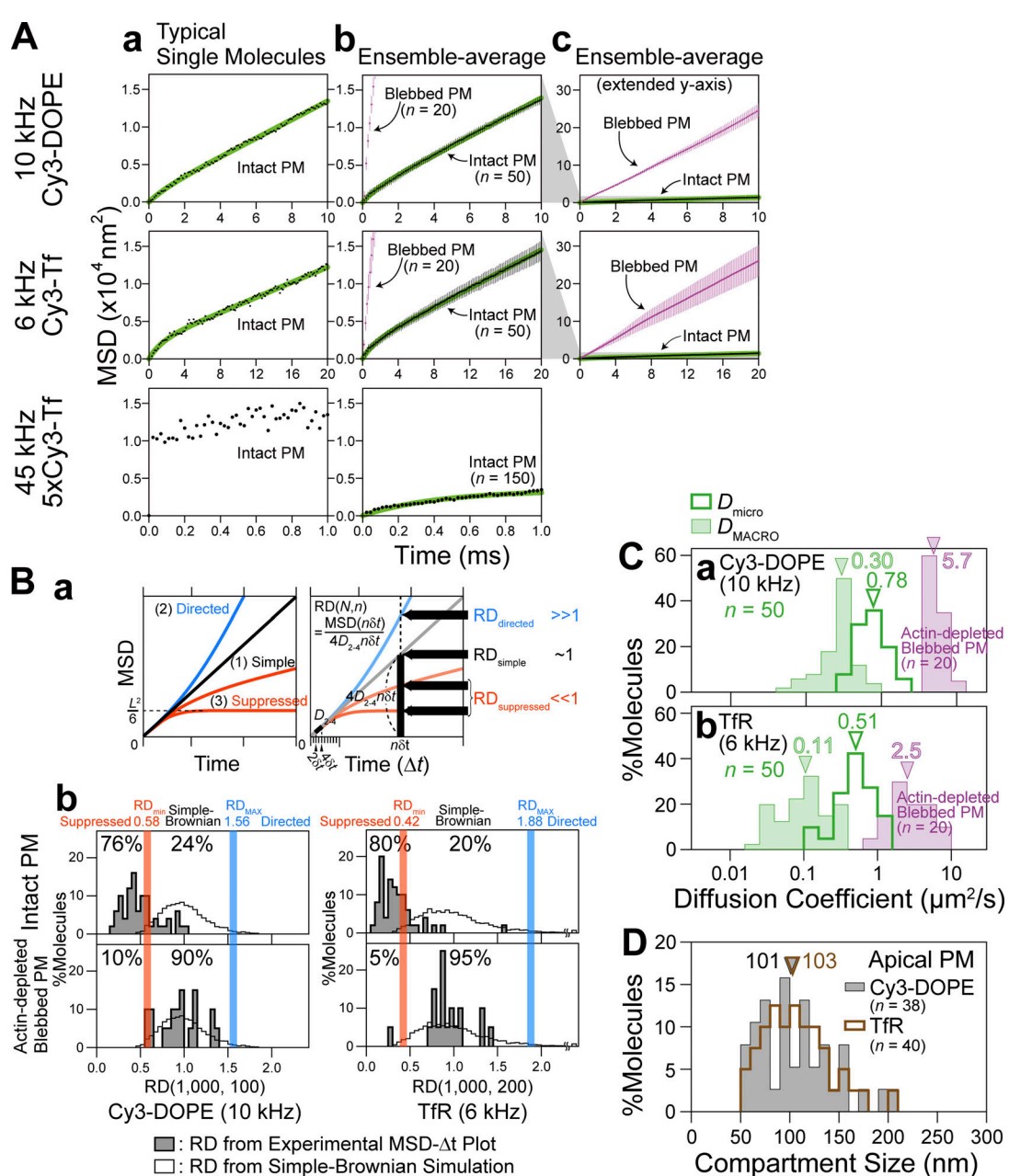

**Figure 4. Ultrafast SFMI can detect hop diffusion of Cy3-DOPE and TfR in the apical PM, previously found by ultrafast single gold-particle tracking. (A a–c)** Typical MSD-Δt plot for a single molecule (a), and the ensemble-averaged MSD-Δt plots with two different y-axis scales (b and c: 20× of b) in the intact apical PM (black) and the blebbed apical PM (purple). The MSD-Δt plots of the top and middle boxes in (b) are those after subtracting $4\sigma_{xy}^2$, shown in Fig. S4 ($\sigma_{xy}$ = single-molecule localization precision). Only in the typical MSD-Δt plot for a single 5xCy3-Tf molecule bound to TfR obtained at 45 kHz (a, bottom), $4\sigma_{xy}^2$ was not subtracted due to large errors in its estimation. Green curves are the best-fit functions for the hop-diffusion fitting (top and middle rows) and confined-diffusion fitting (bottom row for 45 kHz observations of 5xCy3-Tf). **(B)** Approximately 80% of Cy3-DOPE and TfR undergo suppressed diffusion in the intact apical PM, whereas >90% of them undergo simple-Brownian diffusion in the actin-depleted blebbed PM (shaded histograms in b), as revealed by the method for the statistical classification of each single-molecule trajectory into a suppressed-, simple-Brownian-, or directed-diffusion mode. **(B a)** The basic idea for the classification of trajectories into suppressed, simple-Brownian, and directed diffusion: the parameter RD (relative deviation) describes the extent to which the observed diffusion deviates from simple-Brownian diffusion at a time sufficiently later from time 0; i.e., the actual MSD divided by the calculated MSD from the short-term diffusion coefficient ($D_{2-4}$) assuming simple-Brownian diffusion. See Materials and methods. The RD value is <<, ≈, or >> 1, when the molecules are undergoing suppressed, simple-Brownian, or directed diffusion, respectively (Fujiwara et al., 2002; Kusumi et al., 1993; Murase et al., 2004; Suzuki et al., 2005). The suppressed-diffusion mode includes both the confined-diffusion and hop-diffusion modes. **(B b)** Classification of individual trajectories based on the RD histograms for simulated simple-Brownian particles (open bars; n = 5,000). The RD values giving the 2.5 percentiles of the particles from both ends of the distribution, referred to as $RD_{min}$ and $RD_{MAX}$, were obtained (red and blue vertical lines, respectively). Each experimental single-molecule trajectory was classified into suppressed (confined and hop), simple-Brownian, or directed diffusion when its RD value was smaller than $RD_{min}$, between $RD_{min}$ and $RD_{MAX}$, and greater than $RD_{MAX}$ (no trajectory fell in this category), respectively. The distributions of RDs for the Cy3-DOPE and TfR trajectories are shown by shaded histograms (n = 50 and 20 for the intact and actin-depleted blebbed PM, respectively). **(C a and b)** In the blebbed apical PM, Cy3-DOPE (a) and Cy3-Tf bound to TfR (b) exhibited single diffusion coefficients that are ≈20× greater than $D_{MACRO}$ in the intact apical PM. The distributions of $D_{micro}$ (=$D_{2-4}$) are underestimated

due to the insufficient time resolution for measuring $D_{micro}$ within a compartment, even at a 0.1-ms resolution. Arrowheads indicate the median values (summarized in Table 2). These diffusion coefficients in the blebbed PM are slightly smaller than those obtained with gold probes (Fujiwara et al., 2002; Fujiwara et al., 2016). This is probably due to the residual actin filaments in the T24 cells employed here, since the membrane-bound actin filament meshes are much denser in T24 cells than the NRK cells used previously. **(D)** Cy3-DOPE and TfR exhibited similar compartment size distributions, suggesting that the underlying mechanism for the compartmentalization would be the same for the phospholipid and the transmembrane protein. Arrowheads indicate the median values. The statistical test methods, parameters (number of experiments), and P values are summarized in Table 2.

compartmentalized PM suggests that $D_{micro}$ is about the same as the diffusion coefficient of unhindered diffusion in the actin-depleted PM, but in all the molecules and cells examined thus far, the $D_{micro}$s in the intact PM were smaller than those for the unhindered diffusion coefficients in the actin-depleted PM (Table 2; Fujiwara et al., 2002; Fujiwara et al., 2016; Murase et al., 2004; Suzuki et al., 2005). This is probably because the time resolutions of 25—100 µs achieved thus far (10—40 kHz) are not sufficient to observe trajectories near the compartment boundaries, thus effectively reducing the step sizes in the trajectory, which will reduce the diffusion coefficient within the compartment, $D_{micro}$ (Ritchie et al., 2005).

**Direct determination of the dwell lifetime of each membrane molecule within each compartment using ultrafast SFMI**
We directly evaluated the dwell lifetime of each molecule in each compartment using ultrafast SFMI and developing an improved method for detecting the moment (instance) at which the observed molecule undergoes the hop movement across the compartment boundary. The moments of hops of the observed molecule can be found in its single-molecule trajectory by the method of detecting a Transient Increase of the effective Local Diffusion (TILD) in the trajectory (Fig. S5). TILDs are likely to occur when a molecule hops between two membrane compartments, but the analysis itself remains model independent. TILDs were detected in all of the experimental trajectories obtained at 0.1- and 0.167-ms resolutions (for Cy3-DOPE and TfR, respectively) in the intact PM (see Fig. 3, C and D); however, in the blebbed PM, where all of the trajectories were statistically classified into the simple-Brownian diffusion mode, TILDs were detected in only 4% (9%) of the trajectories (one TILD per 750-step trajectory) for Cy3-DOPE [TfR]. These results support the occurrences of hop diffusion in the intact PM and simple-Brownian diffusion in the blebbed PM.

The duration between two consecutive TILD events represents the dwell duration within a compartment. The distributions of the dwell durations are shown in Fig. 5 A. They could be fitted with single exponential decay functions, with lifetimes of 9.2 ± 0.34 ms for Cy3-DOPE and 23 ± 1.5 ms for TfR. The exponential shape of this distribution is consistent with the prediction by hop-diffusion theory developed here (see "Supplemental theory 1. Expected distribution of the dwell lifetimes: Development of the hop diffusion theory" in the Supplemental text). The decay time constants will be called the "dwell lifetimes" in the remaining part of this paper. The dwell lifetime for TfR is longer than that for Cy3-DOPE by a factor of 2.5, probably because the fence and picket effects both act on TfR, whereas only the picket effect works on Cy3-DOPE.

The average dwell lifetime within a compartment could also be evaluated from the median values of $L$ and $D_{MACRO}$ determined by the hop-diffusion fitting of the MSD-$\Delta t$ plot, using the equation $\tau = L^2/4D_{MACRO}$ (Fig. 4, C and D; and Table 2; Supplemental theory 1 in the Supplemental text). This provided dwell lifetimes of 8.5 ms for Cy3-DOPE and 24 ms for TfR, which are in good agreement with the values obtained based on the dwell lifetime distributions determined by the TILD method (Table 2).

In our previous studies using gold probes, we obtained $L$ from ultrafast single-gold-particle tracking, but needed to separately obtain $D_{MACRO}$ using fluorescently tagged molecules (by observing them at video rate) to obtain the average dwell lifetime within a compartment (as $L^2/4 D_{MACRO}$; Fujiwara et al., 2002, 2016; Murase et al., 2004). This is because gold probes induced crosslinking of the target molecules, elongating the dwell lifetime, and thus provided reduced $D_{MACRO}$ values. Using the ultrafast SFMI, we can directly determine the dwell lifetime of each molecule in each compartment (not just the averaged lifetime).

**An anomaly analysis provided further proof of the hop diffusion of Cy3-DOPE and TfR in the compartmentalized PM**
Generally, the MSD can be expressed as a function of the time interval $\Delta t$,

$$MSD = C\Delta t^\alpha (C, \alpha : constant),$$

where $0 \leq \alpha \leq 2$ ($\alpha = 1$ for simple-Brownian diffusion, $0 \leq \alpha < 1$ for suppressed anomalous diffusion, $1 < \alpha < 2$ for super diffusion, and $\alpha = 2$ for ballistic motion; Saxton, 1994, 1996; Feder et al., 1996; Simson et al., 1998; Fujiwara et al., 2002). This equation can be rewritten as

$$log(MSD/\Delta t) \propto (\alpha - 1)log(\Delta t).$$

This relationship is plotted in Fig. 5 B. A very broad time range of ~5 orders of magnitude (from 0.044 ms to 2 s) was covered, which was made possible by the development of the ultrafast SFMI.

In this display, simple-Brownian type diffusion is represented by a flat line (since $\alpha = 1$ in simple-Brownian cases, the slope $\alpha-1\approx0$; i.e., no time dependence) and suppressed diffusion is characterized by negative slopes ([$\alpha-1$] <0; i.e., $\alpha$ <1). Namely, the level of the diffusion anomaly can be parameterized by the parameter $\alpha$ (Feder et al., 1996; Simson et al., 1998). In the actin-depleted blebbed PM, both Cy3-DOPE and TfR exhibited flat lines in the entire time range of 0.3—30 ms, unequivocally demonstrating that they both undergo simple-Brownian diffusion in the blebbed PM.

On the contrary, in the intact apical PM, the plots for Cy3-DOPE and TfR (Cy3-Tf and 5xCy3-Tf) exhibited suppressed

**Table 2. Examination of the ultrafast SFMI developed here to determine whether it can correctly evaluate hop-diffusion parameters for Cy3-DOPE and TfR in the apical PM, previously found by ultrafast gold-particle imaging (dwell lifetimes were previously determined using additional data from video-rate observations of fluorescently labeled molecules)**

| | Mode of Motion (%) | | Cmpt. Size (L, nm)[b] | $D_{MACRO}$ (µm²/s)[c] | $D_{micro}$ (µm²/s) | Dwell Lifetime (τ, ms)[d] | n (mole-cules) |
|---|---|---|---|---|---|---|---|
| | Supp[a] | Simp[a] | Median Mean ± SEM Fig. No. | Median Mean ± SEM Fig. No. | Median Mean ± SEM Fig. No. | Mean ± SEM Fig. No. | |
| Cy3-DOPE | 76 | 24 | 101 107 ± 6.2[#1] Fig. 5 D | 0.30 0.30 ± 0.021[#2] Fig. 5, C a | 0.78 0.83 ± 0.056 Fig. 5, C a | 9.2 ± 0.34[#3] Fig. 6 A | 50 |
| gold-DOPE[e] | 100 | 0 | 110 120 ± 9.7 | 0.34 0.35 ± 0.019 (30 Hz Cy3-DOPE, n = 60) | NR | 8.9 | 35 |
| gold-DOPE[e] (NRK cells) | 85 | 15 | 230 240 ± 11 | 1.1 1.2 ± 0.071 (40 kHz gold-DOPE, n = 90) | 5.4 (median) | 13 | 90 |
| Cy3-TfR | 80 | 20 | 103 107 ± 5.4[1] Fig. 5 D | 0.11 0.13 ± 0.014[2] Fig. 5, C b | 0.51 0.57 ± 0.045 Fig. 5, C b | 23 ± 1.5[3] Fig. 6 A | 50 |
| gold-TfR[e] | 97 | 3 | 100 120 ± 10 | 0.17 0.19 ± 0.0081 (30 Hz Cy3-TfR, n = 174) | NR | 15 | 38 |
| gold-TfR[e] (NRK cells) | 91 | 9 | 260 270 ± 10 | 0.29 0.29 ± 0.011 (30 Hz Cy3-TfR, n = 61) | 5.2 (median) | 58 | 107 |
| Cy3-DOPE (Actin-depleted Blebbed PM) | 10 | 90 | NA | 5.7[f] 6.0 ± 0.38[f] Fig. 5, C a | NA | NA | 20 |
| gold-DOPE[e] (Actin-depleted Blebbed PM of NRK cells) | 13 | 87 | NA | 8.5[f] 8.9 ± 0.47[f] | NA | NA | 30 |
| Cy3-TfR (Actin-depleted Blebbed PM) | 5 | 95 | NA | 2.5[f] 3.3 ± 0.52[f] Fig. 5, C b | NA | NA | 20 |
| gold-TfR[e] (Actin-depleted Blebbed PM of NRK cells) | 26 | 74 | NA | 8.1[f] 8.0 ± 0.71[f] | NA | NA | 19 |

[a]Supp = Suppressed diffusion. Simp = Simple-Brownian diffusion.

[b]Cmpt. Size = Compartment size (L), determined by the hop-diffusion fitting, using frame rates at 10 and 40 kHz for fluorescently tagged and gold-tagged molecules, respectively. Fig. No. = The figure in which the histogram (distribution) is shown.

[c]$D_{MACRO}$s for Cy3-DOPE and Cy3-TfR were determined by the hop-diffusion fitting. $D_{MACRO}$s for gold-DOPE and gold-TfR were generally underestimated, due to the crosslinking effect of the gold probes. Therefore, previously they were mostly determined by the observations of Cy3-DOPE and Cy3-TfR at 30 Hz (Fujiwara et al., 2016). $D_{MACRO}$ of gold-DOPE in NRK cells was determined for the diffusion in the time window of 30 ms (using the data obtained at 40 kHz). In the apical PM of the NRK cell, the PM exhibited nested double compartments, and the $D_{MACRO}$ of DOPE over smaller compartments could not be determined using fluorescent probes (Fujiwara et al., 2002; Fujiwara et al., 2016).

[d]Dwell lifetime within a compartment (τ) for Cy3-labeled molecules was determined by the exponential fitting of the distributions of the residency duration for each molecule in each compartment, obtained by the TILD analysis (Fig. S5). SEM was determined from the fitting error of the 68.3% confidence interval. τ for gold-tagged molecules was calculated from the median values of L and $D_{MACRO}$, using the equation $τ = L^2/4D_{MACRO}$.

[e]Gold-DOPE and gold-TfR data are from Fujiwara et al. (2002), Fujiwara et al. (2016).

[f]Since DOPE and TfR labeled with Cy3 or gold undergo simple-Brownian diffusion in the actin-depleted blebbed PM, the MSD-Δt plot was fitted by a linear function, and the diffusion coefficient was obtained by the slope/4.

#[Y], and [N]. Results of statistical tests. The distributions selected as the basis for the comparison are shown by the superscript, [e]. Different numbers (1–3) indicate different bases for the Brunner–Munzel test (1, 2) and the log-rank test (3). The superscript Y (or N) with numbers indicates that the distribution is (or is not) significantly different from the base distribution indicated by the number after [e], with P values smaller (or greater) than 0.05. N1: P = 0.85, Y2: P = 4.3 × $10^{-14}$, Y3: P = 1.2 × $10^{-29}$.

NR: Not Reported. NA: Not Applicable because most molecules exhibited simple-Brownian diffusion.

diffusion (negative slopes) in the time ranges of 0.5—30 and 0.13—70 ms, respectively, indicating that the suppressed diffusion of Cy3-DOPE and TfR was detectable in these time ranges. Meanwhile, in longer and shorter time ranges, the plots asymptotically approached flattened plots, consistent with simple-Brownian diffusion in these time regimes.

These results can readily be explained by the hop-diffusion model in the compartmentalized PM for both Cy3-DOPE and TfR. In the shorter time regime (<0.5 and 0.13 ms, respectively),

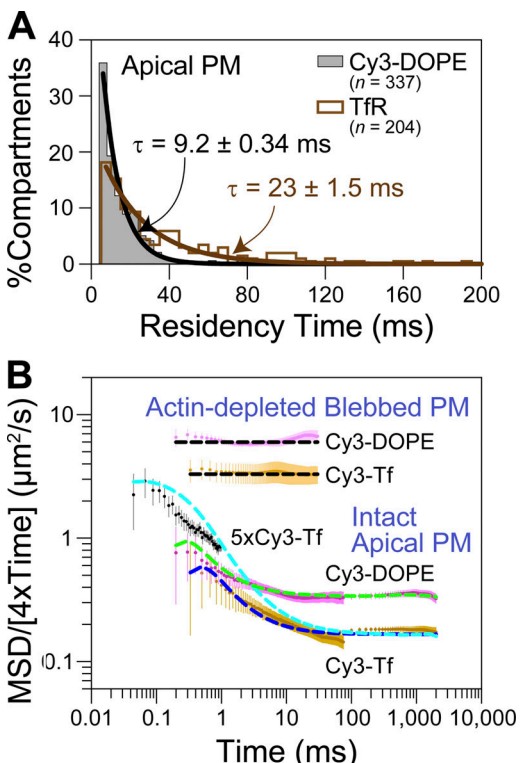

Figure 5. **Ultrafast SFMI further supports the concept of hop diffusion of membrane molecules in the apical PM, by enabling two new analyses: the distribution of the residency time of each molecule within each compartment and the diffusion anomalies based on log(MSD/time) vs. log(time) in the time range over five orders of magnitude. (A)** The distributions of the residency times within a compartment for Cy3-DOPE and TfR, determined by the TILD analysis, with the best-fit exponential curves (see "Expected distribution of the residency times: development of the hop diffusion theory" in the caption to Fig. S5). **(B)** An anomaly analysis of single-molecule trajectories based on the plot of log(MSD/time) vs. log(time), supporting the hop-diffusion model in the compartmentalized PM for both Cy3-DOPE and TfR (see the main text). Using the data obtained at time resolutions of 0.022 ms (45 kHz; only for TfR labeled with 5xCy3-Tf), 0.1 ms (10 kHz; for Cy3-DOPE), 0.167 ms (6 kHz; for TfR labeled with Cy3-Tf), and 33 ms (30 Hz; for Cy3-DOPE and TfR labeled with Cy3-Tf), the mean values of log(MSD/time) averaged over all trajectories were obtained and plotted as a function of log(time). The results of TfR using Cy3-Tf and 5xCy3-Tf were different in the time ranges shorter than 1 ms, due to the differences in the observation frame rates (6 and 45 kHz, respectively). During 0.167 ms, which is the frame time in 6-kHz observations, TfR still collides with the compartment boundaries, but this occurs much less often when the frame time is 0.022 ms (frame time in 45-kHz observations). Therefore, in shorter time ranges, the results obtained at 45 kHz (using 5xCy3-Tf) are better (see the simulation results shown by the dashed cyan curve). For the same reason, the 10-kHz data using Cy3-DOPE show that due to the insufficient time resolution, the pure simple-Brownian diffusion within a compartment could not be measured even at this frame rate. Dashed curves represent the results of the Monte Carlo simulations, resembling the experimental data (see Materials and methods for the simulation parameters). Note that the phospholipid probes are located in the PM outer leaflet, and yet they undergo hop diffusion. This is probably because, as proposed previously (Fujiwara et al., 2002), the transmembrane proteins anchored to and aligned along the actin mesh (pickets; see Fig. 3 A) form the diffusion barrier in both the outer and inner leaflets of the PM. The picket effect is due not only to the steric hindrance of the picket proteins, but also to the hydrodynamic-friction-like effect from the surface of the immobilized picket proteins on the surrounding medium (Fujiwara et al., 2002). Monte Carlo simulations showed that 20—30% occupancy of the compartment boundary by the immobile picket proteins

Cy3-DOPE and TfR collide with the compartment boundaries less frequently, approaching simple-Brownian diffusion and thus providing the diffusion coefficient of $D_{micro}$ within a compartment (although this is still underestimated due to the collisions with the compartment boundaries. However, in this time regime, translocation across the compartment boundaries hardly occurs). In the longer time regime (>30 and 70 ms, respectively), the diffusion of Cy3-DOPE and TfR represents that among compartments, because the detailed shorter term behaviors are averaged out, and their movements thus resemble simple-Brownian diffusion again, with a diffusion coefficient of $D_{MACRO}$. In the time ranges of 0.5—30 and 0.13—70 ms for Cy3-DOPE and TfR (Cy3-Tf and 5xCy3-Tf), respectively, since the effect of confinement within a compartment becomes increasingly evident with a lengthening of the observed time window until occasional hops across the compartment boundaries start alleviating this confinement, the MSD/(4t), which is the y-axis in this plot, is expected to decrease from $D_{micro}$ until it reaches $D_{MACRO}$.

The dashed curves in Fig. 5 B represent the results of the Monte Carlo simulations that resemble the experimental data. The cyan curve would be the best for representing the diffusion of TfR in the apical PM. In the longer time regime (>70 ms), it approached the experimental result for Cy3-Tf at 30 Hz, while in the shorter time regime (<0.8 ms), it approached the experimental data for 5xCy3-Tf at 45 kHz, and at its shorter limit, it approached the flat line of the Cy3-Tf observed in the blebbed PM.

## TfR and Cy3-DOPE undergo hop diffusion in the basal PM, exhibiting virtually the same compartment sizes and dwell lifetimes as those in the apical PM

Single molecules of TfR (TfR's N-terminus is located in the cytoplasm and it was fused to the Halo-tag protein and labeled with TMR) were observed in the basal PM outside the focal adhesion region marked with mGFP-paxillin (called bulk basal PM), at a frame rate of 6 kHz (0.167-ms resolution; 200 times faster than normal video rate) and compared with the results obtained in the apical PM (Fig. 6 A and Video 5).

Using the MSD-Δt plot for each TfR trajectory, we found that the majorities of the trajectories were categorized into the suppressed diffusion mode in the basal PM, like those in the apical PM (72 and 80%, respectively; Fig. 6 B and Table 3). Consistently, the MSD-Δt plots of 81 and 82% of the trajectories obtained in the basal and apical PM, respectively, could be fitted by the hop-diffusion fitting equation, indicating that most TfR molecules undergo hop diffusion in the basal PM as well as in the apical PM. Interestingly, the median compartment size in the basal PM (109 nm) was quite comparable to that in the apical PM (103 nm; Fig. 6 C; non-significant difference; all statistical parameters and test results for diffusion parameters are summarized in Table 3).

The distributions of the TfR residency durations within a compartment in the basal PM, obtained by the TILD analysis,

(bound to the actin fence) is sufficient to cause confined + hop diffusion of the phospholipids in the PM outer leaflet (Fujiwara et al., 2002).

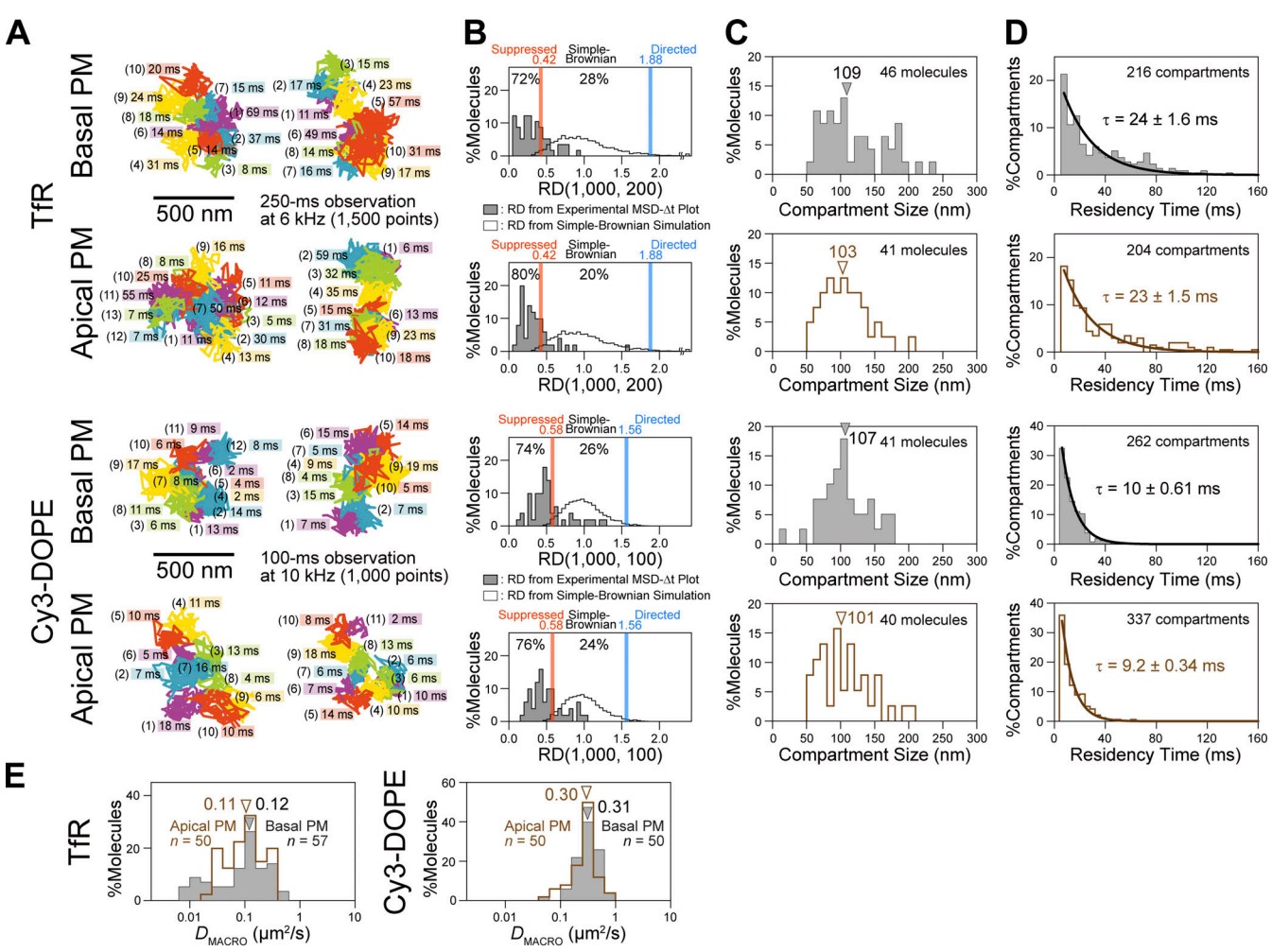

Figure 6. **Ultrafast SFMI of TfR and Cy3-DOPE revealed that the basal PM outside the FAs (bulk basal PM) is compartmentalized like the apical PM and that the dwell lifetimes of TfR and Cy3-DOPE within a compartment in the bulk basal PM are the same as those in the apical PM.** The data about the apical PM are reproduced in figures B–E here from Fig. 4, B and D; and Fig. 5 A for the ready comparison. **(A)** Typical ultrafast single fluorescent-molecule trajectories of TfR (6 kHz) and Cy3-DOPE (10 kHz), diffusing in the bulk basal PM and in the apical PM. The order of the compartments visited by the molecules (parenthesized integers) and their respective dwell lifetimes there, as determined by the TILD analysis (Fig. S5), are shown in the figure. TMR was used to label the Halo-tag protein fused to the cytoplasmic domain of TfR (N-terminus). Since Cy3 (and Alexa555), which exhibited superior throughput among all the dyes tested (Figs. S2 and S3), is membrane impermeable, the availability of a membrane-permeable dye, TMR, for ultrafast observations is very useful. **(B)** Motional mode classification based on $RD$, performed as described in Fig. 4 B. Shaded and open bars show the $RD$ distributions, obtained by experiments and simulation for simple-Brownian particles, respectively. The percentages of molecules categorized into the suppressed and simple-Brownian diffusion modes are indicated. **(C)** Distributions of the compartment sizes determined from the hop-diffusion fitting of the MSD-$\Delta t$ plot for each TfR and Cy3-DOPE molecule. The MSD-$\Delta t$ plot data that could not be fit (due to large noise and the closeness of $D_{micro}$ and $D_{MACRO}$) were not included. Arrowheads indicate the median values. **(D)** Distributions of the TfR and Cy3-DOPE residency times within a compartment determined by the TILD analysis, with the best-fit single exponential functions. The decay time constants of these curves provide the dwell lifetimes (see the caption to Fig. 5 A). **(E)** Distribution of $D_{MACRO}$ determined by the hop-diffusion fitting of the MSD-$\Delta t$ plot for each TfR and Cy3-DOPE molecule (shaded histogram, bulk basal PM; open histogram, apical PM). Arrowheads indicate the median values. The statistical test methods, parameters (number of experiments), and P values are summarized in Table 3.

could be fitted by a single exponential decay function, as predicted by the hop-diffusion theory (Supplemental theory 1 in the Supplemental text), providing the dwell lifetimes of TfR in a compartment of 24 ± 1.6 ms in the basal PM as compared with 23 ± 1.5 ms in the apical PM (Fig. 6 D). Namely, between the apical PM and the basal PM of T24 cells, TfR dwell lifetimes within a compartment as well as the compartment sizes were very similar to each other, despite that the apical PM is facing the cell-culture medium whereas the basal PM is located very closely to the coverslip and responsible for the cell's binding to the coverslip.

This was quite surprising to us, and so, the average residency time ($\tau$) within a compartment was evaluated by another method, using the equation $\tau = L^2/4D_{MACRO}$. The median values of $L$ were 109 and 103 nm (Fig. 6 C) and those of $D_{MACRO}$ were 0.12 and 0.11 µm²/s (Fig. 6 E) in the basal and apical PMs, respectively. These provided the average residency times of 25 and 24 ms in the basal and apical PM, respectively, in agreement with the values obtained by the TILD method. Therefore, we conclude that the TfR's dwell lifetimes in a compartment were virtually the same in both the basal and apical PMs.

**Table 3.** Motional mode, compartment size (L), $D_{MACRO}$, and dwell lifetime (τ), characterizing the hop diffusions of Cy3-DOPE, TfR, and EGFR in apical and basal PMs of T24 cells

| | Motional Mode (%) | | Cmpt. Size (L, nm)[b] | $D_{MACRO}$ (µm²/s)[b] | Residency Lifetime (τ, ms)[b] | n[c] |
|---|---|---|---|---|---|---|
| | Supp[a] | Simp[a] | Median Mean ± SEM Histogram Fig. No.[d] | Median Mean ± SEM Histogram Fig. No. | Mean ± SEM Histogram Fig. No. | |
| Cy3-DOPE (Apical PM) | 76 | 24 | 101 <br> 107 ± 6.2[#1,#2] <br> Fig. 6 C | 0.30 <br> 0.30 ± 0.021[#3,#4] <br> Fig. 6 E | 9.2 ± 0.34[#5,#6] <br> Fig. 6 D | 50 |
| Cy3-DOPE (Basal PM) | 74 | 26 | 107 <br> 107 ± 5.6[#7,#8,N2] <br> Fig. 6 C | 0.31 <br> 0.34 ± 0.022[#9,N4] <br> Fig. 6 E | 10 ± 0.61[#10,N6] <br> Fig. 6 D | 50 |
| TfR (Apical PM) | 80 | 20 | 103 <br> 107 ± 5.4[#11,N1] <br> Fig. 6 C | 0.11 <br> 0.13 ± 0.014[#12,Y3] <br> Fig. 6 E | 23 ± 1.5[#13,Y5] <br> Fig. 6 D | 50 |
| TfR (Basal PM; Outside FA) | 72 | 28 | 109 <br> 121 ± 6.9[#14,N7,N11] <br> Fig. 6 C | 0.12 <br> 0.14 ± 0.016[Y9,N12] <br> Fig. 6 E | 24 ± 1.6[Y10,N13] <br> Fig. 6 D | 57 |
| EGFR (Basal PM; Before stimulation) | 73 | 27 | 106 <br> 119 ± 6.5[#15,N8,N14] <br> Fig. 7 B | 0.19 <br> 0.21 ± 0.024[#16] <br> Fig. 7 D | 18 ± 1.2[#17] <br> Fig. 7 C | 41 |
| EGFR (Basal PM; 2.5–5 min after stimulation) | 85 | 15 | 106 <br> 110 ± 6.0[N15] <br> Fig. 7 B | 0.11 <br> 0.12 ± 0.019[Y16] <br> Fig. 7 D | 27 ± 1.5[Y17] <br> Fig. 7 C | 39 |

[a]Supp = Suppressed diffusion. Simp = Simple-Brownian diffusion.

[b]Cmpt. Size = Compartment size (L). L and $D_{MACRO}$ in the intact PM were estimated by the hop-diffusion fitting. Residency Lifetime (τ) = exponential decay lifetime (mean ± SEM; SEM was determined from the fitting error of the 68.3% confidence interval) within a compartment, based on the residency time distributions obtained by the TILD analysis (Fig. S5).

[c]The number of inspected molecules.

[d]Histogram. Fig. No. = The figure in which the histogram (distribution) is shown.

#, Y, and N. The results of the statistical tests. The distributions selected as the basis for the comparisons are shown by the superscript, #. Different numbers (1–17) indicate different bases for the Brunner-Munzel test (1–4, 7–9, 11 12, 14–16, 15, and 16) and the log-rank test (5, 6, 10, 13, and 17). The superscript Y (or N) with numbers indicates that the distribution is (or is not) significantly different from the base distribution indicated by the number after #, with P values smaller (or greater) than 0.05 as shown below.

N1: P = 0.85, N2: P = 0.70, Y3: P = 4.3 × 10⁻¹⁴, N4: P = 0.20, Y5: P = 1.2 × 10⁻²⁹, N6: P = 0.27, N7: P = 0.40, N8: P = 0.31, Y9: P = 2.2 × 10⁻¹⁶, Y10: P = 6.0 × 10⁻³¹, N11: P = 0.26, N12: P = 0.81, N13: P = 0.17, N14: P = 0.81, N15: P = 0.43, Y16: P = 8.7 × 10⁻⁴, Y17: P = 0.044.

The phospholipid, Cy3-labeled L-α-dioleoylphosphatidylethanolamine (Cy3-DOPE), also undergoes hop diffusion, exhibiting practically the same median compartment sizes as that for TfR in the basal PM and those for Cy3-DOPE and TfR in the apical PM (Fig. 2 C). The dwell lifetime of Cy3-DOPE within a compartment was shorter than that of TfR by a factor of ≈2.5 in both the basal and apical PMs (Fig. 6 D). Consistently, the $D_{MACRO}$ values of Cy3-DOPE were the same in the basal and apical PMs (Fig. 6 E). Overall, the bulk basal PM is compartmentalized in virtually the same manner as the apical PM, in T24 cells, with regard to the compartment sizes and temporary confinements of TfR and Cy3-DOPE.

Equality of the compartment size for TfR and Cy3-DOPE in the same membrane suggests that the underlying mechanisms for the compartmentalization are the same for these two molecules. This result is consistent with the model that the PM is compartmentalized by the actin-based fences and transmembrane protein pickets anchored to and aligned along the fences (Fig. 3 A).

Since the compartment size distributions of the apical and basal PMs are quite similar, the following possibility was

examined: the hop diffusion detected by this camera might simply be an apparent phenomenon caused by the photo response non-uniformity (PRNU) of the developed camera system. PRNU might affect the single-molecule localization precisions through pixel-to-pixel variations in the sensitivity, which might make the molecules appear like those undergoing hop diffusion. The test result, shown in Fig. S1 of the companion paper (Fujiwara et al., 2023), indicated that PRNU of the developed camera is quite similar to that of EM-CCD camera and would scarcely affect the single-molecule localization precisions. Furthermore, as described in the companion paper, we found different compartment sizes within the FA region in the basal PM. Therefore, we conclude that PRNU would not induce apparent hop-like movements for membrane molecules in the PM.

## Prolonged confinement of activated EGF receptor (EGFR) within a compartment

EGFR exists as both monomers and dimers before EGF stimulation, and they interconvert rapidly with a dimer lifetime of

≈13 s (Chung et al., 2010). After stimulation, EGFR tends to form rather stable dimers and greater oligomers for self-phosphorylation and activation (Chung et al., 2010; Low-Nam et al., 2011; Huang et al., 2016). We examined whether the EGF ligation of the receptor affects its lateral diffusion, because this will determine the rate of activated EGFR spreading along the PM by diffusion, which will provide a basis for a variety of mechanisms for EGFR signal propagation (Verveer et al., 2000; Reynolds et al., 2003; Koseska and Bastiaens, 2020).

EGFR (fused to the Halo-tag protein at the cytoplasmic C-terminus of EGFR, labeled with TMR) was observed in the basal PM at the level of single molecules at a frame rate of 6 kHz (0.166 ms resolution; Video 6). Due to the presence of non-labeled endogenous EGFR, even the fluorescent spots with the intensity of a single EGFR might represent EGFR dimers and greater oligomers. Virtually all of the fluorescent EGFR spots underwent hop diffusion both before and after ligation (Fig. 7, A and B; and Table 3). Interestingly, their median compartment sizes were the same before and after stimulation (106 nm), and quite comparable to the compartment sizes in the basal PM found by observing TfR (109 nm) and Cy3-DOPE (107 nm; nonsignificant differences; Table 3), supporting the PM compartmentalization for all PM-associated molecules.

The dwell lifetime of EGFR within a compartment was increased from 18 ± 1.2 ms before stimulation to 27 ± 1.5 ms during 2.5—5 min after stimulation (Fig. 7 C), which reduced $D_{MACRO}$ by ≈42% (Fig. 7 D). Since EGFR is likely to form stable dimers upon stimulation, the slowed macroscopic diffusion or prolonged dwell lifetime within a compartment could be induced by the lower hop probabilities of the larger diffusant; i.e., engaged EGFR dimers. TfR is also a single-pass transmembrane protein and exists as constitutive dimers and their dwell lifetime in the basal PM was 25 ms, quite comparable to the dwell lifetime of EGFR after stimulation, suggesting that engaged EGFR primarily exists as dimers. This result is consistent with the "oligomerization-induced trapping" model proposed previously (Iino et al., 2001; Murakoshi et al., 2004; Heinemann et al., 2013). The longer confinement of engaged receptors within or near the compartment where the ligand EGF was originally received will be beneficial for localizing the signal, which might in turn induce the local reorganization of the actin cytoskeleton required for membrane ruffling and chemotaxis.

## Discussion

The ultrahigh-speed camera system developed in this study has enabled the fastest SFMI ever performed. This would represent the ultimate rate possible with the currently available fluorescent molecules, e.g., 0.1 ms resolution with a 20-nm localization precision for single Cy3 molecules, using a saturating TIR illumination laser intensity of 79 µW/µm², for a frame size of 256 × 256 pixels (14 × 14 µm²). Under these conditions, ~1.6% of the single Cy3 molecules could be tracked for periods longer than 100 frames (10 ms). Approximately 14% of single Cy3 molecules could be tracked for 100 frames under our standard conditions (oblique-angle illumination at a laser intensity of 23 µW/µm² with a 39-nm localization precision). When better fluorophores

(faster emission and slower photobleaching) are developed, the time resolution for single-molecule imaging could be enhanced to 22 µs or more (for a frame size of 7.5 × 3 µm² in 128 × 64 pixels), without affecting other parameters (Fig. S3 and Table 1). Presently, the fastest practical time resolution for SFMI is 33 µs with single-molecule localization precision of 34 nm for a frame size of 7.1 × 6.2 µm² (128 × 112 pixels).

For tracking just one molecule at a time, MINFLUX has recently been developed, and its performance on the glass is superb (Balzarotti et al., 2017; Eilers et al., 2018; Schmidt et al., 2021). It has achieved single-molecule localization precisions of 2.4 nm and <20 nm with 0.4- and 0.12-ms temporal resolutions, respectively. However, it can track only one molecule at a time (but could be expanded to track a few isolated molecules at a time). In contrast, the camera-based single-molecule imaging allows the observations of several tens to hundreds of molecules simultaneously in a field of view (like an entire cell). The camera-based method can thus be used to investigate the interactions and assemblies of molecules in live cells and study the different behaviors of molecules simultaneously in various regions of the cell, which is impossible with one molecule/cell technologies.

In live E. coli, MINFLUX provided a single-molecule localization precision of 48 nm with a 125-µs time resolution, under the conditions of observing the same molecule 742 times (Balzarotti et al., 2017). This is quite comparable to the results we obtained using the camera-based method, which allows us to simultaneously observe many single molecules at single-molecule localization precisions of 20 and 34 nm (27 and 55 nm) at time resolutions of 100 and 33 µs using TIR (oblique-angle) illumination (Table 1). However, practically speaking, the same single molecules could be observed for 100—300 frames (10—30 ms at 10 kHz; localizing them 100—300 times), as compared with localizing one molecule 742 times with MINFLUX. Therefore, the ultrafast camera-based single-molecule imaging and MINFLUX provide complementary methods for examining the very fast dynamics of membrane molecules in living cells: MINFLUX is useful for tracking a single molecule for longer periods, whereas the ultrafast camera-based single-molecule imaging provides a method for observing many molecules at once to investigate their interactions and the location-dependent behaviors of single molecules in a cell. Single molecules labeled with two or three different colors could also be investigated readily using two or three cameras, respectively.

The ultrafast camera system developed here was tested by examining whether it could detect the hop diffusion of Cy3-tagged DOPE and TfR in the apical PM of live cells, as expected from the previous observations using gold-tagged molecules (Fujiwara et al., 2002; Fujiwara et al., 2016; Murase et al., 2004). We found that it can indeed detect the hop diffusion of Cy3-tagged molecules (Fig. 3, 4, and 5; and Table 2) and the same compartment size as found by ultrafast gold-tracking (≈100 nm in the PM of T24 cells; Fig. 4 D and Table 2).

Ultrafast SFMI has brought forth new knowledge about the hop diffusion of membrane molecules. First, a new method to measure the dwell lifetime for each molecule in each compartment (not the overall average lifetime) was established. This

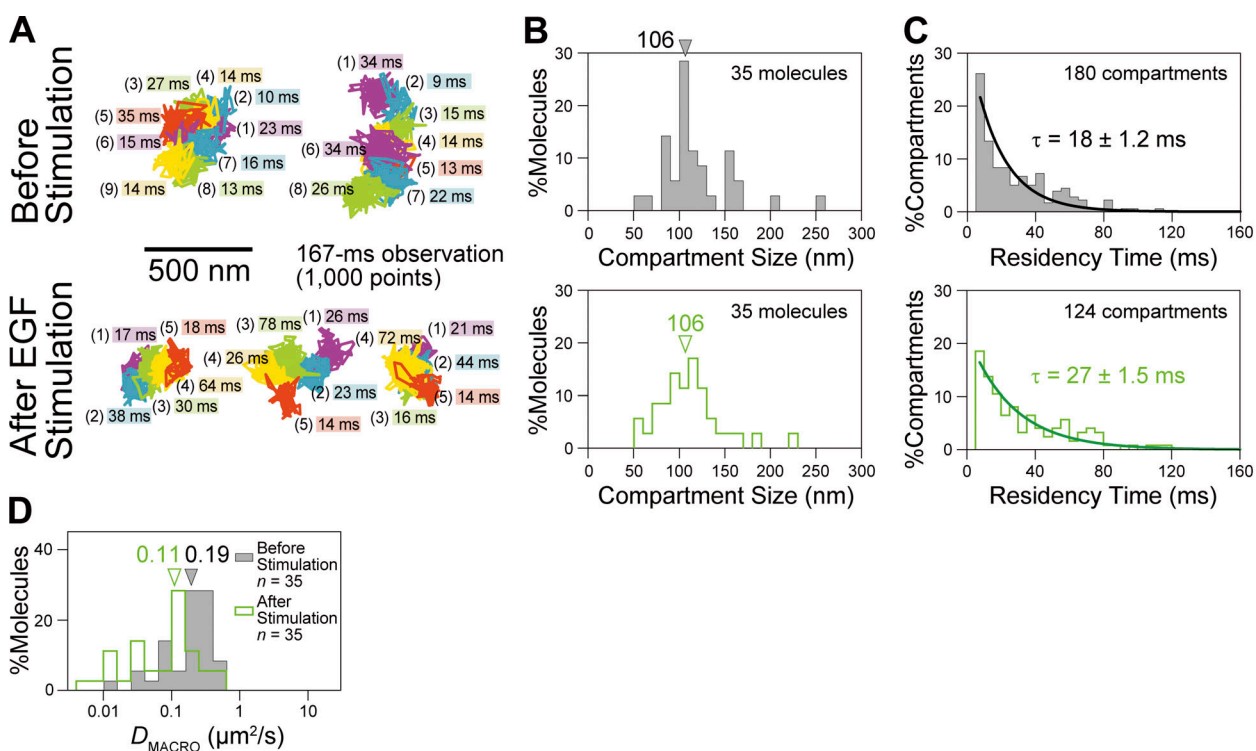

**Figure 7. EGFR and ligand-engaged EGFR in the basal PM detected virtually the same compartment sizes as those found with TfR and Cy3-DOPE, supporting the PM compartmentalization, and the dwell lifetime of the engaged EGFR was longer than that of non-engaged EGFR (same compartment sizes).** Following the stimulation with 10 nM EGF, microscope observations were performed between 2.5 and 5 min after the EGF addition. **(A)** Typical ultrafast (6 kHz) single fluorescent-molecule trajectories of EGFR, before and after EGF stimulation ($n$ = 41 and 39 trajectories, respectively). **(B)** Distributions of the compartment sizes detected by the single-molecule diffusion of EGFR in the basal PM, before (top) and after (bottom) EGF stimulation. Arrowheads indicate the median values of 106 nm for both before and after stimulation. **(C)** Distributions of the EGFR residency times within a compartment determined by the TILD analysis, with the best-fit exponential curves, providing the dwell lifetimes ($\tau$). Before (top) and after (bottom) EGF stimulation. Statistically significant difference between before and after stimulation with P = 0.044, using the log-rank test. **(D)** Distributions of $D_{MACRO}$ for EGFR in the basal PM determined by the hop-diffusion fitting of the MSD-$\Delta t$ plot for each EGFR molecule (shaded histogram, before stimulation; green open histogram, after stimulation). Arrowheads indicate the median values. $D_{MACRO}$ was reduced by a factor of 1.7 after stimulation. The statistical test methods, parameters (number of experiments), and P values are summarized in Table 3.

now enables us to obtain the distribution of the dwell lifetimes for each visit of each molecule to each compartment. Furthermore, we developed a theory to describe the dwell lifetime distribution. The developed theory together with the experiments made possible with the ultrafast camera system has firmly established the method to describe the hop-diffusion characteristics of membrane molecules in the PM (Fig. 5 A), opening up the possibility that PM compartmentalization can be studied readily in various cell types.

Second, ultrafast SFMI allowed an anomaly analysis of a very broad time range of ≈5 orders of magnitude (from 0.044 ms to 2 s) for determining the diffusion properties of Cy3-tagged molecules. The anomaly analysis confirmed the hop diffusion of Cy3-DOPE and TfR in the PM and their simple-Brownian diffusion in the actin-depleted blebbed PM (Fig. 5 B).

Third, ultrafast SFMI revealed that not only the apical PM, which could be studied by single gold particle tracking before, but also the basal PM is compartmentalized, and that the hop diffusion and PM compartment properties in the basal PM are virtually the same as those in the basal PM (Fig. 6). In the apical and basal PMs, the compartment sizes are 102 and 108 nm, respectively (arithmetic averages for the values obtained for TfR

and Cy3-DOPE); the dwell lifetimes of TfR in a compartment are 23 ± 1.5 and 24 ± 1.6 ms; and those of Cy3-DOPE are 9.2 ± 0.34 and 10 ± 0.61 ms. This answered the long-term question about whether the bulk basal PM is significantly different from the bulk apical PM. Even in non-polarized cells, the architecture of the PM facing the substrate (coverslip) within distances <40 nm (because gold particles of 40 nm in diameter do not enter the space between the basal PM and the coverslip) might be quite different from the PM facing just the cell culture medium. However, the results obtained in this work unequivocally demonstrated that the compartment organization and molecular hop diffusion in the basal PM are very similar to those in the apical PM.

Fourth, ultrafast SFMI showed that EGFR before and after stimulation undergoes hop diffusion, supporting the compartmentalization of the basal PM (Fig. 7). The engaged EGFR is confined in a compartment ≈2.5× longer than the non-engaged EGFR, which might be useful for inducing a location-dependent downstream signal.

Meanwhile, the simple-Brownian diffusion of both Cy3-DOPE and TfR in the actin-depleted blebbed PM indicates that actin filaments (and their associated transmembrane picket

proteins) are responsible for compartmentalizing the PM (Fig. 3 A). This conclusion is consistent with previous observations, including the changes of the compartment sizes after the treatments with actin-modulating chemicals, latrunculin, cytochalasin D, and jasplakinolide (Fujiwara et al., 2002; Fujiwara et al., 2016; Murase et al., 2004), the equality of the compartment sizes determined by the lipid diffusion data and the actin mesh sizes on the apical PM cytoplasmic surface determined by electron tomography (Morone et al., 2006) and super-resolution microscopy (Xia et al., 2019; Garlick et al., 2022), and the results of single-molecule optical trapping (Sako and Kusumi, 1995; Sako et al., 1998).

We believe that the PM compartmentalization is fundamentally important for understanding the PM function. For instance, PM compartmentalization would permit the cells to locally enhance phosphorylation at particular places on the PM, by sequestering the kinases in the compartment so that the local concentrations of kinases there exceed those of phosphatases, which are generally more abundant in the cytoplasm and have higher enzymatic activities than kinases; Kalay et al., 2012; Kusumi et al., 2012a). Compartmentalization could also confine stimulation-induced stabilized raft domains (Kusumi et al., 2004; Kusumi et al., 2011; Kusumi et al., 2012a; Kusumi et al., 2012b; Suzuki et al., 2012; Kusumi et al., 2020) and engaged receptor oligomers (oligomerization-induced trapping; Iino et al., 2001). Nevertheless, only a few researchers have directly investigated and identified hop diffusion in the PM due to the difficulties in using large gold probes and visualizing them at high frame rates. The developed ultrafast camera system with single fluorescent-molecule sensitivities allows the researchers to investigate PM compartmentalization and hop/confined diffusion of membrane molecules quite readily, enabling faster progress of PM structure, dynamics, and function. Furthermore, simultaneous two-color ultrafast SFMI imaging is also possible (Koyama-Honda et al., 2020).

We believe that the developed ultrafast camera has the immediate value and broad utility to the cell biology community by the following reasons. The highest time resolutions in SFMI of many single molecules at once are now available to the cell biology community. Ultrafast single-molecule imaging method is now enhanced by the development of the theoretical framework for the analysis of single-molecule trajectories in the PM, which provides the simple means to examine the effects of PM compartmentalization on biological processes in/on the PM (previously, very special single gold-particle imaging and tracking were necessary) and to elucidate the principles governing the PM organization. The ultrafast camera will facilitate monitoring very fast molecular interactions, which have probably been missed in cell biology research.

Furthermore, as shown in the companion paper, the newly developed ultrafast camera system also offers an unprecedented opportunity to simultaneously perform the fastest data acquisition (1 kHz) of PALM (using the mEos3.2 probe with a 29-nm localization precision) and dSTORM (using the HMSiR probe with a 19-nm localization precision) in a view-field as large as 640 × 640 pixels (35.3 × 35.3 μm² with the pixel size of 55.1 nm), together with ultrafast SFMI. The data acquisition of 333—10,000

frames requires merely 0.33—10 s, enabling live-cell PALM. Therefore, we conclude that the ultrafast camera system developed in the present work holds enormous potential as an invaluable tool for the field of cell biology.

## Materials and methods
### Ultrahigh-speed intensified CMOS camera system: Design and operation
*Detailed description of the camera system (Figs. 1 and 2; and Figs. S1, S2, and S3; and Table 1).*
See the schematic diagram of the developed camera system shown in Fig. 1 A. The new ultrahigh-speed camera system consists of the following three major components.

(a) A Hamamatsu image intensifier (V8070U-74), comprising a third-generation GaAsP photocathode with a quantum efficiency of 40% at 570 nm (Fig. 1, A a α), a three-stage microchannel plate allowing maximal gain over $10^6$, in which a non-linear noise increase was virtually undetectable (Fig. 1, A a β), and a P46 phosphor screen with a decay time of ~0.3 μs from 90 to 10% (Fig. 1, A a γ).

(b) A CMOS sensor (unless otherwise specified, we report the results obtained with the sensor developed for a Photron 1024PCI camera, but for some tests, we additionally report the results obtained with the newer sensor developed for a Photron SA1 camera), with a global shutter exposure was used (Table 1). The 1024PCI (SA1) sensor is composed of 1,024 × 1,024 pixels with a 17 × 17-μm (20 × 20-μm) unit pixel size and operates up to 1 kHz or every 1 ms (5.4 kHz or every 0.19 ms) for the full-frame readout. The 1024PCI sensor can be operated at 10 kHz (0.1-ms resolution) with a frame size of 256 × 256 pixels, at 45 kHz (0.022-ms resolution) with a frame size of 128 × 64 pixels, and at 110 kHz (0.009-ms resolution) with a frame size of 128 × 16 pixels. The SA1 sensor can also be operated at 16 kHz (0.063-ms resolution) with a frame size of 512 × 512 pixels, and at 45 kHz (0.022-ms resolution) with a frame size of 256 × 256 pixels (Table 1).

However, the actual observation frame rates for single-molecule tracking with reasonable single-molecule localization precisions (say ≤50 nm) were limited to 10–30 kHz. This is because the number of fluorescent photons that can be emitted from a single fluorophore during <0.033—0.1 ms (the integration time of a camera frame); i.e., the frequency that a single fluorophore can be excited, is limited, as quantitatively described in "Estimation of the number of photons that can be emitted by a single Cy3 molecule during 0.1 ms: Triplet bottleneck saturation" in the Materials and methods (also see Fig. 1, E and F; and Fig. S3). Namely, with the development of our ultrafast camera system, the instrument is no longer the limitation for achieving higher time resolutions (faster than 10—30 kHz) for SFMI, and now the availability of fluorescent dyes has become the limitation. With the future development of new dyes with faster excitation and emission, without making a transition to the triplet state, observations even at 110 kHz would be possible. Indeed, we showed that when an average (mean) of five

Cy3 molecules were attached to a single Tf protein (5xCy3-Tf), it can be observed at a rate of 45 kHz (Fig. 1 I; and Fig. S2 A and Fig. S3; and Table 1).

The readout noise, including the dark noise, of the 1024PCI and SA1 sensors at the full-speed readout at 21°C was 37 root-mean-square electrons/pixel, as measured by the manufacturer (Photron), and provided dynamic ranges of ≈800 and ≈1,200, respectively. The quantum efficiency of the CMOS sensor employed here was 40% (for both sensors), but since the CMOS sensor is placed after the image intensifier, the lower quantum efficiency of the CMOS sensor is not a critical issue in the developed camera system (the quantum efficiency of the image intensifier photocathode matters).

(c) A straight (1:1) optical-fiber bundle directly adhered to the phosphor screen of the image intensifier on one side (Fig. 1, A a γ, input side) and to the CMOS sensor on the other side (Fig. 1, A b, output side). This enhanced the signal reaching the CMOS sensor by a factor of 5—10, as compared with the lens coupling used for our standard camera system employed for SFMI (Suzuki et al., 2012; Kinoshita et al., 2017).

The electrons generated at the CMOS sensor were transferred, amplified (Fig. 1, A d), and digitized (Fig. 1, A e) in 64 and 8 parallel paths, respectively, and then transferred to the host PC.

### Basic design concept of the camera system

(a) The use of a CMOS sensor rather than an sCMOS sensor or an EM-CCD sensor.

To achieve ultrafast SFMI, we used a CMOS sensor rather than an sCMOS sensor or an EM-CCD sensor, which are broadly employed in fluorescence microscopy. This is because the maximum frame rates attainable with the sCMOS sensors and EM-CCD sensors (typically 1.2—4 ms/frame for a 256 × 256-pixel image) are at least 10 times slower than those of the CMOS sensors employed here (0.1 and 0.022 ms/frame for a 256 × 256-pixel image, using the Photron 1024PCI sensor and SA1 sensor, respectively; no binning). To avoid image distortions and achieve higher rates, we employed CMOS sensors with a global shutter (most sCMOS cameras employ rolling shutters; although some newer sCMOS cameras are equipped with global shutters, their frame rates are slower than those when rolling shutters are employed).

(b) The use of an image intensifier (also see the beginning of Results in the main text).

An obvious problem when using the CMOS camera is its high readout noise (30—40 vs. <5 root-mean-square electrons/pixel/frame for CMOS vs. sCMOS; the term "readout noise" used in this report always includes the dark noise of the sensor). As described in the previous section, the readout noise of the 1024PCI and SA1 sensors employed in this study was 37 root-mean-square electrons/pixel/frame (at the full-speed readout at 21°C). Meanwhile, the background noise signal ($b$ in the equation in the caption to Fig. S2) was 0.035 ± 0.058 detected photons/pixel/frame on the glass using the oblique illumination (mean ±

SD; in the case of TIR illumination, $b$ = 0.038 ± 0.059 detected photons/pixel/frame) and 0.068 ± 0.17 detected photons/pixel/frame on the cellular apical PM (oblique illumination). Therefore, the background noise signal is much smaller than the readout noise, as expected (i.e., $N_p$ << $Nr$ as described in the subsection "Basic conceptual strategy to address the large readout noise of the CMOS sensor" in the main text). As shown in Eq. 3 there, $G$ should be much greater than $Nr/N_p$, i.e., $G$ >> $Nr/N_p$. Therefore, $G$>>37/0.035 (or 0.068) or $G$>>1,100 (or 540) for single fluorescent molecules on the glass (apical PM). As described in the following paragraphs, by also considering $S_p$, we found a total amplification gain $G$ of 8,100 useful for SFMI.

Next, we address the signals from the fluorophores. Consider our standard test conditions for observing a single Cy3 molecule, using a frame rate of 10 kHz and oblique illumination for excitation at 23 µW/µm² (i.e., instrumentally less optimal as compared with the TIR illumination, but necessary for observing molecules in the apical PM as well as in the basal PM). Under these conditions, we obtained 34 ± 2.4 detected photons/molecule/frame (Fig. 1 E top; including a quantum efficiency of the image intensifier photocathode of 0.4 at 570 nm; i.e., the number of photons that arrived at the photocathode of the intensifier was 85 ± 6.0 photons/molecule/frame), which realized a single Cy3-molecule localization precision of 38 ± 1.7 nm on the coverslip (Fig. 1 F top). Namely, an average (total number of) of 34 ± 2.4 photons is detected on the CMOS sensor in the two-dimensional Gaussian image of a standard deviation (SD; radius) of 123 ± 1.1 nm (2.2 ± 0.020 pixels × 55.1 nm/pixel) on the sample plane. (This image size was determined by the Gaussian fitting of each image for 50 Cy3 molecules immobilized on the glass under the TIR illumination at 79 µW/µm²; this higher illumination was employed to determine the Gaussian image size more precisely than that obtainable under the oblique illumination at 23 µW/µm², by generating ∼3 times more detected photons; see Fig. 1 E top; image magnification by a factor of 300 [150 × 2]).

Consider how the 34 photons are distributed in the pixelized Gaussian image; i.e., the 2-dimensional intensity profile (point-spread function) of a single-molecule emitter pixelized by the CMOS sensor. The expected number of photons in the pixel located at the integer position $(m,n)$, $I(m,n)$, is given as (Huang et al., 2011):

$$I(m, n) = I_{total}E_mE_n. \qquad (5)$$

Here $I_{total}$ is the total number of detected photons in the image; i.e., 34 photons, and $E_m$ and $E_n$ are

$$E_m = \frac{1}{2}\left\{ erf\left(\frac{m - m_0 + \frac{1}{2}}{\sqrt{2}s}\right) - erf\left(\frac{m - m_0 - \frac{1}{2}}{\sqrt{2}s}\right) \right\}, \qquad (6)$$

$$E_n = \frac{1}{2}\left\{ erf\left(\frac{n - n_0 + \frac{1}{2}}{\sqrt{2}s}\right) - erf\left(\frac{n - n_0 - \frac{1}{2}}{\sqrt{2}s}\right) \right\}, \qquad (7)$$

where $erf$ represents a Gaussian error function, $(m_0,n_0)$ is the sub-pixel emitter position, and $s$ is the SD of the Gaussian spot profile; i.e., 2.2 ± 0.020 pixels. Using these equations, the peak pixel intensity $I(0,0)$ (the number of detected photons at $(m,n)$ =(0,0)) when a single-molecule emitter is located exactly at the center of the pixel (0,0) is estimated to be only 1.1 photons.

Under such low signal conditions, in which the pixel intensities within the Gaussian spot profile fluctuate frame by frame at the level of single photons, the photoelectrons emitted at the photocathode of the image intensifier should be amplified until the readout noise of 37 root-mean-square electrons/pixel/frame, which also fluctuates spatiotemporally, becomes negligible. To satisfy this condition, the photon amplification gain of the image intensifier of 33,200 (the ratio of the number of photons emitted from the phosphor screen of the image intensifier vs. the number of detected photons) was used throughout this study (see the following estimates for details). The image intensifier was coupled to the CMOS sensor by an optical fiber bundle. Therefore, due to the 39% loss (61% coupling efficiency) by the optical fiber bundle and the quantum efficiency of the CMOS sensor of 40%, the overall electron amplification (after photon detection) was a factor of 8,100 (33,200 × 0.61 × 0.4). Since the quantum yield of the photocathode of the image intensifier was 0.4, the total amplification of the incident photon of this camera system was 3,240 (one photon that arrived on the photocathode of the image intensifier generated an average of 3,240 electrons in the CMOS chip).

The overall electron amplification of 8,100× (after photon detection) was selected based on the following two considerations. First, the probability that the electron signal at the CMOS sensor amplified from a single detected photon (at the photocathode of the image intensifier) becomes greater than the readout noise should be sufficiently high. As shown in Fig. S1 E, the probability of detecting a single photon (as normalized by the number of detected photons/frame at the saturating levels of electron amplification) was increased to 90.0% at an overall electron amplification of 8,100×.

Second, the upper limit of the overall electron amplification is given by the full-well capacity of 30,000 electrons for the individual pixel (for the 1024PCI sensor employed for most experiments in this study). Considering the stochastic fluctuation of the signal, we set the electron amplification of the image intensifier so that the average signal in the peak pixel of 1.1 electrons/pixel/frame was amplified to <1/3 of the full well capacity (<10,000 electrons); i.e., amplification by a factor of <9,090 (= 10,000/1.1). Combining these two considerations, we employed an overall electron amplification by a factor of 8,100 (33,200 at the image intensifier), 10% less than the factor 9,090, to further reduce the possibility of pixel intensity saturation. (For detecting dimers and larger oligomers, the image intensifier gain should be reduced, although the probability of missing monomers would increase slightly.)

(c) The design and conditions for visualizing and tracking single molecules for many frames (Fig. 2 C)

Note that these illumination and amplification conditions were selected so that we could track single molecules for many frames in the apical PM (e.g., for longer than 99 frames for 14% of the Cy3 molecules immobilized on the glass, under the conditions of three-frame gap closing, as shown in Fig. 2 C). When only much shorter tracking is required (such as a single frame or in the case of PALM), much higher excitation laser power density could be employed for better single-molecule localization

precisions. The best single-molecule localization precision we achieved with this camera system was 2.6 ± 0.099 nm (mean ± SEM), with a maximum number of detected photons/molecule/frame of 11,400 ± 700 (mean ± SEM; $n$ = 50 Cy3 molecules; Fig. 1, G and H; and Fig. S2 D), using a TIR illumination power density at 79 µW/µm² and a signal integration time of 16.7 ms. This is because, under these conditions, most Cy3 molecules become photobleached within a single frame (16.7 ms; i.e., virtually all of the photons possibly obtained from single molecules are concentrated in a single frame) due to the use of a TIR illumination density of 79 µW/µm², by which the photon emission rate from a single Cy3 molecule is saturated (Fig. 1 E).

**Estimation of the number of photons that can be emitted by a single Cy3 molecule during 0.1 ms: Triplet bottleneck saturation (Fig. 1, E and F; and Figs. S2 and S3)**
In ultrafast SFMI, the number of photons emitted from a single molecule during the duration of a single camera frame is the key limiting factor. This critically depends on how fast a molecule returns to the ground state from the excited state. In a conventional three-state model (ground, singlet-excited, and triplet-excited states), the maximum photon emission rate, $k_{em}$ (photons/s), can be described as (Schmidt et al., 1995):

$$k_{em} = \frac{\frac{\Phi}{\tau_s}}{(1 + k_{isc}\tau_T)} = \frac{\Phi}{(\tau_s + \Phi_T\tau_T)}, \qquad (8)$$

where $\tau_s$ and $\tau_T$ represent the lifetimes of the singlet- and triplet-excited states, respectively, $k_{isc}$ (= $\Phi_T/\tau_s$) is the intersystem crossing rate, $\Phi$ is the fluorescence quantum yield, and $\Phi_T$ is the quantum yield for triplet formation. Namely, the maximum photon emission rate, $k_{em}$, critically depends on the triplet yield and lifetime, and hence this phenomenon is known as "triplet bottleneck saturation."

For Cy3, $\tau_s \approx 1$ ns, $\tau_T \approx 1$ µs (in a specimen equilibrated with atmospheric molecular oxygen, the triplet lifetime is dominated by the collision rate of molecular oxygen with the fluorophore, which is about $10^6$/s; Kusumi et al., 1982), $\Phi$ = 0.15, and $\Phi_T$ = 0.003 were employed (Sanborn et al., 2007; Dempsey et al., 2011). These values provide a $k_{em}$ of 3.8 × $10^7$ photons/s. Therefore, the maximum number of photons available to form an image of a single Cy3 molecule, during the integration duration of 0.1 ms for each frame, is ∼3,800. Although this number would vary, depending on the spectroscopic parameters in the given environment, it provides a baseline for understanding such fast imaging. Assuming that 5—10% of the emitted photons will reach the intensifier photocathode (Rasnik et al., 2007; Gould et al., 2009), 190—380 photons will reach the photocathode of the image intensifier and be detected with a ≈40% quantum efficiency of the GaAsP photocathode at the Cy3 peak emission wavelength of 570 nm, providing the maximum number of photons detected during a 0.1-ms camera frame time of 80—150.

This estimate agrees well with the experimental results for evaluating the maximal photon emission rate of Cy3 by TIR illumination (here, since we are discussing saturation conditions, we focus on the results obtained by the TIR illumination rather

than the oblique illumination) shown in Fig. 1 E top and Fig. S2 A left (the data displayed in Fig. S2 A left show the relationship of the single-molecule localization error with the number of detected photons/spot/frame obtained at a frame rate of 10 kHz, which was found to be consistent with the theory developed previously; Mortensen et al., 2010). In Fig. 1, E and F (top), with an increase of the illumination intensity, both the number of detected photons/molecule/frame and the single-molecule localization precision were enhanced. However, the extents of the enhancement were saturated, because the number of photons that a single molecule can emit during the 0.1-ms frame time was limited. Under the near-saturation conditions at a laser power density of 79 μW/μm² at the sample plane, the number of detected photons was ≈100 photons/frame (Fig. 1 E top). This confirmed that the experimental result is consistent with the estimate of the maximum photon emission rate based on the three-state model (ground, singlet-excited, and triplet-excited states; 80—150 photons during 0.1 ms) as described in the previous paragraph, indicating that the saturation of Cy3 photoemission can be explained by the "triplet bottleneck saturation." Under the Cy3 saturation conditions, the improvement of the single-molecule localization precisions with an increase of the laser excitation intensity was limited to (saturated at) ≈20 nm for recordings at 10 kHz (Fig. 1 F top and Fig. S2 A left).

Among the eight dyes examined here, JF549, TMR, JF646, Atto647N, SeTau647, and Cy5 exhibited saturation more readily than Cy3 and Alexa555 (Fig. S2 B). Furthermore, the average numbers of detected photons from single molecules during the 0.1-ms frame time obtainable at saturation were smaller than those of Cy3 and Alexa555, and thus their single-molecule localization precisions at 10 kHz at saturation were worse than those of Cy3 and Alexa555 (Fig. S3). At saturation, Cy3 provided slightly better single-molecule localization precision at 10 kHz than Alexa555 (Fig. S3 B). Therefore, we primarily employed Cy3 throughout the remaining part of this report. Note that this result depended not only on the photophysical properties of individual dyes, but also on the wavelength dependence of the photocathode quantum yield of the image intensifier. The quantum yield is lower (≈0.2) for the near-IR dyes, JF646, Atto647N, SeTau647, and Cy5, as compared with that (≈0.4) for the Cy3, Alexa555, JF549, and TMR dyes.

### Determination of the number of detected photons/molecule/frame ($N$) (Fig. 1, E and G; and Fig. 2, A and B; and Figs. S2 and S3)

All of the fluorescent dye molecules used for determining the number of detected photons/molecule/frame were covalently bound to the coverslip coated with 3-APS, and 5xCy3-Tf was adsorbed on the coverslip coated with poly-D-lysine (see the previous subsection). The number of detected photons/molecule/frame was evaluated by multiplying the pixel intensity and the gain conversion factor of the developed camera system. This factor was determined by detecting the same number of photons on both the developed camera system and an Andor iXon DU-897 back-illuminated EM-CCD camera, with the known gain conversion factor of 0.03622 photons/pixel count at an EM gain of 50 and an imaging depth of 16 bits (measured by Zeiss for an

ELYRA P.1 PALM system). Briefly, the images of fluorescent beads immobilized on a coverslip were split by a 50/50 beam splitter, and each image was projected onto the new camera system and the EM-CCD camera at the same time. The number of detected photons from each fluorescent bead/frame was obtained by multiplying the gain conversion factor of the EM-CCD camera and the entire pixel intensities of the spot on the EM-CCD camera. The number of photons that reached the EM-CCD camera was then obtained by dividing it by the quantum efficiency of the EM-CCD camera (0.95). The same number of photons must have reached the photoelectric cathode of the developed camera, and therefore, by multiplying with the quantum efficiency of the developed camera (0.4), the number of detected photons/frame/spot was obtained. By comparing this number with the entire pixel count of the fluorescent spot, the conversion factor of the developed camera system was evaluated. At the typical overall electron amplification of 8,100× employed for the observations at a frame rate of 10 kHz (0.1-ms frame time) in the present research, the gain conversion factor of the developed camera system was estimated to be 0.00363 (0.00136) photons/pixel count in an imaging depth of 10 (12) bits for the 1024PCI (SA1) sensor.

### Detecting single-molecule fluorescence spots, determining their positions in the image, and linking the positions to generate trajectories

Each and every fluorescent spot in an image was detected and its position was determined, using an in-house computer program based on a spatial cross-correlation matrix (Gelles et al., 1988; Fujiwara et al., 2002; Fujiwara et al., 2016). The image (signal intensity profile) was correlated with a symmetric 2-dimensional Gaussian point spread function (PSF) with a standard deviation of 123 nm (kernel; 123 nm was used for Cy3 and the value was individually determined for each fluorescent probe; see the caption to Fig. S2; Mashanov and Molloy, 2007). The "maximum entropy" thresholding (Sahoo et al., 1988) was applied to the cross-correlation matrix. The spot representing a single molecule was detected as that containing an area with a size ≥5 pixels, which was defined as the size of the area where the values of the cross-correlation matrix were greater than or equal to the threshold value (a constant value for the whole image). The spot position (x, y coordinates) was determined in the following way. First, the centroid of the cross-correlation matrix was calculated, and then an 830-nm-diameter circular region (55.1 nm/pixel x 15 pixels) with its center placed at the centroid was determined as the region that includes the intensity profile of the fluorescent spot. Second, the spot position was determined by fitting the intensity profile in the circular region to a symmetric 2-dimensional Gaussian PSF, using the following formula:

$$\text{PSF}(x,y) = A exp\left(-\frac{(x-x_0)^2 + (y-y_0)^2}{2s_{xy}^2}\right) + B, \qquad (9)$$

where the fitting parameters were $A$ (amplitude), $(x_0, y_0)$ (the sub-pixel center position of the spot), and $s_{xy}$ (standard deviation of the symmetric Gaussian PSF), and the value for $B$ (background

intensity offset) was measured (background signal intensity divided by the number of pixels).

To generate trajectories from the determined coordinates of the fluorescent spots in an image sequence, the positions were linked when the displacement between the spots in the consecutive frames was smaller than the distance giving the ≥99.7% cumulative probability for a particle undergoing simple Brownian diffusion with an expected diffusion coefficient and a position localization error (Schütz et al., 1997; Sahl et al., 2010).

**Observing the time series of the number of detected photons/0.1-ms-frame from single Cy3 molecules, determining the signal-to-noise ratios (SNRs) of single Cy3 molecule images, and evaluating the durations in which single Cy3 molecules could be consecutively observed (the bright/dark periods [on/off-periods] in the images of single Cy3 molecules and the gap closing for short off-periods; Fig. 2)**

Time-dependent changes in the number of detected photons/0.1-ms-frame from single Cy3 molecules immobilized on a coverslip were observed at 10 kHz, under our standard oblique-angle laser illumination conditions of 23 μW/μm² (Fig. 2 A). The molecules that were observable in the first frame of the image sequence were selected. The number of detected photons during a single image frame in an 830 nm diameter (55.1 nm/pixel × 15 pixels) circular region including a fluorescent spot, called the "spot intensity," was measured and plotted against the frame number at 10 kHz. To evaluate the signal intensity in the background, the number of detected photons during a single image frame in an adjacent region with the same shape and size, called the "BG signal intensity" was measured. On- and off-periods were determined based on the thresholding employed for the spot detection.

The signal-to-noise ratio (SNR) for the image of a single molecule was determined in the following way (Fig. 2 B). From the time-dependent fluorescence signal intensities, the histograms of the numbers of detected photons/0.1-ms for the images of single molecules; i.e., the histograms for the "spot signal intensities" and the "background [BG] signal intensities" during the on-periods, were obtained, and their mean signal intensities ($I_{Spot}$ and $I_{BG}$, respectively) and standard deviations ($\sigma_{Spot}$ and $\sigma_{BG}$, respectively) were calculated. The SNR was evaluated using the following equation:

$$SNR = \frac{I_{Spot} - I_{BG}}{\left(\sigma_{Spot}^2 + \sigma_{BG}^2\right)^{1/2}}. \tag{10}$$

The mean SNR of single Cy3 molecules immobilized on the coverslip was 2.5 ± 0.11 for 50 Cy3 molecules with on-periods of 15—150 frames at 10 kHz, whereas the mean SNR on the apical PM was 1.8 ± 0.056 for 50 Cy3-DOPE molecules with on-periods of 1,000 frames at 10 kHz, as obtained under our standard oblique-angle laser illumination conditions of 23 μW/μm².

The distributions of the durations of the on-periods and those after neglecting the off-periods lasting for 1, 2, or 3 frames (called "gap closing") are shown in Fig. 2 C. The off-periods are induced by the occurrences of non-emission periods of a single dye molecule (which could induce no or dim images in a frame, depending on the duration of the non-emitting period; the off-periods tend to occur more often during the observations of living cells, probably due to the existence of various chemicals in the cell and the culture medium). The off-periods also occur when the signal from a single molecule is missed for short periods, due to the fluctuations of the fluorophore signal intensity ($I_{Spot}$) as well as the higher background ($I_{BG}$) and its fluctuation ($\sigma_{BG}$). Therefore, gap closing for 1–3 frames is often employed in single-molecule tracking studies. For describing the results obtained in living cells in the present report, we used 3-frame gap closing.

## Cell culture

Human T24 epithelial cells (the same as the ECV304 cell line used in the previous research [Murase et al., 2004], which turned out to be a sub-clone of T24; Dirks et al., 1999) were grown in Ham's F12 medium (Sigma-Aldrich) supplemented with 10% fetal bovine serum (Sigma-Aldrich). Cells were cultured on 12-mm diameter glass-bottom dishes (IWAKI), and single-molecule observations were performed two days after inoculation. For culturing cells expressing mGFP-paxillin, the glass surface was coated with fibronectin by an incubation with 5 μg/ml fibronectin (Sigma-Aldrich) in phosphate-buffered saline (PBS; pH 7.4) at 37°C for 3 h.

To form PM blebs (up to 20 μm in diameter) depleted of the actin-based membrane-skeleton, the cells were incubated with 1 mM menadione for 5 h at 37°C (Fujiwara et al., 2002), and then treated with 100 nM latrunculin A (a gift from G. Marriott, University of California, Berkeley) on the microscope stage for 5 min at 37°C. All measurements were performed within 5—15 min after the addition of latrunculin A (Suzuki et al., 2005; Fujiwara et al., 2016).

## Examination of the photo-induced damage to the cell during ultrafast SFMI (Fig. 2, D and E)

Cell viability was assessed by staining with 1 μM TOTO-3 iodide (Molecular Probes), which only stains dead cells (Zuliani et al., 2003), at 37°C for 5 min and then observing the stained cells using epi-illumination with a 594-nm laser. To obtain reference images of dead cells, the cells were treated with 100 μM $H_2O_2$ at 37°C for 1 h. The fluorescence intensity of TOTO-3 in the 5.5 × 5.5-μm area inside the nucleus was measured, and the histograms of signal intensities were obtained. Based on the negative and positive controls (Fig. 2, D and E), a threshold fluorescence intensity of $4.0 \times 10^5$ (arbitrary unit = AU) was selected to categorize the live and dead cells (96% [90%] of the negative [positive] control cells were categorized as alive [dead]).

Note the following. Once the desired single-molecule localization precision is chosen for each experiment, the required total number of photons emitted from the molecule during a frame to achieve the precision will be the same, irrespective of the frame rate (inverse of the frame time). Therefore, the total number of photons elicited by laser excitation during a frame time will be the same, regardless of whether the observations are made at video rate (33-ms frame time) or at 10 kHz (0.1-ms frame time), if photon emission saturation does not occur). Namely, the direct photo-damage to the cell per frame due to the

excitation laser illumination is expected to remain constant even when the laser illumination intensity and the frame rate are increased, as long as the single-molecule localization precision required for the experiment and the number of observed frames (not the total observation duration) are the same (in the absence of photon emission saturation).

However, direct experiments are required to test the photo-toxicity to cells, as shown here. This is partly because slight photon-emission saturation occurred under our standard illumination conditions of 23 µW/µm² (Fig. 1 E), and also because higher illumination photon densities could induce two-photon events and enhance the secondary reactions of the photo-induced molecules produced at high densities in the cell, which could generate cytotoxic substances.

### Preparation of Cy3-DOPE and Cy3-Tf and cell surface labeling

Cy3-DOPE was prepared and incorporated in the PM as described previously (Fujiwara et al., 2002; Murase et al., 2004), except that the final concentration of Cy3-DOPE added to the cells was 10 nM. Cy3-Tf used for the measurements at 6 kHz (0.167-ms resolution) was generated by incubating 6.5 µM monofunctional sulfo-Cy3 (GE Healthcare) with 6.3 µM human holo Tf (Sigma-Aldrich) in 0.1 M carbonate buffer (1 ml, pH 9.0) at 25°C for 60 min, followed by desalting column chromatography (PD-10, GE Healthcare, equilibrated and eluted with PBS) to remove the unreacted dye. The dye/protein molar ratio of the Cy3-Tf was 0.2. At this ratio, more than 90% of the fluorescent spots in the image are expected to represent single Cy3 molecules, based on the Poisson distribution, which was confirmed by single-step photobleaching of the fluorescent spots of Cy3-Tf (as well as Cy3-DOPE). Tf conjugated with multiple Cy3 molecules, used for the measurement at 45 kHz (0.022-ms resolution), was generated by incubating 1.25 mM monofunctional sulfo-Cy3 with 6.3 µM human holo Tf (mixed at a molar ratio of ~200:1), which provided Cy3-Tf with the dye/protein molar ratio of ≈5 (called 5xCy3-Tf).

To label TfR in the PM with Cy3-Tf, first the Tf (originated from FBS) already bound to TfR on the cell surface was partially removed by incubating the cells in 1 ml Hanks' balanced salt solution buffered with 2 mM TES (Dojindo), at pH 7.4 (HT medium) and 37°C for 10 min, and then after washing, HT medium containing 10 nM Cy3-Tf (or 5xCy3-Tf) was added to the cells at a final concentration of 0.5 nM. At this concentration, single molecules of Cy3-Tf bound to the apical PM could be observed without replacing the medium with the Cy3-Tf-free medium. The observations were completed within 10 min after the Cy3-Tf addition.

### Halo-TfR and mGFP-paxillin expression in T24 cells and TMR labeling of Halo-TfR

The cDNA encoding human TfR (GenBank: M11507.1) fused to the Halo-tag protein at the TfR's N-terminus (Halo-TfR) was generated by replacing the cDNA encoding the EGFP protein, in the EGFP-TfR plasmid, with that of the Halo-tag protein (Promega), with the insertion of a 45-base linker (15 amino acids, with the sequence SGGGG ×3) between Halo and TfR. The cDNA encoding human paxillin isoform alpha (NCBI reference

sequence: NM_002859.3; cloned from the human WI-38 cell line) fused to mGFP at the N-terminus was subcloned into the EBV-based episomal vector, pOsTet15T3 (gift from Y. Miwa, University of Tsukuba, Japan), which bears the tetracycline-regulated expression units, the transactivator (rtTA2-M2), and the TetO sequence (a Tet-on vector; Shibata et al., 2012; Shibata et al., 2013). To avoid perturbing the FA structure by over-expressing mGFP-paxillin, doxycycline induction was not employed, but leaky expression was used. However, due caution is required when interpreting how effectively mGFP-paxillin imitates the behaviors of native paxillin.

T24 cells were transfected with the cDNA encoding mGFP-paxillin, using the Lipofectamine LTX and Plus reagents (Invitrogen) and with the cDNA encoding Halo-TfR using Nucleofector 2b (Lonza), following the manufacturers' recommendations. To covalently link TMR to Halo-TfR, T24 cells coexpressing Halo-TfR and mGFP-paxillin were incubated in HT medium, containing 10 nM TMR-conjugated Halo-ligand (Promega) at 37°C for 1 h, and then washed three times with HT medium. The remaining unbound ligand in the cytoplasm was removed by incubating the cells in HT medium for 30 min, and then washing the cells three times with HT medium.

### EGFR-Halo expression in T24 cells and its TMR labeling

The cDNA encoding human EGFR (GenBank: X00663.1) tagged with Halo (EGFR-Halo) was generated by replacing the cDNA encoding the YFP protein, in the EGFR-YFP plasmid (a gift from I. R. Nabi, University of British Columbia, Canada), with that of the Halo 7-tag protein (Promega), with the insertion of a 63-base linker (21 amino acids, with the sequence RVPRDPSGGGGSG GGGSGGGG) between EGFR and Halo.

T24 cells were transfected with the cDNA encoding EGFR-Halo, using a Nucleofector 2b device, and then starved in serum-free Ham's F12 medium for ≈16 h. The covalent complex between TMR and EGFR-Halo was formed in the same way as in the labeling of Halo-TfR. For EGF stimulation, HT medium containing 20 nM human EGF (Sigma-Aldrich) was added to the cells at a final concentration of 10 nM (1 ml of 20 nM EGF in HT medium was added to the glass-bottom dish containing 1 ml HT medium). The measurements were performed before and during 2.5—5 min after the EGF addition.

### Ultrafast SFMI of fluorescent probes bound to coverslips and the molecules in the live-cell PM

Each individual fluorescently labeled molecule was observed at 37 ± 1°C, using the oblique-angle and TIR illumination modes of a home-built objective lens-type TIRF microscope (based on an Olympus IX70 inverted microscope), which was modified and optimized for the camera system developed here. The beam was attenuated with neutral density filters, circularly polarized, and then steered into the edge of a high numerical aperture (NA) oil immersion objective lens (UAPON 150XOTIRF, NA = 1.45, Olympus), focused on the back focal plane of the objective lens. Both TIR and oblique-angle illuminations were used for Cy3 at 10 and 30 kHz, and 5xCy3-Tf at 45 kHz. For other molecules and conditions, only the TIR illumination was used. For exciting Cy3, Alexa555 (Invitrogen), JF549 (Janelia Fluor 549; Tocris

Bioscience), and TMR (Molecular Probes and Promega), a 532-nm laser (Millennia Pro D2S-W, 2W, Spectra-Physics; 1.8–16 mW for the ≈14-μm diameter observation area) was employed. For exciting JF646 (Janelia Fluor 646; Tocris Bioscience), Atto647N (ATTO-TEC), SeTau647, and Cy5 (sulfo-Cy5; GE Healthcare), a 660-nm laser (Ventus, 750 mW, Laser Quantum; 0.41–4.5 mW for the ≈12-μm diameter observation area) was used.

The illumination laser power density was determined as follows. When the field stop was fully open, the excitation laser illuminated a circular (2-dimensional Gaussian) area with a 12.5-μm radius (standard deviation) on the sample plane. By adjusting the field-stop size, the central part of the excitation laser beam was selected to form a circular area with a radius of 7 μm on the sample plane, approximating the largest view-field of 14 × 14 μm² employed in this research, and the laser power after the objective lens was measured. Since the dimmest intensity in the 7-μm Gaussian area was ≈85% of the peak intensity, the laser power density was calculated assuming a uniform laser intensity in the circular area with a 7-μm radius; i.e., the measured power was simply divided by the area size of 154 μm² (π × 7²).

To observe single fluorescent molecules immobilized on coverslips, dye molecules were covalently linked to 3-aminopropyltriethoxysilane (APS)-coated coverslips (custom-ordered from Matsunami). Briefly, 0.2—1 nM succinimidyl ester-modified dye molecules in HT medium were placed on the 3-APS-coated coverslip at 25°C (10-min incubation), and then the coverslip was washed five times with 1 ml HT medium. To immobilize 5xCy3-Tf molecules on the glass surface, the 12-mm diameter glass-bottom dishes were coated with poly-D-lysine, and then incubated in HT medium containing 0.1 nM 5xCy3-Tf at 25°C for 10 min.

The method for evaluating the position determination precisions of Cy3-labeled molecules bound on the coverslips is described in the caption to Fig. S2. The position determination precisions for the molecules in the apical PM were estimated using the ensemble-averaged MSD-Δt plots for single-molecule trajectories of Cy3-DOPE and Cy3-Tf (5xCy3-Tf) bound to TfR, as shown in Fig. S4.

To observe fluorescent molecules bound to the phospholipid and transmembrane proteins TfR and EGFR in the apical and basal PMs of live cells, the oblique-angle and TIR illumination modes were employed, respectively. To observe single molecules located outside the focal adhesion in the basal PM using ultrafast SFMI, the focal adhesion was visualized using mGFP-paxillin. Immediately before conducting ultrafast SFMI, the mGFP-paxillin image was obtained in the same view-field using the TIR illumination (0.063 μW/μm² at the specimen using a Spectra-Physics (Cyan-PC5W) 488-nm laser; using a frame rate of 60 Hz, averaged over 10 s).

## Quantitative analysis of single-molecule trajectories based on the MSD-Δt plots (Fig. 4, A a and B)

For each single-molecule trajectory, the one-dimensional MSD(x or y) for the x- or y-direction for every time interval was calculated according to the following formula:

$$MSD_x(n\delta t) = \frac{1}{N-1-n} \sum_{j=1}^{N-1-n} \{x(j\delta t + n\delta t) - x(j\delta t)\}^2. \quad (11)$$

$$MSD_y(n\delta t) = \frac{1}{N-1-n} \sum_{j=1}^{N-1-n} \{y(j\delta t + n\delta t) - y(j\delta t)\}^2, \quad (12)$$

where $\delta t$ is the frame time, ($x(j\delta t + n\delta t)$ and $y(j\delta t + n\delta t)$) describe the position of the molecule following a time interval $n\delta t$ after starting at position ($x(j\delta t), y(j\delta t)$), $N$ is the total number of frames in the sequence, $n$ and $j$ are positive integers, and $n$ determines the time increment. In the quantitative analysis of the trajectories, 1-dimensional MSD-Δt plots were employed.

Each single-molecule trajectory was classified into a suppressed-, simple-Brownian-, or directed-diffusion mode using the 2-dimensional MSD-Δt plot, which is the sum of the MSD-Δt plots for the x- and y-directions (Fig. 4, A a). The basic idea for the classification is shown in Fig. 4, B a (Kusumi et al., 1993). The classification was based on the relative deviation (RD), which describes the long-term deviation of the actual mean-square displacement, $MSD(N,n)$ at the time $n\delta t$, from the expected MSD based on the initial slope of the MSD-Δt plot for molecules undergoing ideal simple-Brownian diffusion, $4D_{2-4}\,n\delta t$; i.e., $RD(N,n) = MSD(N,n)/[4D_{2-4}\,n\delta t]$ ($N$ = the number of frames in a full trajectory, $n$ = the increment number of frames used for the analysis with the MSD-Δt plot, $1 \le n \le N$, and $\delta t$ = duration of each frame. Therefore, $\Delta t = n\delta t$, which is the x-axis of the $MSD(N,n)$-Δt plot; and $D_{2-4}$ is the short-time diffusion coefficient determined from the slope of the second, third, and fourth points in the MSD-Δt plot). The RD value is <<, ≈, or >> 1, when the molecules are undergoing suppressed, simple-Brownian, or directed diffusion, respectively.

In Fig. 4, B b, the distributions of $RD(N,n)$ s for the Cy3-DOPE ($RD(1,000, 100)$, $\delta t$ = 0.1 ms) and TfR trajectories ($RD(1,000, 200)$, $\delta t$ = 0.167 ms; shaded bars; $n$ = 50 and 20 for the intact and actin-depleted blebbed PM), as well as for simple-Brownian particles generated by a Monte Carlo simulation (open bars; $n$ = 5,000), are shown. Based on the distribution of the $RD(N,n)$ values determined for simulated simple-Brownian particles (open bars), the $RD(N,n)$ values giving the 2.5 percentiles of the particles from both ends of the distribution, referred to as $RD_{min}$ and $RD_{MAX}$, were obtained (red and blue vertical lines in Fig. 4, B b, respectively; they depend on both $N$ and $n$). Each experimental single-molecule trajectory was classified into the suppressed (confined and hop)-diffusion mode if its $RD(N,n)$ value (shaded bar) was smaller than $RD_{min}$ (and into the directed-diffusion mode if its $RD(N,n)$ value was larger than $RD_{MAX}$).

## Monte Carlo simulations for the hop diffusion in the intact PM and simple-Brownian diffusion in the actin-depleted blebbed PM (Fig. 5 B)

Monte Carlo simulations for hop diffusion were performed as described previously (Ritchie et al., 2005). The hop-diffusion simulation parameters used are the following ($p$ represents the probability that a hop movement to an adjacent compartment takes place when the diffusing molecule enters the boundary).

5xCy3-Tf-TfR at 45 kHz and TfR at 6 kHz: $D_{micro}$ = 4.5 μm$^2$/s, $L$ = 100 nm, $p$ = 0.00045.
Cy3-DOPE at 10 kHz: $D_{micro}$ = 9 μm$^2$/s, $L$ = 100 nm, $p$ = 0.00065.

The simulation parameters used for simple-Brownian diffusion in the actin-depleted blebbed PM are the following.

TfR at 6 kHz: $D_{micro}$ = 3.3 μm$^2$/s. Cy3-DOPE at 10 kHz: $D_{micro}$ = 6.0 μm$^2$/s.

A Gaussian localization error of 50 nm was added (see Fig. S4 B). The MSD-$\Delta t$ plot ensemble-averaged over 1,000 trajectories was obtained, and the offset due to the localization error was estimated and subtracted based on the y-intercept (by extrapolation) of the linear-fit function for the second, third, and fourth steps in the MSD-$\Delta t$ plot, as performed for experimental trajectories.

### Online supplemental material

Fig. S1 illustrates the method we used to determine the electron amplification gain of the image intensifier. Fig. S2 displays the relationship of the single-molecule localization precision with the number of detected photons/molecule/frame for various fluorescent dye molecules. Fig. S3 shows that the numbers of detected photons/molecule/frame from single molecules are saturated under stronger excitation laser intensities, and thus the improvement of single-molecule localization precision with an increase of the laser intensity is limited. Fig. S4 presents the MSD-$\Delta t$ plots ensemble averaged over all trajectories obtained for Cy3 molecules immobilized on coverslips or diffusing in the apical and basal PM of T24 cells, providing estimates of single-molecule localization errors. Fig. S5 shows the TILD method. Video 1 exhibits single Cy3 molecules covalently linked to the glass surface observed at 10 kHz. Video 2 shows single Cy3-DOPE molecules diffusing in the intact apical PM in a view-field of 256 × 256 pixels, observed at 10 kHz. Video 3 shows enlarged views of single Cy3-DOPE molecules diffusing in the intact apical PM and the actin-depleted blebbed PM, observed at 10 kHz. Video 4 shows enlarged views of single TfR molecules, bound by Cy3-Tf, diffusing in the intact apical PM and the actin-depleted blebbed PM, observed at 6 kHz. Video 5 shows single TfR (Halo-TMR) molecules diffusing in the basal PM outside (top movies) and inside (bottom movies) the focal adhesion regions marked by mGFP-paxillin, observed at 6 kHz. Video 6 shows single EGFR (Halo-TMR) molecules diffusing in the basal PM before and 2.5 min after the addition of 10 nM EGF, observed at 6 kHz. Supplemental theory 1 describes the expected distribution of the residency times based on the hop-diffusion theory we developed. Supplemental theory 2 describes the function describing the MSD-$\Delta t$ plot for particles undergoing hop diffusion used for "hop-diffusion fitting."

### Data availability

Data supporting the findings of this study are available from the corresponding author upon reasonable request. The codes for the TILD analysis and for calculating the MSD based on the theoretical equation describing hop diffusion (to be used for hop-diffusion fitting using the MATLAB nonlinear fit function)

are freely available for academic use at https://github.com/kusumi-unit/WinTILD/ and https://github.com/kusumi-unit/hop-diffusion-function/, respectively. Our method for obtaining the coordinates of the observed single molecules is basically the same as that described by Crocker and Grier (1996). The MATLAB codes based on this method written by Blair and Dufresne in 2009 are available at https://site.physics.georgetown.edu/matlab/code.html. Our codes for this have been integrated into a much larger, complex software package and they cannot be extracted in a useful way. However, the entire software is available from the corresponding author upon reasonable request (we will provide personal guidance on how to use it, as about half of its manual and comments are written in Japanese). The camera system developed here can now be custom ordered from Photron. The camera systems are also made available in our laboratories upon reasonable request.

## Acknowledgments

We thank Profs. G. Marriott of the University of California, Berkeley, Y. Miwa of the University of Tsukuba, T. Fujimoto of Nagoya University School of Medicine, and L. Looger of the Janelia Research Campus, for their kind gifts of latrunculin A, the pOsTet15T3 vector, the cDNA encoding caveolin-1-EGFP, and the mEos2 plasmid, respectively. We are grateful to Profs. P.T. Kanchanawong of the National University of Singapore, K. Jacobson of the University of North Carolina, and K. Gaus of the University of New South Wales Sydney for their critical reading of the manuscript, constructive comments, and revision suggestions. We also thank Mr. K. Hanaka and Prof. M. Kengaku of Kyoto University for their enthusiastic encouragement of this research, Ms. M. Yahara and Ms. A. Chadda for constructing various cDNAs, Ms. J. Kondo-Fujiwara and Mr. K. Kanemasa for preparing the figures, and the members of the Kusumi laboratory for helpful discussions.

This work was supported in part by Grants-in-Aid for Scientific Research from the Japan Society for the Promotion of Science (JSPS; Kiban B to T.K. Fujiwara [16H04775, 20H02585], Kiban B to K.G.N. Suzuki [18H02401, 21H02424], and Kiban S and A to A. Kusumi [16H06386 and 21H04772, respectively]), Grants-in-Aid for Challenging Research (Exploratory) from JSPS to T.K. Fujiwara (18K19001) and to A. Kusumi (22K19334), and Grants-in-Aid from the Ministry of Education, Culture, Sports, Science and Technology of Japan (MEXT) for Transformative Research Areas (A) to T.K. Fujiwara (21H05252) and for Innovative Areas to K.G.N. Suzuki (18H04671), a Japan Science and Technology Agency (JST) grant in the program of the Core Research for Evolutional Science and Technology (CREST) in the field of "Biodynamics" to A. Kusumi and that in the field of "Extracellular Fine Particles" to K.G.N. Suzuki (JPMJCR18H2), a JST grant in the program of "Development of Advanced Measurements and Analysis Systems" to A. Kusumi and T.K. Fujiwara, and by a grant from the Takeda Foundation to K.G.N. Suzuki WPI-iCeMS of Kyoto University is supported by the World Premiere Research Center Initiative (WPI) of MEXT.

Author contributions: T.K. Fujiwara and A. Kusumi conceived and formulated the project. T.K. Fujiwara, S. Takeuchi,

Y. Nagai, K. Iwasawa, and A. Kusumi developed the ultrahigh-speed camera system, T.K. Fujiwara, K. Iwasawa, T.A. Tsunoyama, and A. Kusumi developed an ultrafast SFMI station based on the newly developed camera system, and T.K. Fujiwara, T. Kalkbrenner, T.A. Tsunoyama, and A. Kusumi tested the camera system on the developed station. T.K. Fujiwara, T.A. Tsunoyama, K.G.N. Suzuki, and A. Kusumi designed the biological experiments and participated in discussions. T.K. Fujiwara performed virtually all of the ultrafast SFMI experiments and the data analysis. T.K. Fujiwara and K.P. Ritchie developed the compartment detection algorithm. Z. Kalay, based on discussions with T.K. Fujiwara, derived the equation describing the MSD-$\Delta t$ plot for particles undergoing hop diffusion and developed the theory for the distribution of the dwell lifetimes within a compartment for particles undergoing hop diffusion. T.K. Fujiwara, K.P. Ritchie, Z. Kalay, and A. Kusumi evaluated the data. T.K. Fujiwara, Z. Kalay, and A. Kusumi wrote the manuscript, and all authors discussed the results and participated in revising the manuscript.

Disclosures: S. Takeuchi and Y. Nagai are employees of Photron Limited, a manufacturer of high-speed digital cameras for industrial and scientific applications. T. Kalkbrenner is an employee of Carl Zeiss Microscopy GmbH, a manufacturer of microscope systems for life sciences and materials research. Authors T.K. Fujiwara, Z. Kalay, T.A. Tsunoyama, K. Iwasawa, K.P. Ritchie, K.G.N. Suzuki, and A. Kusumi declare that they have no competing interests.

Submitted: 28 October 2021

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

# Supplemental material

**JCB**

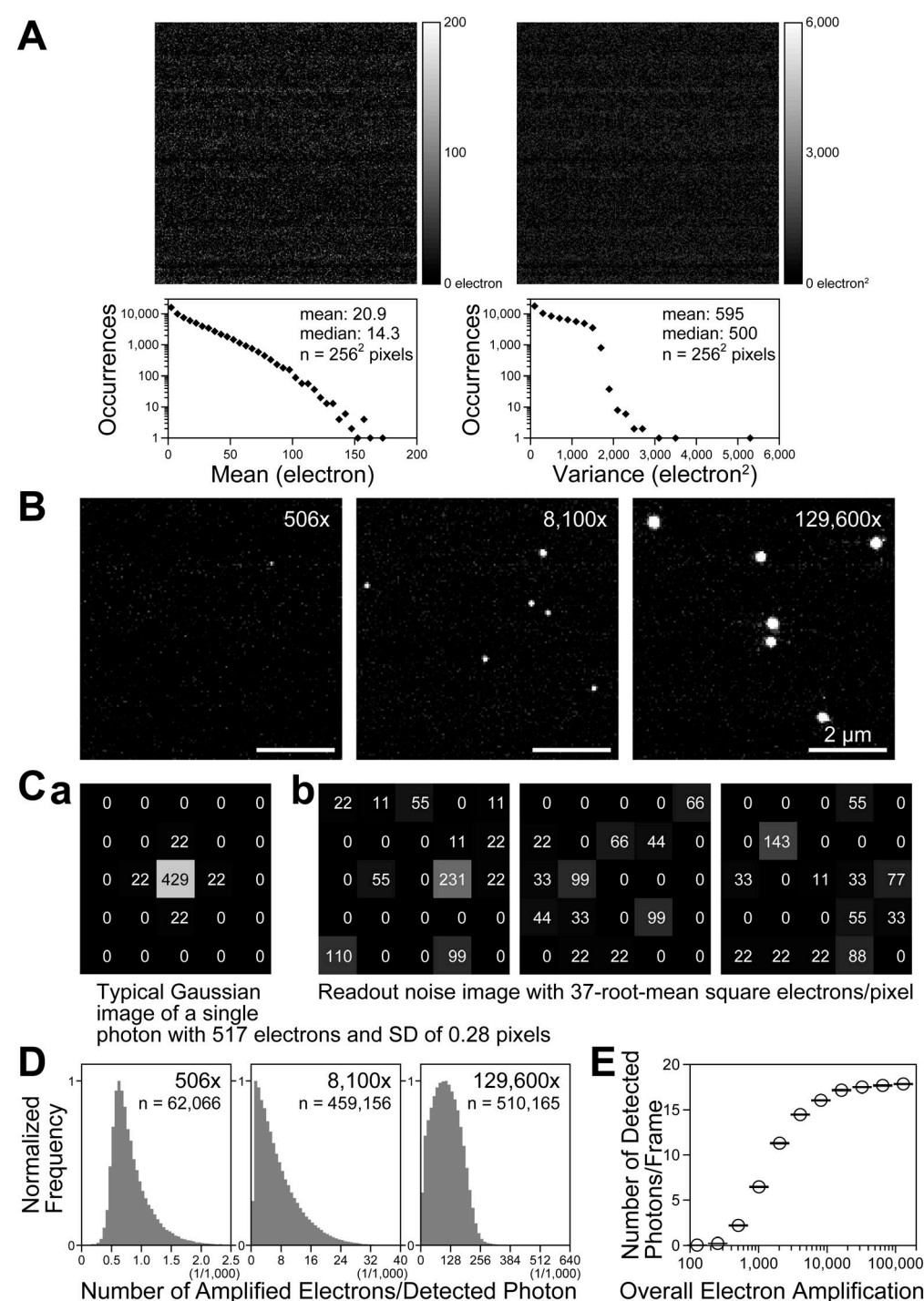

Figure S1. **Establishing the intensifier set up for detecting single photons using the electron amplification of the image intensifier (8,100×). (A)** The readout noise of individual pixels of the CMOS chip used here. The map (top) of the means (left) and variances (right) of pixels (256 × 256 pixels) and their histograms (bottom; Huang et al., 2013) for 1,000 frames obtained at 10 kHz. **(B)** Typical images of single photons obtained as a function of the intensifier gain, using the newly developed camera system. They were obtained by amplifying single electrons emitted from the image-intensifier photocathode by the arrivals of single photons (uniform Köhler illumination by the strongly attenuated halogen lamp of the microscope). The amplifications were 506, 8,100, and 129,600 (increases by a factor of 16 from left to middle and middle to right images). Note that the term "overall electron amplification (of the camera system)" always excludes the 40% quantum efficiency of the image intensifier photocathode throughout this report, because we discuss the amplification of the number of electrons from the photoelectrons emitted by the photocathode. The range of the gray levels of the images shown here (both B and C) was set for 8 bits from 0 to 550 electrons/pixel at the CMOS sensor (from black to white). **(C)** As an example, we show that an overall electron amplification of 506× by the camera system would not be sufficient for the consistent detection of single photons, due to the readout noise. **(C a)** A schematic figure showing a 7 × 7 pixelated image for a single detected photon amplified to a total of 517 electrons (without noise and background; computer-generated, assuming an overall amplification of 506× with an SD of 0.28 ± 0.085 pixels as in B left). The number of amplified electrons in each pixel on the CMOS sensor is shown. A full well of 45,000 electrons/pixel of the CMOS sensor (SA1) is scaled to 12 bits (4,095 camera counts, and hence a unit camera count = 10.99 [≈11] electrons per pixel), and thus

each number is a multiple of 11 electrons. **(C b)** Three arbitrarily selected (experimentally obtained) 7 × 7 pixelated images representing the spatial distributions of the readout noise of the CMOS sensor employed in this study (37-root-mean-square electrons/pixel/frame; see "Ultrahigh-speed intensified CMOS camera system: Design and operation" in Materials and methods). To detect a (photon-converted) emitted electron, its image, such as that shown in a, must be detectable in the presence of spatiotemporally varying noise, as shown here. The detectability will be enhanced by an increase of the electron amplification. See D. **(D)** Stochastic gain variations (fluctuations) of the image intensifier. At the level of detecting single photons, the gain variations are large (at the level of detecting single molecules, relative variations will become smaller due to averaging over all detected photons). In these histograms, the distributions of the number of total electrons stored at the CMOS sensor of the camera system for each single detected photon at the image-intensifier photocathode (i.e., for each discernible spot in images like those in B) are shown for the overall electron amplifications of 506×, 8,100×, and 129,600× (the y axis is normalized by the peak value in each histogram; the full x scale is increased by a factor of 16 from the left figure to the middle figure and from the middle figure to the right figure). The spots in the images (like those in B) were identified and the total number of electrons in each spot was evaluated using the functions of the ThunderSTORM plugin of ImageJ (Ovesný et al., 2014). For the spot detection with localization precisions at the pixel level (the peak pixel), we employed the "Wavelet filtering" (B-Spline order = 3 and B-Spline scale = 2.0) and the "Local maximum method" (Peak intensity threshold = 4.5 and Connectivity = 8-neighborhood). For the determination of the total number of electrons in a spot (and subpixel localization of each spot), we performed the Gaussian fitting of the image (i.e., the number of electrons/pixel in 11 × 11 pixels surrounding the peak pixel) and then integrated the best-fit function, using the Subpixel localization of molecules. These histograms were obtained using six 5,000-frame image sequences recorded at 10 kHz with a frame size of 13.4 × 13.4 µm on the focal plane, detecting 62,066, 459,156, and 510,165 photons (electrons emitted from the photocathode of the image intensifier) for overall amplifications of 506×, 8,100×, and 129,600×, respectively. They represent both the stochastic gain variations and the detectability of a photon image produced by the amplified electrons. See the histogram for 506×. The occurrences of the number of amplified electrons/detected photon sharply decreased when the number of amplified electrons was reduced below ≈650 electrons (the peak in the histogram). This is very likely due to a sharp reduction in the detectability of the spots produced by <650 electrons. Namely, when the signal intensity (the number of electrons) after amplification is small, the chance that the signal becomes less than the readout noise increases, because the readout noise pattern also fluctuates spatiotemporally (see C). **(E)** The probability of detecting a single photon was increased to 90.0% of the saturation level, at an overall electron amplification of 8,100×. Here, the number of detected photons per frame (mean ± SD); i.e., the number of discernible spots in the images like those in B (detected by the method described in D), is plotted as a function of the overall electron amplification (averaged over six 5,000-frame image sequences). With an increase in the overall electron amplification, the number of discernible spots increases. At an overall electron amplification of 8,100×, the number of discernible spots was 90.0% of the saturated number of spots (i.e., at the maximal overall electron amplification possible with the present instrument, which is 129,600× amplification). At the overall amplifications giving the saturated number of spots, virtually every photoelectron emitted from the photocathode is considered to be detected. Therefore, throughout the present research, we employed 8,100× as the overall electron amplification of the image intensifier (see "Ultrahigh-speed intensified CMOS camera system: Design and operation" in Materials and methods).

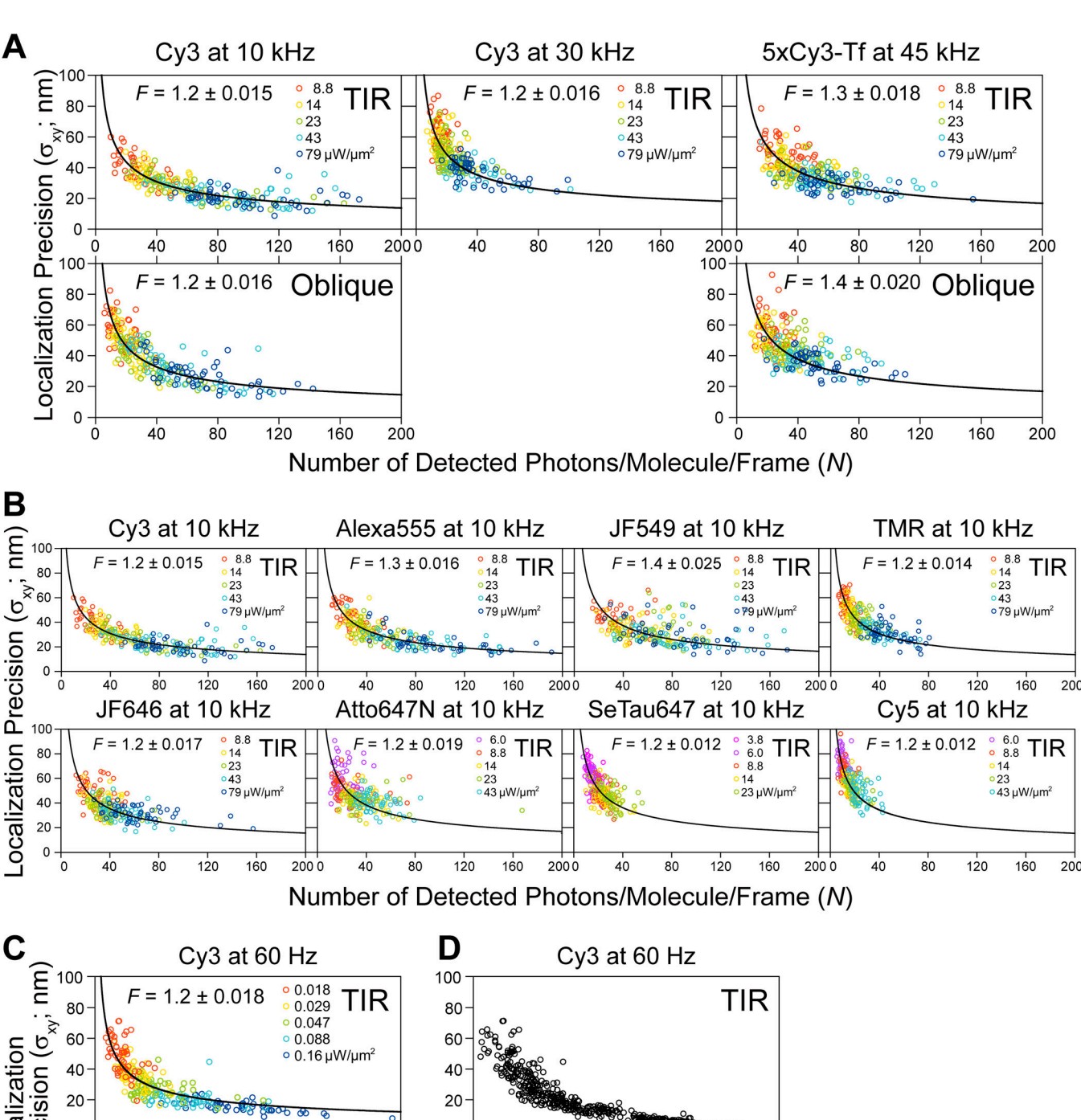

Figure S2. **Establishing the photophysics of various fluorescent probe molecules by independently evaluating the number of detected photons (proportional to the emitted photons) from a single fluorescent molecule during a single frame time ($N$; x-axis) and single-molecule localization precision ($\sigma_{xy} = [\sigma_x + \sigma_y]/2$; y-axis), under various excitation laser powers and single-frame durations.** These plots can be fitted well with the equation later in this legend, indicating that these measurements were performed with satisfactory accuracies. With an increase of the excitation laser power (at the sample), some dyes emit more photons than others, showing that they are more suitable for ultrafast SFMI. These curves are useful for determining the fluorescent probes to be used in the experiments and for predicting the single-molecule localization precisions that can be obtained under the given excitation laser powers. The results for 30, 10, and 0.06 kHz; i.e., the frame times of 0.033, 0.1 and 16.7 ms, respectively, are shown (45 kHz/0.022 ms for 5xCy3-Tf is also shown). For the method to evaluate $N$, see the subsection "Determination of the number of detected photons/molecule/frame ($N$)" in Materials and methods. In high-speed single fluorescent-molecule imaging, one of the crucial problems is whether single fluorescent molecules emit sufficient numbers of photons (during a single frame time) required for obtaining the desired single-molecule localization precisions. The results shown here demonstrate that a 10-kHz frame rate is applicable for various dye molecules, and Cy3 could even be used at 30 kHz. These plots were fitted well by the theoretical equation derived previously (Mortensen et al., 2010), indicating that the developed camera system functions as planned, even at high frequencies. The excess noise factor ($F$) of the developed camera system was evaluated by this fitting. Throughout this report (except for the measurements in the plasma membrane (PM), as described in

Fig. S4), the localization precision ($\sigma_{xy}$) is defined as $[\sigma_x + \sigma_y]/2$, where $\sigma_x$ and $\sigma_y$ are the standard deviations of the $x$ and $y$ position determinations, respectively, following the convention of the super-resolution imaging field (Dietrich et al., 2002; Martin et al., 2002). $\sigma_{xy}$ was determined in 15 consecutive frames for $n = 50$ trajectories for each condition. All of the fluorescent dye molecules were covalently bound to coverslips coated with 3-aminopropylethoxysilane, and 5xCy3-Tf was adsorbed on the coverslip coated with poly-D-lysine (Materials and methods). **(A and B)** Plots for single Cy3 molecules observed at 10 and 30 kHz and single 5xCy3-Tf molecules observed at 45 kHz (A) and those for various fluorescent molecules observed at 10 kHz (B). Five TIR laser illumination intensities were employed for each dye (50 molecules for each laser intensity), as indicated by the different colors of the data points. Various ranges of the laser power densities were used for different dyes, because the dyes are saturated differently (shown in each box). The plots ($\sigma_{xy}$ vs. $N$) shown in A and B could be fitted well (non-linear least-squares fitting by the Levenberg–Marquardt algorithm) using the following equation derived previously (Mortensen et al., 2010).

$$\sigma_{xy} = F\left\{\frac{16(s^2 + a^2/12)}{9N} + \frac{8\pi b^2(s^2 + a^2/12)^2}{a^2 N^2}\right\}^{1/2}$$ , where $F$ is the sole fitting parameter, representing the excess noise factor (a coefficient describing the stochastic

gain fluctuation in the electron amplification process in the image intensifier; $F$ is shown in each box, but its value is 1.2—1.4 for all cases), $s$ is the standard deviation of the Gaussian spot profile, 123 ± 1.1 nm for Cy3 on the sample plane (determined by the Gaussian fitting of each image for 50 Cy3 molecules immobilized on the glass excited by the TIR illumination at 79 µW/µm²; compared with our standard condition of the oblique illumination at 23 µW/µm², these observation conditions provided ∼3 times more detected photons; see Fig. 1 E, top; note that $s$ depends on the observed fluorescent molecules), $a$ is the pixel size (55.1 nm), and $b$ is the standard deviation of the background noise. (For example, 0.038 ± 0.059 detected photons/pixel/frame [mean ± SD] for the TIR illumination and 0.035 ± 0.058 detected photons/pixel/frame for the oblique illumination at 10 kHz; $n = 76,800$ pixels = 32 × 32 pixels × 15 frames x 5 different positions.) The estimated excess noise factor $F$ of the image intensifier shows that it is comparable to or slightly smaller (less noisy) than that of the EM-CCD electron multiplier ($F = 1.4$). Cy3 exhibited the least tendency to saturate, and thus provided better single-molecule localization precisions, consistent with the analysis results shown in Fig. S3, A and C. Note that $s$ and $b$ were determined for each fluorescent probe (with different illumination and excitation wavelengths and optics). 5xCy3-Tf data are considered to represent fluorescent spots generated by various numbers of Cy3 molecules placed within a few nanometers, mostly in the range of 3 to 8 molecules (1 and 2 Cy3 molecules/Tf, representing ≈12% of the 5xCy3-Tf spots, gave low signals, inducing extremely large errors in single-molecule localizations; meanwhile, the probability of 9 or more Cy3 molecules being attached to a Tf molecule will be <7%). Due to the photobleaching of multiple Cy3 molecules bound to a Tf molecule, the numbers detected on a Tf molecule decreased quickly upon laser illumination. **(C)** Plot for single Cy3 molecules observed at 60 Hz, with TIR illumination laser power densities ≤0.16 µW/µm² (indicated by different colors of the data points; 50 molecules for each laser intensity). The excess noise factor $F$ was estimated to be 1.2, consistent with the results shown in A. **(D)** Summary plot for single Cy3 molecules observed at 60 Hz, with TIR illumination laser power densities up to 79 µW/µm²: 0.018, 0.029, 0.047, 0.088, 0.16 (employed for the plot in c), 0.48, 1.6, 4.8, 14, 23, 43, and 79 µW/µm² (note that in this plot, in contrast to the others, the x-axis is in the log scale). The single-molecule localization precisions obtained with the laser power densities equal to and >14 µW/µm² were calculated using the equation above with $F = 1.2$, as found in C. This method for obtaining the single-molecule localization precisions employed here is different from that used for evaluating the precisions shown in Fig. 1 F and Fig. S2, A–C; and Fig. S3, B and C. Since most Cy3 molecules were photobleached within a single 16.7 ms frame, the more-prevalent method could not be employed. The x-axis of this figure covers the entire practical scale for the number of detected photons/molecule/frame ($N$) for a single Cy3 molecule, from 25.0 ± 1.4 at a laser power density of 0.018 µW/µm² up to 11,400 ± 700 at 79 µW/µm² (mean ± SEM). This upper limit was given by the photobleaching and excitation power saturation of Cy3, and provided the best single-molecule localization precision of 2.6 ± 0.099 nm (mean ± SEM) for Cy3 (no further improvements could be obtain even by employing higher laser intensities; Fig. 1, G and H).

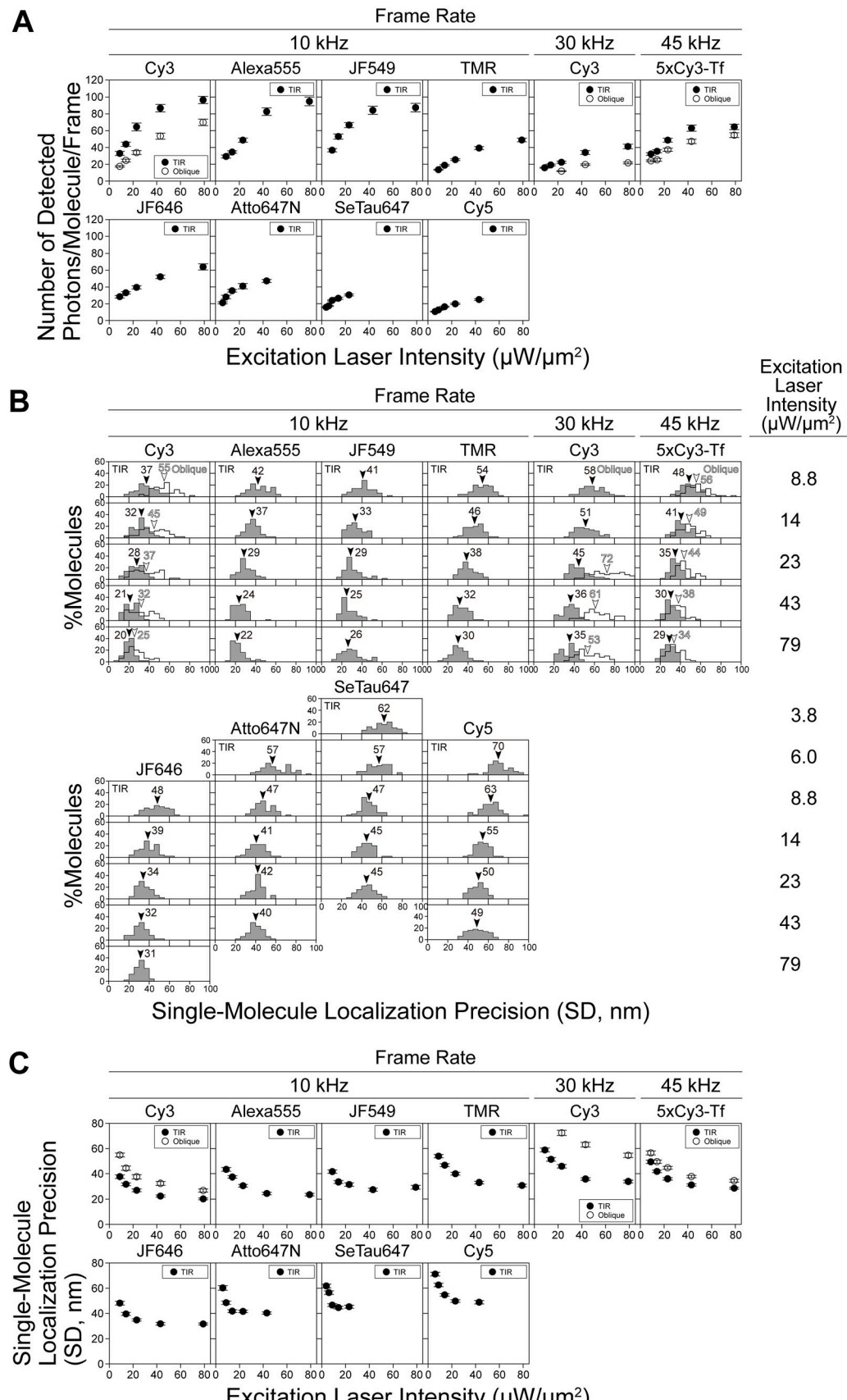

Figure S3. **The numbers of detected photons/frame from single molecules (***N***, proportional to the emitted photons during a frame time) are saturated under stronger excitation laser intensities, and thus the improvement of single-molecule localization precision (***σ$_{xy}$***) with an increase of the**

**laser intensity is limited.** For experimental details, see the caption to Fig. S2. **(A)** For various commonly used fluorescent probes, the numbers of detected photons/molecule/frame are plotted against the laser power density at the focal plane (TIR illumination; the additional examinations using oblique illuminations were only performed for Cy3 and 5xCy3-Tf), showing that Cy3 and Alexa555 are less prone to laser-power saturation. With an increase in the excitation laser intensity, the number of detected photons from single dye molecules initially increased proportionally, and then leveled off (saturation occurred), probably due to "triplet bottleneck saturation" (see "Estimation of the number of photons that can be emitted by a single Cy3 molecule during 0.1 ms: Triplet bottleneck saturation" in Materials and methods). This occurred from around 23 $\mu$W/$\mu$m$^2$ for Cy3 (the results of Cy3 at 10 and 30 kHz shown here are the same as those shown in Fig. 1 E, and are reproduced here for ease of comparison with the results of other dyes). Cy3 and Alexa555 are less prone to saturation, as compared with the other dyes tested here. In the present study, we primarily used Cy3. **(B)** Distributions of the localization precisions of single molecules of eight fluorophores observed at 10 kHz (an integration time of 0.1 ms), single Cy3 molecules at 30 kHz (0.033 ms), and 5xCy3-Tf at 45 kHz (0.022 ms), evaluated at various laser illumination intensities (provided on the right). Arrowheads indicate the median values. **(C)** For various commonly used fluorescent probes, single-molecule localization precisions (mean ± SEM) are plotted against the laser intensity (the results for Cy3 at 10 and 30 kHz shown here are the same as those shown in Fig. 1 F, and are reproduced here for ease of comparison with the results of other dyes). These results show that Cy3 and Alexa555 provide better single-molecule localization precisions at higher laser intensities, due to their lower tendency to saturate. Since Cy3 provided slightly better single-molecule localization precision at 10 kHz at saturation than Alexa555, we primarily used Cy3 throughout the remaining part of this report. Based on the results described in A and C, and also due to the versatility of the oblique-angle illumination to enable the observations of single molecules in both the basal and apical PMs, as well as in endomembranes and the cytoplasm in general (see the subsection "TIR and oblique illuminations for ultrafast SFMI" in the main text), we comprehensively tested and performed ultrafast single-molecule imaging-tracking under the oblique-angle illumination conditions, with a laser power density of 23 $\mu$W/$\mu$m$^2$ at the specimen plane. These are the "standard test conditions," using Cy3 molecules on the coverslip as the standard sample.

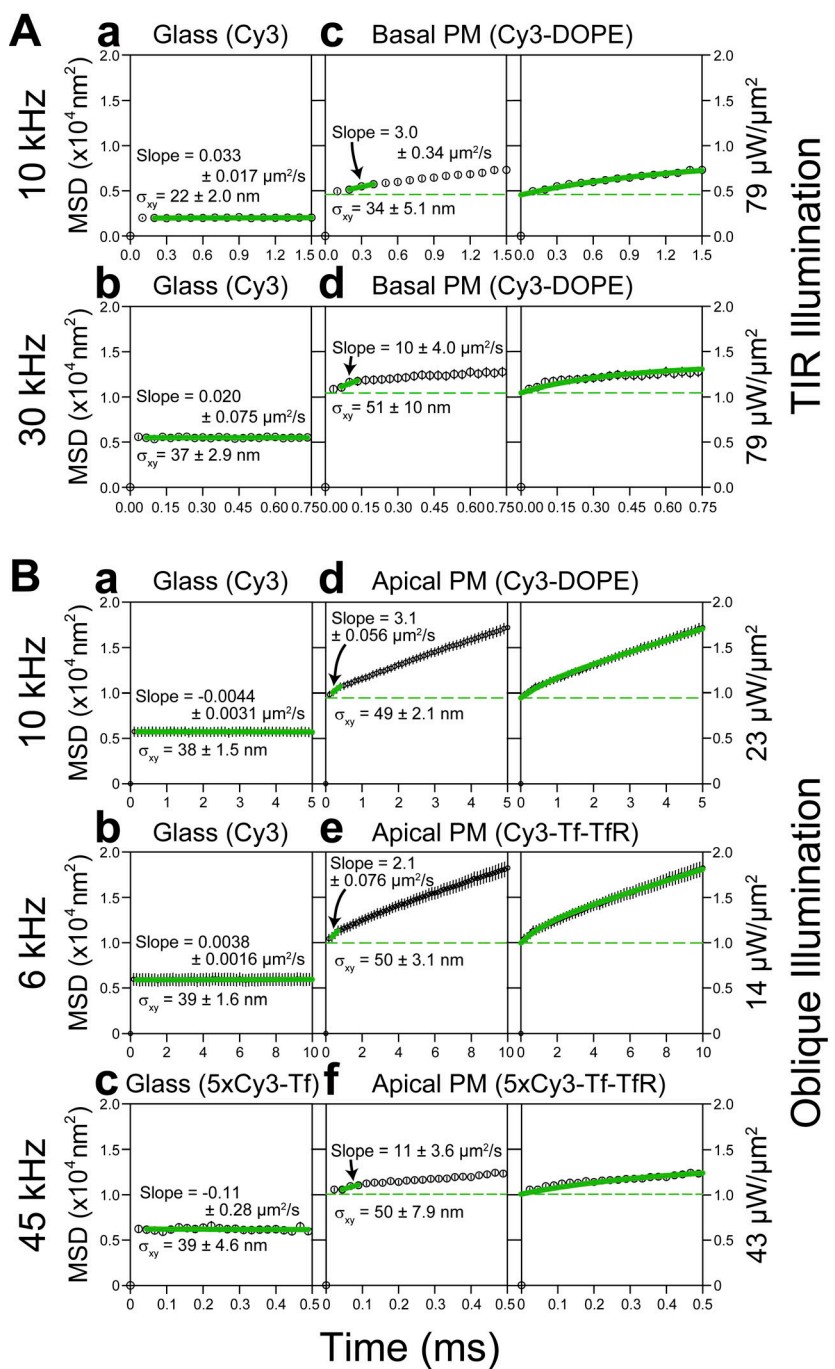

Figure S4. **MSD-Δ*t* plots ensemble averaged over all trajectories, obtained by ultrahigh-speed single-molecule imaging of Cy3 molecules immobilized on coverslips or diffusing in the apical and basal PMs of T24 cells (using oblique-angle and TIR illuminations, respectively), providing estimates of single-molecule localization errors for diffusing molecules (as well as immobile molecules).** All SEMs, including the error bars, are shown in the figure. The purposes of showing these figures are (1) to explain how to determine the single-molecule localization precisions of diffusing molecules in the PM using the MSD-Δt plot (because the method described in the caption to Fig. S2 is only useful for immobilized molecules) and (2) to show the actual localization precisions of diffusing molecules in the apical and basal PMs. First, see the panels in the left column, showing the MSD-Δt plots for molecules immobilized on the glass. Experimental MSD-Δt plots even for immobile molecules are expected to exhibit an offset, due to the position determination error; i.e., the flat MSD-Δt plot with a constant value (against Δt), which equals $4\sigma_{xy}^2$ (where $\sigma_{xy} = [\sigma_x + \sigma_y]/2$; Dietrich et al., 2002; Martin et al., 2002). The linear fitting indeed showed that the slopes were ≈0; the localization precisions determined here for Cy3 on the glass at 10 and 30 kHz were 22 and 37 nm, respectively (TIR illumination at 79 µW/µm²). These results are consistent with those for immobilized molecules determined by the first method described in Fig. S2 (based on the standard deviations of the x and y position determinations for 15 consecutive frames) and shown in Fig. S3, B and C (20 and 34 nm, respectively; all SEMs for these values are provided in the figure; also see Fig. 1 F and Table 1). Next, see the panels in the middle and right columns, showing the MSD-Δt plots for molecules undergoing diffusion in the PM. As shown in the panels in the left column (immobilized molecules), the MSD values almost reach a plateau (which is the offset value) by the second step (Δt = 0.2 and 0.066 ms for Cy3 at 10 and 30 kHz, respectively). This means that the offset value of the MSD-Δt plot for diffusing molecules in the PM can be estimated as the y-intercept (by extrapolation) of the linear-fit function for the second, third, and fourth

steps in the MSD-Δ$t$ plot (Fujiwara et al., 2002; middle column; see green keys), which is $4\sigma_{xy}^2$. Hence, the MSD-Δ$t$ plot that represents the diffusion effects can be obtained by plotting MSD − $4\sigma_{xy}^2$ against Δ$t$, as shown in Fig. 4 A. The single-molecule localization precisions determined this way (middle column) for Cy3-DOPE in the basal PM at 10 and 30 kHz were 34 and 51 nm, respectively, which were inferior to those found for the same molecules fixed on the glass (22 and 37 nm, respectively; left column). For Cy3-DOPE in the apical PM at 10 kHz, the localization precision was 49 nm, which was much worse than that in the basal PM (34 nm). The single-molecule localization precisions in the basal PM were worse than those determined on the glass, probably due to the higher background caused by cellular autofluorescence and the diffusional blurring of single-molecule spots. **(A)** The TIR laser illumination results of the ensemble-averaged MSD-Δ$t$ plots, for single Cy3 molecules covalently linked to the cover-glass surface coated with 3-aminopropyltriethoxysilane (left column) and for single Cy3-DOPE incorporated in the basal PM (middle and right columns), using the highest laser intensities of TIR illumination available for this instrument at 532 nm (79 μW/μm²). **(a–d)** This provided the best single-molecule localization precisions for recordings of Cy3 at 10 and 30 kHz, which were 22 and 37 nm on the glass (left column, a and b; $n$ = 40 and 50, respectively), and 34 and 51 nm on the basal PM (middle column, c and d; $n$ = 50 and 50, respectively). In the panels in the right column, the green curves are the best-fit functions describing the MSD-Δ$t$ plots for the confined-diffusion model, in which molecules undergo free diffusion while totally confined within a limited area during the observation period (Eqs. 11–13 in Kusumi et al., 1993). Since the observation durations for single molecules (1.5 and 0.75 ms full x-axis scales) were shorter as compared with the dwell time of Cy3-DOPE within a compartment, the confined fitting, rather than the hop-diffusion fitting, was employed. **(B)** The oblique-angle laser illumination results of the ensemble-averaged MSD-Δ$t$ plots, for single Cy3 molecules and 5xCy3-Tf on the coverslip (left column) and for single Cy3-DOPE, Cy3-Tf (with a dye-to-protein molar ratio of 0.2, so that virtually all of the single Tf molecules are labeled with either 0 or 1 Cy3 molecule) bound to TfR, and 5xCy3-Tf bound to TfR in the apical PM (middle and right columns). The oblique-angle laser illumination is widely applicable and useful because it can illuminate molecules located deeper in the cytoplasm as well as those present in the apical PM. Therefore, it is extensively used in the present research (standard conditions using an oblique-angle laser illumination power density of 23 μW/μm²). The numbers of examined spots: a and b, $n$ = 17; d and e, $n$ = 50; c, $n$ = 40; f, $n$ = 150 (a–c and d–f are on glass and on the apical PM, respectively). The illumination laser power densities were selected so that they were just beneath the level where dye saturation is obvious, and the single-molecule localization errors for various Cy3 specimens observed at different frame rates were similar to each other (see Fig. S3, B and C). More specifically, Cy3 and Cy3-DOPE at 10 kHz and 23 μW/μm² (B a and d), Cy3 and Cy3-Tf at 6 kHz and 14 (23 × [6/10]) μW/μm² (B b and e; because the frame time is [10/6]-times longer at 6 kHz), and 5xCy3-Tf at 45 kHz at 43 μW/μm² (B c and f). Panels in the left column (molecules on the glass): The localization precisions determined here for Cy3 at 10 kHz (38 nm at 23 μW/μm²) and for 5xCy3-Tf at 45 kHz (39 nm at 43 μW/μm²) were consistent with those found in Fig. S3 B (37 and 38 nm, respectively). Panels in the middle and right columns (molecules in/on the apical PM): The single-molecule localization precisions determined for Cy3-DOPE (10 kHz, 23 μW/μm²), Cy3-Tf (6 kHz, 14 μW/μm²), and 5xCy3-Tf (45 kHz, 43 μW/μm²) in/on the apical PM were determined to be 49, 50, and 50 nm, respectively, which were inferior to those found for the same molecules fixed on the glass (38, 39, and 39 nm, respectively; left column). The localization errors were greater in the apical PM, probably due to the higher background caused by cellular autofluorescence and the blurring of single-molecule spots due to molecular diffusion within a frame time. In the right column in d and e, the green curves are the best-fit functions describing the MSD-Δ$t$ plots for an idealized hop-diffusion model (hop-diffusion fitting; see Supplemental theory 2 in the Supplemental text). In f, since the observation duration for single 5xCy3-Tf molecules employed here (0.5 ms full x-axis scale) was shorter as compared with the dwell time of TfR within a compartment, the confined fitting, rather than hop-diffusion fitting, was employed.

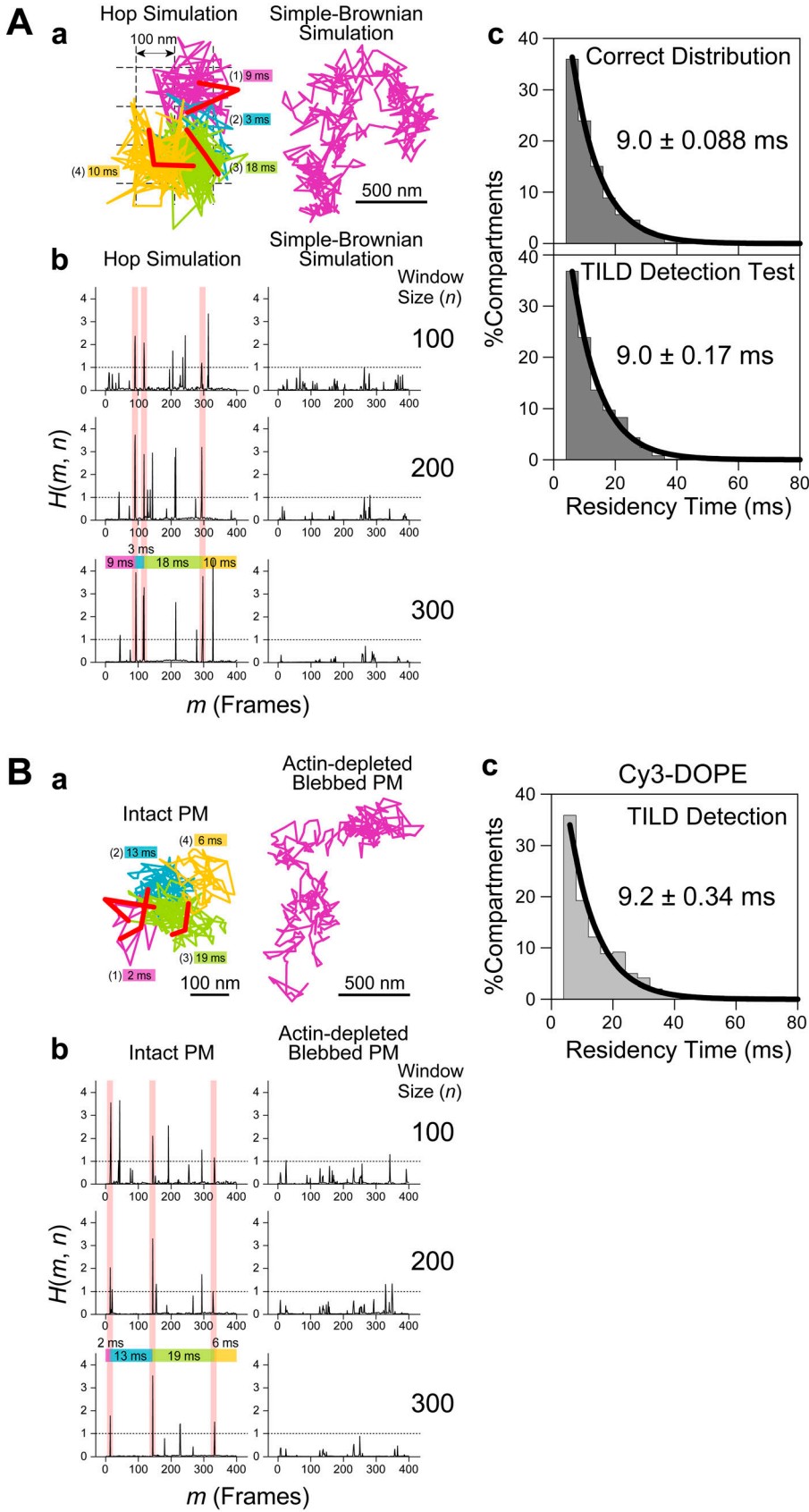

Figure S5. **The TILD method for detecting the moment (instance) when a diffusing molecule in the PM undergoes the hop movement from a compartment to an adjacent one in the PM.** We developed an improved method for detecting the hop moment (instance). This method detects the Transient Increase of the effective Local Diffusion (TILD) in a single-molecule trajectory. TILDs are likely to occur when a molecule hops between two membrane

compartments, but our analysis itself remains model-independent of any particular model. During an intercompartmental hop, a molecule experiences two compartments instead of one within the time window that includes the hop instance, thus leading to an increase in the effective local diffusion coefficient. To identify TILDs in a given trajectory, consider a window with a size of $n$ frames ($n$ steps), starting from a frame $m$ ($m \sim m + n$). Within this window, the center of the $n$ recorded coordinates ($n$ points) is defined, the radial displacements for all of these $n$ points from this center are calculated, and then their maximal value $R_{MAX}(m,n)$ is determined. Next, the relative diffusion coefficient for this window is defined as $D_{rel}(m,n) \equiv \frac{\frac{R_{MAX}^2(m,n)}{4n\delta t}}{D_{2-4}}$, where $\delta t$ is the time step in the trajectory (inverse of the frame rate), and $D_{2-4}$ is the average short-term diffusion coefficient determined for the time interval between $2\delta t$ and $4\delta t$ (averaged over the whole trajectory; Kusumi et al., 1993), which is included for normalization. For free Brownian diffusion, $D_{rel}(m,n)$ is $\approx 1$ (allowing for statistical variations), independent of $m$ or $n$. If a molecule is temporarily trapped in a finite domain, then as the window size $n$ increases, $D_{rel}(m,n)$ decreases due to entrapment. When the window size has increased sufficiently to include the release of the diffusing particle from the finite domain, there will be a sharp increase in $D_{rel}(m,n)$, due to the extended range of diffusion. To flag releases, the function $H(m,n) = |1/D_{rel}(m + 1, n) - 1/D_{rel}(m, n)|$ was employed, and by scanning all possible $m$ and $n$ pairs in the full trajectory, a map of $H(m,n)$ was produced. Releases from the trapped domain are flagged by sharp peaks in $H(m,n)$ for both the special starting position (e.g., if position $m$ is before a hop and position $m+1$ is after a hop, then for all window sizes $n$, $D_{rel}(m,n)$ will be greater than $D_{rel}(m+1,n)$) and for the combination of a starting position and a window size (e.g., if the trajectory, starting from position $m$ with a window size $n$, and ending at point $p = m + n$, is wholly within an entrapped domain, and if extending the window size by 1 includes release from the entrapped domain, then $D_{rel}(m,n+1)$ will be greater than $D_{rel}(m,n)$ for all $m$ and $n$ values, such that $m + n = p$). The minimum number of points within an entrapped domain required to allow the detection of the moments of TILDs is determined by the stochastic nature of diffusion (+ the diffusion coefficient within the domain and the domain size) and the noise present in the position determination. As such, due caution was exercised to avoid choosing a plethora of small compartments erroneously. Typically, the total length of each trajectory analyzed here was 1,000 steps. The $n$ value was varied from 21 to $N-m$, where $N$ is the total number of steps in a trajectory (= 1,000). In addition to this anterograde direction of analysis over a single trajectory, the same trajectory was analyzed in the retrograde direction in the same way as for the anterograde direction, in order to determine whether the sudden increase in $H(m,n)$ can also be observed for the same $m$ in the anterograde analysis. Therefore, for a single $m$ value, $H(m,n)$ was calculated for $[N-40]$ windows ($[N-m-20] + [m-20]$). Here, we first describe the results of testing the TILD method using Monte-Carlo simulated hop and simple-Brownian trajectories with a single-molecule localization precision of 50 nm (error for Cy3 in the apical PM, which is the worst precision in the present report; A). Second, we describe the application of the TILD method to detect the hop moments in single-molecule trajectories of Cy3-DOPE, observed in the intact apical PM as well as in the actin-depleted blebbed PM (B). **(A)** Testing the TILD detection method using computer-generated hop and simple-Brownian diffusion trajectories, and examination of the exponential distributions of the residency times within a compartment. **(A a)** Typical hop-diffusion (left) and simple-Brownian (right) trajectories generated by Monte Carlo simulation (every 0.1 ms; the initial 400-steps of the 1,000-step-long trajectories are shown here, whereas the TILD analysis was performed for the full 1,000 steps for all of the trajectories). The moments of TILDs determined by the developed protocol are shown by the thick, short red subtrajectories (three-frame trajectories defined by the TILD moment ±1 frame; note their tilde-like shapes). This particular hop-diffusion trajectory (400-frames long) on the left contained three TILDs. The simple-Brownian trajectory (400-frames long) on the right exhibits no TILD. Hop-diffusion trajectories were generated as described previously (Ritchie et al., 2005), except that the unit time step was 0.1 μs. Using a two-dimensional square array of partially permeable barriers separated by 100 nm ($L$), with a probability of transmission per attempt of 0.0005, the experimental $D_{MACRO}$ for Cy3-DOPE in the intact apical PM (0.30 μm²/s; Table 2) was reproduced ($D_{micro}$ was set at 9 μm²/s, as shown in Fujiwara et al., 2002). A single-molecule localization error of 0, 25, or 50 nm was added as the Gaussian noise to each x- and y-coordinate in the hop trajectories (here, the trajectories including a 50-nm localization error are shown). Among 100 trajectories generated by the simulation (1,000 frames per trajectory), 97, 94, and 80 trajectories were statistically classified into the suppressed diffusion mode in the presence of single-molecule localization errors of 0, 25, and 50 nm, respectively. Test trajectories of simple-Brownian particles were generated by Monte Carlo simulations, using a diffusion coefficient of 6 μm²/s, as experimentally obtained in blebbed PMs (Table 2; employing 9 μm²/s made virtually no difference), and Gaussian localization errors were added. **(A b)** Typical plots of $H(m,n)$ vs. $m$ (only the results with window sizes of $n = 100$, 200, and 300 for the trajectories shown in A a. The sharp changes (peaks) in $H(m,n)$ are likely to represent the hop movements (transient increase of local diffusion coefficient), and spurious peaks from statistical variations and noise can mostly be distinguished because they only appear in the displays for limited numbers of windows. Briefly, for the same frame number $m$, $H(m,n)$ was calculated for all possible $n$'s ($N-40$ windows), and when the percentage of windows in which $H(m,n) \geq 1$ was >20% among a total of $[N-40]$ windows, the molecule was regarded as undergoing the process of intercompartmental hopping. In this figure, the peaks that satisfied these thresholds are highlighted by vertical pink bars, indicating the occurrences of TILDs, i.e., hop events. Other peaks in this display did not satisfy the threshold conditions explained here, and do not represent TILDs. In the application of the TILD detection method to individual trajectories, a three-frame running average (replacing the position of the $k$th frame with the position averaged for the $k-1$, $k$, and $k+1$ frames) was first applied to each trajectory, to minimize the effect of apparently large displacements that stochastically occurred due to the 50-nm single-molecule localization error. The detectability percentages of hop events (given by the simulation program) for simulated trajectories classified into suppressed diffusion were 82, 76, and 66%; and the accuracies of the predicted hop events were 75, 66, and 60% at localization errors of 0, 25, and 50 nm, respectively. In our standard conditions for long-term single-molecule tracking experiments (1,000 steps; 0.1-ms resolution; a 532-nm laser excitation laser power of 23 μW/μm² at the sample), the single-molecule localization error was 49 nm in the apical PM. **(A c)** Distributions of the residency lifetimes within a compartment for Monte-Carlo simulated test particles undergoing hop diffusion. The residency time of a particle within a compartment was obtained as the duration between two consecutive TILDs. Top (Ground Truth Distribution Given by the Simulation): The correct distribution of the residency times determined from the hop events (given by the simulation program), which occurred in 100 simulated 1,000-frame long hop trajectories. The residency times shorter than 40 frames (4 ms) were neglected, due to various uncertainties in the short time ranges. The distribution could be fitted with a single exponential function with a decay time constant of 9.0 ± 0.088 ms (mean ± SEM; SEM is provided as the fitting error of the 68.3% confidence interval). The exponential distribution of the residency times found for simulated hop-diffusion trajectories can actually be predicted theoretically, as summarized in Supplemental theory 1 in the Supplemental text. The theory also predicts that its decay time constant can be described by $L^2/4D_{MACRO}$. In the present simulation, the average dwell time calculated using this equation was 8.3 ms ($L = 100$ nm, the average $D_{MACRO}$ obtained from the simulation was 0.3 μm²/s). This value agrees quite well with the dwell lifetimes obtained by simulated hop-diffusion trajectories (9.0 ± 0.088 ms). Bottom (TILD Detection Test): The residency time distribution, determined by the TILD-detection method from 100 simulated 1,000-frame long hop trajectories that included a single-molecule localization precision of 50 nm (residencies in 763 compartments with durations longer than 4 ms). The decay time constant was 9.0 ± 0.17 ms. This agrees well with the correct distribution, suggesting that the developed protocol is useful for evaluating the residency lifetime, although at the level of individual hops, our software misses hops (66% detectability) and incorrectly detects hops (60% accuracy). The number of detected TILDs per 1,000-frame simulated simple-Brownian trajectories, using a diffusion coefficient of 6 μm²/s and a localization precision of 50 nm, was only 0.35/trajectory ($n = 100$ trajectories; 0.33 when a diffusion coefficient of 9 μm²/s was assumed). **(B)** Detecting TILDs in Cy3-DOPE and TfR trajectories obtained in intact and actin-depleted blebbed PMs. **(B a)** Typical Cy3-DOPE trajectories (0.1-ms resolution) in the intact apical PM (left, the 40-ms-long initial part of the trajectory shown in Fig. 3 C) and in the actin-depleted blebbed apical PM (right; typical among 50 and 20 trajectories, respectively). The moments of TILDs, detected as shown in

B b, are shown by the thick, red three-step subtrajectories. The trajectory obtained in the intact apical PM contained three TILDs, whereas that in the actin-depleted blebbed apical PM exhibited no TILDs. In the trajectory obtained in the intact apical PM (left), the colors of the trajectories are changed across the short red TILD trajectory, and this convention was used throughout this report. **(B b)** The plots of $H(m,n)$ vs. $m$ for the trajectories shown in B a. TILDs were detected in all of the experimental trajectories of Cy3-DOPE and TfR obtained in the intact apical PM (see Fig. 3 D; 50 trajectories with a length of 1,000 frames for Cy3-DOPE at a frame rate of 10 kHz and for TfR at a frame rate of 6 kHz). The average numbers of detected TILDs (with intervals longer than 2 ms) per 1,000-frame long trajectory classified into the suppressed-diffusion mode were 7.8 (= 297 events/38 trajectories) for Cy3-DOPE and 4.4 (= 175 events/40 trajectories) for TfR. Meanwhile, in the actin-depleted blebbed PM, where more than 90% of the trajectories were statistically classified into the simple-Brownian diffusion mode (20 trajectories were examined for both Cy3-DOPE and TfR; Table 2), the numbers of detected TILDs per 1,000-frame long trajectory classified into the simple-Brownian diffusion mode were only 0.22 (= 4 events/18 trajectories) for Cy3-DOPE and 0.68 (= 13 events/19 trajectories) for TfR. **(B c)** Distributions of the dwell times within a compartment for Cy3-DOPE (50 trajectories; 337 residencies) in the intact apical PM, with the best-fit exponential curves and dwell lifetimes of 9.2 ± 0.34 ms. The exponential shape of this distribution is consistent with the hop-diffusion model (under strong-type confinements; see Supplemental theory 1). Furthermore, the exponential residency lifetime found for Cy3-DOPE (9.2 ms) agrees well with that found for the simulated hop-diffusion trajectories, using parameters similar to the experimentally determined values for Cy3-DOPE (9.0 ms; A c, top).

Video 1. **Single Cy3 molecules covalently linked to the glass surface, observed at 10 kHz (every 0.1 ms; observation period of 1,500 frames = 150 ms; Replay, 167×-slowed from real time) using oblique-angle illumination at 23 µW/µm².** a, b, and c represent the fluorescent spots corresponding to those in Fig. 2, A and B. The on-periods including short off-periods of up to three frames are highlighted by the orange circle, and the off-periods are highlighted by the cyan circle.

Video 2. **Single Cy3-DOPE molecules diffusing in the intact apical PM, observed at 10 kHz (every 0.1 ms; observation period of 750 frames = 75 ms; Replay, 167×-slowed from real time) using oblique-angle illumination at 23 µW/µm².** Most fluorescent spots were photobleached within 100 frames after starting the illumination, but a small percentage of the spots lasted longer than 300 frames. The long-lasting spot enclosed within the yellow square in the movie on the left is enlarged and highlighted by the circle in the movie on the right.

Video 3. **Single Cy3-DOPE molecules diffusing in the intact apical PM and the actin-depleted blebbed PM (left and right respectively), observed at 10 kHz (every 0.1 ms; see Fig. 3, C and D; observation periods of 750 and 300 frames = 75 and 30 ms, respectively; Replay, 167×-slowed from real time).** The statistical analysis method developed in this study classified the trajectories into the suppressed-diffusion and simple-Brownian diffusion modes, respectively. The flickering signals seen in the movies, in addition to the long-lasting spots (indicated by the trajectories), are probably due to the fluctuation of the autofluorescence signal from the cytoplasm, because they are also found in the movies of Cy3-Tf (Video 4).

Video 4. **Single TfR molecules, bound by Cy3-Tf, diffusing in the intact apical PM or the actin-depleted blebbed PM (left and right, respectively), observed at 6 kHz (every 0.167 ms; observation periods of 600 and 180 frames = 100 and 30 ms, respectively; Replay, 167×-slowed from real time; see Fig. 3 D).** The statistical analysis method developed in this study classified the trajectories into the suppressed-diffusion and simple-Brownian diffusion modes, respectively.

Video 5. **Single TfR (Halo-TMR) molecules diffusing in the basal PM, observed at 6 kHz (every 0.167 ms; observation period of 1,500 frames = 250 ms; Replay, 50×-slowed from real time; see Fig. 6 A).** Movies on the top (bottom) show the movement of a single TfR molecule outside (inside) the focal adhesion regions. Movies on the left (larger view-fields): green areas, focal adhesion regions marked by mGFP-paxillin; magenta spots, single TfR molecules. The regions in the yellow squares are enlarged in the movies on the right, showing single-molecule trajectories outside (top) or inside (right) the focal adhesion zone.

Video 6. **Single EGFR (Halo-TMR) molecules diffusing in the intact basal PM before (left) and 2.5 min after (right) the addition of 10 nM EGF, observed at 6 kHz (every 0.167 ms; observation period of 1,500 frames = 250 ms; Replay, 50×-slowed from real time; see Fig. 7 and its legend).** The statistical analysis method developed here classified both trajectories into the suppressed-diffusion mode. Engaged EGFR molecules hopped less frequently, as compared with non-ligated EGFR, probably due to dimerization/oligomerization (oligomerization-induced trapping).

### Supplemental theory 1. Expected distribution of the residency times: Development of the hop-diffusion theory

In the hop-diffusion literature, the average residency time within a compartment has been intuitively approximated as $L^2/4D_{MACRO}$. However, this expression has never been proven rigorously. In addition, the distribution of the residency times has remained

unknown even for an idealized hop-diffusion model, in which molecules undergo free diffusion (in viscous media) in the presence of equally spaced, equi-potential semi-permeable diffusion barriers. Here, we demonstrate that the residency time distribution is given by the sum of exponential distributions, which can be approximated well by a single exponential function with a decay constant of $L^2/4D_{MACRO}$, if the confinement effect is strong.

Consider a Brownian particle in a square region with semi-permeable boundaries, and calculate the distribution of first exit times. The residency time distribution is analogous to the distribution of the first time a particle exits a certain region. We assume that diffusion along the $x$ and $y$ directions is independent, which allows us to solve the problem for a one-dimensional (1D) system, and then generalize the results to the two-dimensional (2D) case.

Consider a Brownian particle in 1D, initially placed at $x = x_0$ between two boundaries located at $x = 0$ and $x = L$, that can diffuse freely in this bounded region. When the particle attempts to leave this region, it faces resistance, and thus the boundaries can be considered as partially permeable barriers. Once the particle crosses the boundary, it can never go back. Therefore, the time when it leaves the region is always the first exit time. The probability distribution of such a particle is governed by the diffusion equation,

$$\frac{\partial \rho}{\partial t} = D\frac{\partial^2 \rho}{\partial x^2}, \quad (13)$$

with the boundary conditions,

$$-D\frac{\partial \rho(x,t)}{\partial x}\bigg|_{x=0} = -p\rho(0,t),$$

$$-D\frac{\partial \rho(x,t)}{\partial x}\bigg|_{x=L} = p\rho(L,t), \quad (14)$$

where $D$ is the microscopic diffusion coefficient within a domain, and $p$ is a constant related to the permeability of the boundaries, such that $p\to 0$ and $p\to\infty$ correspond to impenetrable and completely permeable boundaries, respectively. The solution can easily be obtained using separation of variables (Carslaw and Jaeger, 1986), and is given in the following form:

$$\rho(x,t) = \sum_{n=1}^{\infty} \beta_n(x_0)e^{-\lambda_n^2 Dt/L^2}\phi_n(x),$$

$$\phi_n(x) = \gamma_n(\alpha\lambda_n\cos(\lambda_n x/L) + \sin(\lambda_n x/L)),$$

$$\gamma_n = \sqrt{\frac{2}{L}}\left((1 + \alpha + \alpha^2\lambda_n^2) + \alpha\left[\frac{\alpha^2\lambda_n^2 + 1}{\alpha^2\lambda_n^2 - 1}\right]^2\cos^2\lambda_n\right)^{-1/2}, \quad (15)$$

where $\beta_n(x_0)$'s are determined by the initial conditions, $\alpha = D/pL$, and $\lambda_n$'s, $n = 1,2,3,...$, are the positive solutions of

$$\tan\lambda_n = -\frac{2\alpha\lambda_n}{1 - (\alpha\lambda_n)^2}. \quad (16)$$

If the particle is initially at $x = x_0$ such that $\rho(x,t) = \delta(x-x_0)$, then the coefficient $\beta_n$ becomes

$$\beta_n = \phi_n(x_0) \quad (17)$$

so that

$$\rho(x,t) = \sum_{n=1}^{\infty} e^{-\lambda_n^2 Dt/L^2}\phi_n(x_0)\phi_n(x). \quad (18)$$

The cumulative probability distribution can be expressed as

$$\rho_{cdf}(x,t) = \int_0^x dy\rho(y,t),$$

$$= \sum_{n=1}^{\infty} e^{-\lambda_n^2 Dt/L^2}\frac{\gamma_n L}{\lambda_n}(1 - [\cos(\lambda_n x/L) - \alpha\lambda_n\sin(\lambda_n x/L)])\phi_n(x_0). \quad (19)$$

Finally, the distribution of exit times $f_{ex}(t)$, and its cumulative $F_{ex}(t)$, can be calculated from the cumulative probability distribution above (Redner, 2001):

$$f_{ex}(t) = -\frac{d}{dt}\rho_{cdf}(L,t),$$

$$F_{ex}(t) = \int_0^t ds f_{ex}(s) = 1 - \rho_{cdf}(L,t). \quad (20)$$

Now we need to generalize this result to two dimensions and obtain the distribution of exit times from a square compartment of area $L^2$. This was readily achieved by realizing that the time of the first exit is the minimum of the first exit time in the $x$ and $y$ directions. Using the well-known result for the distribution of the minimum of two independent random variables (Gumbel, 2004), we obtain the cumulative probability distribution for the exit time distribution as

$$F_{ex}(t) = F_{ex,x}(t) + F_{ex,y}(t) - F_{ex,x}(t)F_{ex,y}(t), \quad (21)$$

where $F_{ex,w}(t)$ stands for the probability that the particle exits through one of the boundaries along the $w$ direction until time $t$, and is explicitly given by

$$F_{ex,w}(t) = 1 - \sum_{n=1}^{\infty} e^{-\frac{\lambda_n^2 Dt}{L^2}} \frac{\gamma_n L}{\lambda_n} \phi_n(w_0)\left(1 - [\cos\lambda_n - \alpha\lambda_n\sin\lambda_n]\right),$$

where $w_0$ is the initial position along the $w$ axis. Usually, the initial conditions are not accessible, so it is reasonable to average all of the different initial states. By averaging $F_{ex,w}(t)$ over all initial positions $w_0$ between 0 and $L$ with equal weight, we obtain

$$\bar{F}_{ex,w}(t) = 1 - \sum_{n=1}^{\infty} e^{-\frac{\lambda_n^2 Dt}{L^2}} \xi_n,$$

$$\xi_n = \frac{\gamma_n^2 L}{\lambda_n^2}\left(1 - [\cos\lambda_n - \alpha\lambda_n\sin\lambda_n]\right)^2. \quad (23)$$

Using Eq. 23, Eq. 21 further simplifies to

$$\bar{F}_{ex}(t) = \bar{F}_{ex,w}(t)\left(2 - \bar{F}_{ex,w}(t)\right),$$

$$\bar{F}_{ex}(t) = 1 - \left[\sum_{n=1}^{\infty} e^{-\frac{\lambda_n^2 Dt}{L^2}} \xi_n\right]^2. \quad (24)$$

as there is no difference in the statistics of the position between the $x$ and $y$ directions after averaging over the initial position. The probability distribution for the exit times, or the residency time distribution, is given by the first derivative of Eq. 24 with respect to time:

$$\bar{f}_{ex}(t) = 2\sum_{n=1}^{\infty}\sum_{m=1}^{\infty} e^{-(\eta_n + \eta_m)t}\xi_n\xi_m\eta_m, \quad (25)$$

where $\eta_n = \lambda_n^2 D/L^2$.

The contributions of the higher order terms in Eq. 25 are quickly minimized, due to the exponential term. In many cases of practical interest, the first term with $n = 1, m = 1$ alone could be a good approximation of the result. To assess the validity of this argument, let us consider the ratio of the exponential factors in the first two terms of the double summation in Eq. 25. This ratio is equal to

$$e^{-\left(\lambda_2^2 - \lambda_1^2\right)Dt/L^2}. \quad (26)$$

Inspecting the equation for $\lambda_n$s, Eq. 16, we notice that $\lambda_{n+1} \approx \lambda_n + \pi$, such that

$$e^{-\left(\lambda_2^2 - \lambda_1^2\right)Dt/L^2} \approx \left(e^{-\frac{Dt}{L^2}}\right)^{\pi^2 + 2\lambda_1\pi}. \quad (27)$$

Therefore, the higher order terms decay with almost the 10th or greater powers of $e^{-Dt/L^2}$, which are always <1 for $t > 0$. Similarly, it can be shown that the coefficient of the exponential, $\xi_n\xi_m\eta_m$, also decreases as $n$ and $m$ increase.

When the confinement effect is strong, the Brownian particle spends a sufficient amount of time in each compartment to cover it uniformly before escaping. Mathematically, this corresponds to the limit

$$\alpha \gg 1, \quad (28)$$

in which $\xi_n$ is approximately equal to 1. Thus, the residency time distribution given in Eq. 25 can be approximated by a single exponential

$$\bar{f}_{ex}(t) \approx \frac{1}{\tau}e^{-\frac{t}{\tau}}, \quad (29)$$

$$\bar{\tau} = \frac{L^2}{2D\lambda_1^2}. \quad (30)$$

As the confinement effect becomes stronger, we would expect the average residency time to diverge. Therefore, in the strong confinement limit, $\lambda_1$ would be close to 0 such that $tan\lambda_1 \approx \lambda_1$, and Eq. 16 can be replaced by an approximate form

$$\lambda_1 = -\frac{2\alpha\lambda_1}{1 - (a\lambda_1)^2}, \quad (31)$$

which gives

$$\lambda_1^2 = \frac{2\alpha + 1}{\alpha^2}. \quad (32)$$

Therefore, the average residency time is equal to

$$\bar{\tau} = \frac{L^2}{4D\left(\frac{1}{\alpha} + \frac{1}{2\alpha^2}\right)}. \quad (33)$$

We now need to express $\alpha$ in terms of a quantity that can be experimentally measured. In the strong confinement limit that we are interested in, the Brownian particle behaves much like a random walker in a 2D lattice with lattice spacing $L$, for durations longer than the average residency time. In this picture, each lattice site corresponds to a square compartment of area $L^2$, and the random walker takes steps between adjacent lattice sites at a rate $F = \frac{1}{4\bar{\tau}}$. As the random walker can move in four directions in 2D, the rate of escape is $4F$, such that the average residency time is $\bar{\tau}$. This description is appropriate as long as the random walker explores most of each compartment before it leaves. This ensures that the probability distribution within each compartment quickly becomes uniform and that the escape time distribution can be well approximated by a single exponential. The probability of finding the random walker at a lattice site $(m,n)$ is governed by a Master equation (Hughes, 1995):

$$\frac{dP_{m,n}(t)}{dt} = FP_{m+1,n}(t) + FP_{m-1,n}(t) + FP_{m,n+1}(t) + FP_{m,n-1}(t) - 4FP_{m,n}(t). \quad (34)$$

With a straightforward calculation, the mean square displacement of a random walker that is initially at the origin is given by

$$\langle r^2 \rangle = \sum_{m,n} L^2 P_{m,n}(t)(m^2 + n^2) = 4\frac{L^2}{4\bar{\tau}}t. \quad (35)$$

As this description is appropriate for times longer than $\bar{\tau}$, the diffusion coefficient associated with this random walk is the macroscopic diffusion coefficient,

$$\langle r^2 \rangle \approx 4D_{MACRO}t. \quad (36)$$

Combining Eqs. 33, 35, and 36, we obtain

$$\frac{D_{MACRO}}{D} \approx \frac{1}{\alpha} + \frac{1}{2\alpha^2}. \quad (37)$$

From many published reports (see the Table 1 summary in Kusumi et al., 2005), generally

$$D_{MACRO}/D \sim 1/20. \quad (38)$$

By solving Eq. 37 using this value,

$$\alpha \tilde{\ } 20.5. \quad (39)$$

This value satisfies the condition for strong confinement, consistent with Eq. 28. Accordingly, under the strong confinement conditions, the second order term proportional to $\alpha^{-2}$ in Eq. 33 can be neglected, and Eq. 33 can simply be expressed as

$$\bar{\tau} = \frac{L^2}{4D_{MACRO}}. \quad (40)$$

Therefore, the decay time constant of the exponential distribution of the residency time within a compartment, given by Eq. 33, can be simplified to the expression given by Eq. 40.

We obtained the distribution of the dwell lifetimes in a compartment by directly measuring the dwell time using the TILD analysis (Fig. S5, A c and B c). In the real PM, the dwell time of diffusing molecules within a compartment would be affected by the

variations in the compartment sizes and shapes and in the properties of the compartment boundaries, due to the differences in actin binding proteins and TM picket proteins. However, as shown in Fig. S5 B c and Fig. 5 A, the residency time distribution in the PM is well approximated by an exponential function, probably because these variations both lengthen and shorten the dwell lifetimes quite randomly.

### Supplemental theory 2. Hop-diffusion fitting: The function describing the MSD-Δt plot for particles undergoing hop diffusion

We developed an equation for the MSD-Δt plot for particles undergoing idealized hop diffusion; i.e., free diffusion in the presence of equally spaced, equi-potential semi-permeable diffusion barriers. Such hop diffusion can be characterized by the following three parameters: the compartment size (the distance between barriers), $L$, the microscopic diffusion coefficient within a compartment (true diffusion coefficient in the absence of the compartments), $D_{micro}$, and the long-term diffusion coefficient over many compartments, $D_{MACRO}$. One of the key parameters for hop diffusion, the residency time within a compartment, can be calculated from these parameters as $L^2/4D_{MACRO}$, as shown in Supplemental theory 1.

By employing the notations $\gamma = D_{MACRO}/(D_{micro}-D_{MACRO})$ and $\tau = 4D_{micro}t/L^2$, the 1-dimensional MSD in the real time domain, which is the inverse Laplace transform of Eq. 18 in Kenkre et al. (2008) for a 1-dimensional MSD averaged over all initial locations, can be obtained as

$$MSD(\tau) = \frac{L^2}{2}\left\{\tau - \frac{1}{\gamma}[\tau - res_0(\tau) - 2res(\tau)]\right\}, \quad (41)$$

where the terms $res_0(\tau)$ and $res(\tau)$ are the sum of the residues that arise from the inverse Laplace transform. The zeroth residue, $res_0(\tau)$, is expressed as

$$res_0(\tau) = \frac{3\tau\gamma + \gamma + 3\tau}{3(\gamma + 1)^2}. \quad (42)$$

whereas the term $res(\tau)$, representing half the sum of all other residues, is expressed as

$$res(\tau) = -\sum_{k=2}^{N} \frac{e^{-\tau r^2(k)}}{(\gamma\tan^2(r(k)) + \gamma + 1)r^2(k)}, \quad (43)$$

where $r(k)$ ($k = 2,3,4,...$) is the $k$ th root of $-\frac{1}{\gamma}y = \tan(y)$, for $y \geq 0$. As performed in Kenkre et al. (2008), we employed the bisection method to find the roots numerically with high precisions and obtained the accurate result for a sufficiently large $N$. Namely, by rearranging terms in Eq. 43, we obtain the following equation.

$$MSD(\tau) = \frac{L^2\tau\gamma}{2(\gamma + 1)} + \frac{L^2}{6(\gamma + 1)^2} - \frac{L^2}{\gamma}\sum_{k=2}^{N} \frac{e^{-\tau r^2(k)}}{(\gamma\tan^2(r(k)) + \gamma + 1)r^2(k)}. \quad (44)$$

We used this equation for fitting the experimental MSD values.

As for the physical meaning, the first term characterizes the long-term behavior of MSD, which is approximately equal to $2D_{MACRO}t$ for strong confinement ($\gamma \approx 0$) and $2D_{micro}t$ for weak confinement ($\gamma \to\infty$). The second and third terms describe the transient behavior of the MSD. At around $\tau \approx 0$, the second and third terms cancel each other out. In the longer time limit, the third term becomes negligible, and the second term becomes equal to the intercept of the linear MSD.

