## [Peer Review File · The Journal of Cell Biology]

Development of ultrafast camera-based single fluorescent-molecule imaging for cell biology

Takahiro Fujiwara, Shinji Takeuchi, Ziya Kalay, Yosuke Nagai, Taka Tsunoyama, Thomas Kalkbrenner, Kokoro Iwasawa, Ken Ritchie, Kenichi Suzuki, and Akihiro Kusumi

Corresponding Author(s): Akihiro Kusumi, Okinawa Institute of Science and Technology Graduate University

Review Timeline:

Submission Date:	2021-10-28
Editorial Decision:	2022-01-05
Revision Received:	2023-03-25
Editorial Decision:	2023-04-27
Revision Received:	2023-05-02
Accepted:	2023-05-04

Monitoring Editor: Joerg Bewersdorf

Scientific Editor: Andrea Marat

Transaction Report:

DOI: <https://doi.org/10.1083/jcb.202110160>

January 5, 2022

Re: JCB manuscript #202110160

Prof. Akihiro Kusumi
Okinawa Institute of Science and Technology
Membrane Cooperativity Unit
Onna-son
Okinawa 904-0495
Japan

Dear Prof. Kusumi,

Thank you for submitting your manuscript entitled "Development of ultrafast camera-based imaging of single fluorescent molecules and live-cell PALM". The manuscript was assessed by expert reviewers, whose comments are appended to this letter. We invite you to submit a revision if you can address the reviewers' key concerns, as outlined here.

You will see that while the reviewers express potential enthusiasm about the technical advance, there are concerns regarding the general applicability of the technique to the broader community. As a publication requirement for JCB Tools is that they must describe a new method of immediate value and broad utility to the cell biology community, this concern must be alleviated. If you decide to move some of your data from the Tools paper to the Article paper as suggested, please keep in mind that Tools are also expected to include novel biological insight at proof of principle as to the utility of the technique. Please also ensure that everything is publicly available as requested by reviewer #2, as well as addressing this reviewer's minor experimental concerns.

GENERAL GUIDELINES:

Text limits: Character count for an Tools is < 40,000, not including spaces. Count includes title page, abstract, introduction, results, discussion, acknowledgments, and figure legends. Count does not include materials and methods, references, tables, or supplemental legends.

Figures: Toolss may have up to 10 main text figures. Figures must be prepared according to the policies outlined in our Instructions to Authors, under Data Presentation, <https://jcb.rupress.org/site/misc/ifora.xhtml>. All figures in accepted manuscripts will be screened prior to publication.

*****IMPORTANT:** It is JCB policy that if requested, original data images must be made available. Failure to provide original images upon request will result in unavoidable delays in publication. Please ensure that you have access to all original microscopy and blot data images before submitting your revision. ***

Supplemental information: There are strict limits on the allowable amount of supplemental data. Toolss may have up to 5 supplemental figures. Up to 10 supplemental videos or flash animations are allowed. A summary of all supplemental material should appear at the end of the Materials and methods section.

Please note that JCB now requires authors to submit Source Data used to generate figures containing gels and Western blots with all revised manuscripts. This Source Data consists of fully uncropped and unprocessed images for each gel/blot displayed in the main and supplemental figures. Since your paper includes cropped gel and/or blot images, please be sure to provide one Source Data file for each figure that contains gels and/or blots along with your revised manuscript files. File names for Source Data figures should be alphanumeric without any spaces or special characters (i.e., SourceDataF#, where F# refers to the associated main figure number or SourceDataFS# for those associated with Supplementary figures). The lanes of the gels/blots should be labeled as they are in the associated figure, the place where cropping was applied should be marked (with a box), and molecular weight/size standards should be labeled wherever possible.

As you may know, the typical timeframe for revisions is three to four months. However, we at JCB realize that the

implementation of social distancing and shelter in place measures that limit spread of COVID-19 also pose challenges to scientific researchers. Lab closures especially are preventing scientists from conducting experiments to further their research. Therefore, JCB has waived the revision time limit. We recommend that you reach out to the editors once your lab has reopened to decide on an appropriate time frame for resubmission. Please note that papers are generally considered through only one revision cycle, so any revised manuscript will likely be either accepted or rejected.

Thank you for this interesting contribution to Journal of Cell Biology. You can contact us at the journal office with any questions, cellbio@rockefeller.edu or call (212) 327-8588.

Sincerely,

Joerg Bewersdorf, PhD
Monitoring Editor

Andrea L. Marat, PhD
Senior Scientific Editor

Journal of Cell Biology

Reviewer #1 (Comments to the Authors (Required)):

As this is a dual submission of closely related manuscripts, I will review both together in this report. In the first manuscript, the authors developed an ultrafast digital camera module, which is then incorporated into a single-molecule and super-resolution microscopy platform that are capable of extremely high-speed single-molecule tracking, and high-frame rate PALM super-resolution microscopy imaging. The design, rationale, theoretical foundation, experimental parametrization, and characterization of the camera are described in exhaustive details. The camera system is based on an image intensifier that is fiber-coupled to a high-speed CMOS camera. Cy3 fluorophores was found to be most suitable fluorophore for high-speed single-molecule imaging but the camera speed is such that the photon emission cycle of Cy3 is slower than the capability of the camera, thus the fluorophore photophysics is claimed to be the rate limiting step, with this camera technology.

The authors then investigate diffusion characteristics of Cy3-DOPE and TfR membrane protein in plasma membrane of living cells with up to 10 kHz speed. These measurements yield finely sampled trajectories with large number of small intervals (100s-1000s), that reveal the characteristics of hop diffusion. By perturbing actin cytoskeleton and observing the blebbing membrane, the authors showed that Hop diffusion appears to be dependent on intact actin-based membrane cytoskeleton. In conjunction, the authors also present detailed theoretical frameworks for trajectory analysis to detect hop diffusion. These results appear to be in good agreement with previous study by the authors from ~2 decades ago (also in JCB), which use gold nanoparticle and brightfield imaging. Thus, the current study corroborate the model of plasma membrane/membrane cytoskeleton organization/compartmentalization, while describing a more generalizable experimental platform and a less perturbative tagging (fluorophores vs 40-nm nanoparticles).

Next, the authors describe how the high-speed camera can also be used for Live-cell PALM, which was then applied to image caveolae and focal adhesions live cells.

In the 2nd manuscript, the author used the imaging system thus developed for a more cell biological applications, with the goals of probing membrane organization in the ventral plasma membrane as well as in focal adhesions. Using Transferring Receptor and Cy3-DOPE, the authors showed that these labelled lipid and membrane proteins exhibit similar diffusion characteristics between dorsal and ventral plasma membrane. Similar kinetics is also observed for EGFR, suggesting that these kinetics are dependent on plasma membrane compartmentalization. Subsequently, live-cell PALM is applied to image focal adhesions revealing heterogeneous nanocluster organization of paxillin (so-called 'islands'), while TfR high-speed SPT is performed to probe membrane organization in focal adhesions. Analysis of TfR trajectory indicates that plasma membrane in focal adhesion is also compartmentalized but with smaller compartment and longer dwell time. High-speed SPT of integrin beta 3 was performed which reveal complex trajectory that can periodically be immobilized at paxillin-based 'islands'.

In summary, this body of work is a technical tour de force study by established investigators in the field of single-molecule imaging/membrane biophysics that advance the state-of-the-art in fluorescence imaging capability by orders of magnitude. The experimental execution and supporting theories are sound and rigorous, while the technological advances presented should be generally useful and readily adaptable to other bioimaging modalities. The high-speed capability is likely to be game-changer in addressing a number of key biological/biophysical questions. Thus, I am in support of these works being eventually accepted for publication in JCB. That said, with the current structure of these two manuscripts, though the text itself is well-written, it is still quite challenging to digest such a large amount of information and a revision is strongly advised.

1. Of the two manuscripts, in the current organization, the first manuscript is clearly the strongest as it describes all of the novel results and major technical advances. In contrast, the key results on integrin diffusion in the 2nd manuscript is perhaps somewhat overshadowed by the authors's own previous work in 2018 Nature Chemical Biology Tsunoyama et al., that also describe similar trajectory characteristics of integrin.

As it seems that all the ground-breaking exciting results are all contained in 1st manuscript this has the side effect of depriving the 2nd manuscript of key results. At the same time, this also makes the 1st manuscript quite dense and challenging to digest.

My suggestion is to revise the 1st manuscript to focus on the camera, SPT, hop diffusion, and membrane organization. Then, the live-cell PALM sections can be moved from the 1st to the 2nd manuscript. This way, the first manuscript will be more nicely packaged as an ultrafast SPT & more biophysical membrane organization study, while the 2nd manuscript will be on fast live PALM super-resolution, their characterization, and more cell biological study. The content in both manuscripts may be better balanced and more digestible this way.

2. Some of the figure panels would benefit from more clear captions or sub-titles, so that readers do not need to go to the figure description for every panels.

3. While MINFLUX is another technique capable of high-speed fast single-molecule tracking and thus the authors may feel the need to differentiate their approach from MINFLUX, this reviewer is of the view that MINFLUX is much more of a niche technique compared to the more generalizable and modular capability presented by the fast camera in this study. Perhaps the MINFLUX discussion can be included as supplementary note instead so as not to detract from the main text.

4. One of the main weaknesses of 2nd manuscript starts with the abstract which is mostly descriptive. It is not clear from reading the abstract what is the key 'take-home' biological findings. As the authors appear to intend both manuscripts as a technological demonstration rather than a full-fledged mechanistic dissection, I would suggest to revise according to #1 above and also rewrite the abstract accordingly. Alternatively, additional biological perturbations may be needed for 2nd manuscript to dissect what factors are regulating membrane partitioning in focal adhesions. However, given that there is a vast amount of data in these two manuscripts already, this reviewer is of the view that a revision as in #1 is probably more advisable.

Reviewer #2 (Comments to the Authors (Required)):

This clearly presented, carefully prepared, and convincingly documented manuscript describes the development of an impressive ultrafast camera system for single-molecule imaging of fluorescently tagged molecules at or near the plasma membrane. The precision of 29-34 nm at 33 microseconds for multiple molecules represents a valuable advance for the study of dynamics of individual as well as groups of up to hundreds of labeled molecules without the need for the use of large gold probes. It can be used with TIRF, PALM (75 nm), and potentially STORM to characterize molecule translocation or immobilization spanning an entire cell. Interestingly, it uses a CMOS camera with a global shutter rather than an sCMOS camera. This valuable technology should substantially advance single-molecule tracking to explore a range of questions involving single- and collective-molecule dynamics. It now places the current technical challenge for innovation back to the developers of fluorophores. The only substantive reservation involves the availability of this technology.

1. The only potentially major concern involves the availability of this new technological advance to cell biologists. Although JCB is logically the best place to publish this advance, the instructions for JCB Tools states: "Tools describe new methods ... of immediate value and broad utility to the cell biology community." The key question here is whether this clearly valuable, unique Tool will be of "broad utility" to other cell biology laboratories in terms of practical availability. The specific concern is whether this camera will be: 1) commercially available, or 2) practical to construct by other laboratories based solely on the current information provided, or 3) available for use within the Kusumi laboratory by outside researchers. That is, is this camera only a one-laboratory instrument without availability to other labs in cell biology? For example, concerning point b), the specific components would need better description, including the straight optical-fiber bundle directly adhered to the phosphor screen, as well as how the components are actually assembled together. If any of these 3 types of availability can be provided, this concern would, in my opinion, be resolved.

2. In the spirit of Research Reproducibility, because this is a Tools manuscript, availability of code in GitHub (e.g., for single-molecule position determination) and deposit of an example or two of primary data in some other database (e.g., in Figshare) is strongly encouraged, rather than availability just "from the corresponding author upon reasonable request."

Minor points:

3. Has more than one color of fluorophore been imaged yet by using more than one camera? Doing so could provide valuable information about inter-molecular interactions, e.g., in focal adhesions.

4. Might suppressors of ROS help reduce laser-induced cytotoxicity?
5. Please ensure that abbreviations are defined, e.g., TIR and MSK in Figure 3.

March 25, 2023

Joerg Bewersdorf, Ph.D.
Monitoring Editor

Andrea L. Marat, Ph.D.
Senior Scientific Editor

Re: JCB manuscript #202110160, slightly retitled as,
"Development of ultrafast camera-based single fluorescent-molecule imaging for cell biology"

Dear Joerg and Andrea,

Thank you very much for critically reading and assessing our manuscript for publication as a Tools paper in JCB. We would also like to thank you for obtaining the opinions of the two referees. Attached please find our revised manuscript, addressing the "minor revision" according to your decision. As we requested in our first submission, we hope that this paper can be published back-to-back with its companion normal Article manuscript (#202110162, retitled as "Ultrafast single-molecule imaging reveals focal adhesion nano-architecture and molecular dynamics"). We are submitting these two companion manuscripts at the same time.

We have addressed all of the points raised by you and your referees in the revised manuscript. We have basically complied with all of the recommendations.

In particular, you indicated two criteria required for publication as a Tools paper: (1) immediate value and broad utility of the described method to the cell biology community, and (2) inclusion of novel biological insight as proof of principle for the utility of the technique.

Regarding **your first point**, it is addressed in the **last two paragraphs in the Discussion (p. 24)**. There, we emphasize the following four points. (1) The highest time resolutions in single fluorescent-molecule imaging to date are now available to the cell biology community. (2) The ultrafast camera can be custom ordered from Photron after the publication of this manuscript. (3) The ultrafast single-molecule imaging method is now enhanced by the development of the theoretical framework for the analysis of single-molecule trajectories in the plasma membrane (PM), which provides the simple means to examine the effects of PM compartmentalization on biological processes in/on the PM (previously, very special single gold-particle imaging and tracking were necessary) and to elucidate the principles governing the PM organization. (4) The ultrafast camera will facilitate the monitoring of very fast molecular interactions, which have probably been missed in cell biology research.

Furthermore, as shown in the companion paper, the newly developed ultrafast camera system also offers an unprecedented opportunity to simultaneously perform the fastest data acquisition (1 kHz; $\approx 30\times$ faster than the standard rate employed by many scientists now) of PALM (using the mEos3.2 probe with a 29-nm localization precision) and dSTORM (using the HMSiR probe with a 19-nm localization precision) in a view-field as large as 640 x 640 pixels (35.3 x 35.3 μm^2 with the pixel size of 55.1 nm).

The ultrafast cameras are also available in our laboratories (Kyoto University, Gifu University, and OIST) upon reasonable request, as described in "**Instrument (developed camera) availability**". In addition, the codes and data are publicly available, as described in "**Code availability**" and "**Data availability**", respectively. Therefore, virtually everything is publicly available, as requested by you and Reviewer 2.

We believe that **your second point** has been addressed well in this revision by following Reviewer 1's recommendation. Namely, although we have moved the sections describing the application of the ultrafast camera to the ultrafast live-cell PALM imaging, from this Tools manuscript to the companion Article manuscript, we have also moved the sections describing the compartmentalization of the basal PM, its characteristics, and the effect of compartmentalization on EGFR diffusion before and after the ligand EGF addition, from the companion Article manuscript to this Tools manuscript.

As a result, we believe that this Tools manuscript has been considerably improved and has become "more digestible", as indicated by Reviewer 1.

Meanwhile, I would like to make a strong request that the two manuscripts we are submitting together be considered in combination. Since the descriptions of the design concept of the ultrafast camera, the actual design of the camera based on various measurements, the final specs and the performance of the camera as well as the camera's initial applications that led to important cell biological findings became so voluminous, we were unable to include them in a single paper. Before writing up the results of this work, we considered various publishing schemes, including the possibility of publishing these results in other journals, but we concluded that the best and most impactful way of publishing this work would be to ask you to publish the two closely-linked manuscripts back-to-back in your journal. By publishing these two manuscripts together, we can cover the basic development part and important applications part in ways better understood by the readers.

We would like to thank you and your reviewers again for critically reading our manuscript and providing constructive comments and recommendations. We hope this manuscript is now acceptable for publication in *The Journal of Cell Biology*.

(Meanwhile, in the revised Article manuscript [companion paper], we added entirely new results about dSTORM and simultaneous two-color PALM-dSTORM, as well as their applications to the studies of nano-scale architectures and focal adhesion dynamics. Following the recommendations by Reviewer 2, we greatly extended our analyses of the focal adhesion in the companion Article manuscript.)

The revisions made in the main text and figure captions are indicated by **yellow highlighting** (but small wording changes are not highlighted). The texts and the captions to figures and videos moved from the Article manuscript (with some necessary modifications) are shown by **cyan highlighting**. When the whole sections were moved or rewritten, to increase the readability, only the **section/figure titles** are highlighted in cyan or yellow, respectively. The figures moved from the original Article manuscript (**Figs. 6 and 7**) are surrounded by **blue rectangles**. Our point-by-point responses to your reviewers' comments are provided on the following pages.

Sincerely yours,

Aki (Akihiro Kusumi)

Professor

Membrane Cooperativity Unit

Okinawa Institute of Science and Technology Graduate University (OIST)

e-mail: akihiro.kusumi@oist.jp

Reviewer #1 (Comments to the Authors (Required)):

As this is a dual submission of closely related manuscripts, I will review both together in this report. In the first manuscript, the authors developed an ultrafast digital camera module, which is then incorporated into a single-molecule and super-resolution microscopy platform that are capable of extremely high-speed single-molecule tracking, and high-frame rate PALM super-resolution microscopy imaging. The design, rationale, theoretical foundation, experimental parametrization, and characterization of the camera are described in exhaustive details. The camera system is based on an image intensifier that is fiber-coupled to a high-speed CMOS camera. Cy3 fluorophores was found to be most suitable fluorophore for high-speed single-molecule imaging but the camera speed is such that the photon emission cycle of Cy3 is slower than the capability of the camera, thus the fluorophore photophysics is claimed to be the rate limiting step, with this camera technology.

The authors then investigate diffusion characteristics of Cy3-DOPE and TfR membrane protein in plasma membrane of living cells with up to 10 kHz speed. These measurements yield finely sampled trajectories with large number of small intervals (100s-1000s), that reveal the characteristics of hop diffusion. By perturbing actin cytoskeleton and observing the blebbing membrane, the authors showed that Hop diffusion appears to be dependent on intact actin-based membrane cytoskeleton. In conjunction, the authors also present detailed theoretical frameworks for trajectory analysis to detect hop diffusion. These results appear to be in good agreement with previous study by the authors from ~2 decades ago (also in JCB), which use gold nanoparticle and brightfield imaging. Thus, the current study corroborate the model of plasma membrane/membrane cytoskeleton organization/compartmentalization, while describing a more generalizable experimental platform and a less perturbative tagging (fluorophores vs 40-nm nanoparticles).

Next, the authors describe how the high-speed camera can also be used for Live-cell PALM, which was then applied to image caveolae and focal adhesions live cells.

In the 2nd manuscript, the author used the imaging system thus developed for a more cell biological applications, with the goals of probing membrane organization in the ventral plasma membrane as well as in focal adhesions. Using Transferring Receptor and Cy3-DOPE, the authors showed that these labelled lipid and membrane proteins exhibit similar diffusion characteristics between dorsal and ventral plasma membrane. Similar kinetics is also observed for EGFR, suggesting that these kinetics are dependent on plasma membrane compartmentalization. Subsequently, live-cell PALM is applied to image focal adhesions revealing heterogeneous nanocluster organization of paxillin (so-called 'islands'), while TfR high-speed SPT is performed to probe membrane organization in focal adhesions. Analysis of TfR trajectory indicates that plasma membrane in focal adhesion is also compartmentalized but with smaller compartment and longer dwell time. High-speed SPT of integrin beta 3 was performed which reveal complex trajectory that can periodically be immobilized at paxillin-based 'islands'.

In summary, this body of work is a technical tour de force study by established investigators in the field of single-molecule imaging/membrane biophysics that advance the state-of-the-art in fluorescence imaging capability by orders of magnitude. The experimental execution and supporting theories are sound and rigorous, while the technological advances presented should be generally useful and readily adaptable to other bioimaging modalities. The high-speed capability is likely to be game-changer in addressing a number of key biological/biophysical questions. Thus, I am in support of these works being eventually accepted for publication in JCB. That said, with the current structure of these two manuscripts, though the text itself is well-written, it is still quite challenging to digest such a large amount of information and a revision is strongly advised.

Thank you very much for carefully reading our manuscripts and for your kind, important, and constructive comments. In particular, your advice for reorganizing the two manuscripts was excellent and important. We followed your advice. Thank you very much again.

The revisions made in the text are indicated by **yellow highlighting** (but small wording changes are not highlighted). The texts moved from the Article manuscript (with some necessary modifications) are shown by **cyan highlighting**. When the whole sections were moved or rewritten, to increase the readability, only the **section/figure titles** are highlighted in cyan or yellow, respectively. The figures moved from the original Article manuscript (**Figs. 6 and 7**) are surrounded by **blue rectangles**.

1. Of the two manuscripts, in the current organization, the first manuscript is clearly the strongest as it describes all of the novel results and major technical advances. In contrast, the key results on integrin diffusion in the 2nd manuscript is perhaps somewhat overshadowed by the authors's own previous work in 2018 Nature Chemical Biology Tsunoyama et al., that also describe similar trajectory characteristics of integrin.

As it seems that all the ground-breaking exciting results are all contained in 1st manuscript this has the side effect of depriving the 2nd manuscript of key results. At the same time, this also makes the 1st manuscript quite dense and challenging to digest.

My suggestion is to revise the 1st manuscript to focus on the camera, SPT, hop diffusion, and membrane organization. Then, the live-cell PALM sections can be moved from the 1st to the 2nd manuscript. This way, the first manuscript will be more nicely packaged as an ultrafast SPT & more biophysical membrane organization study, while the 2nd manuscript will be on fast live PALM super-resolution, their characterization, and more cell biological study. The content in both manuscripts may be better balanced and more digestible this way.

As recommended by Reviewer 1, in the Tools manuscript (first manuscript), we now focus on the camera, SPT, hop diffusion, and membrane organization. We have moved the live-cell PALM sections from the Tools manuscript to the Article manuscript (second manuscript). Meanwhile, we have moved the sections describing the compartmentalization of the basal plasma membrane (PM), and the hop diffusion of Cy3-DOPE, transferrin receptor, and EGF receptor before and after the EGF application, from the Article manuscript to the Tools manuscript. In the Article manuscript, in addition to ultrafast live-cell PALM, we now describe the application of the ultrafast camera to ultrafast dSTORM and simultaneous two-color ultrafast PALM-dSTORM. Furthermore, we added interesting results about the molecular architectures and dynamics of focal adhesions (FAs), thus strengthening the Article manuscript. We believe that, by following Reviewer 1's advice, both manuscripts are stronger, clearer, better balanced, and more comprehensible.

2. Some of the figure panels would benefit from more clear captions or sub-titles, so that readers do not need to go to the figure description for every panels.

Following this recommendation, we revised the figure caption titles of **Figs. 1, 2, 4, 5, and S2**, as well as the titles and descriptions of **Figs. 2 D and E, 4 B-D, 5 A and B, and S3 A and C**.

3. While MINFLUX is another technique capable of high-speed fast single-molecule tracking and thus the authors may feel the need to differentiate their approach from MINFLUX, this reviewer is of the view that MINFLUX is much more of a niche technique compared to the more generalizable and modular capability presented by the fast camera in this study. Perhaps the MINFLUX discussion can be included as supplementary note instead so as not to detract from the main text.

According to this advice from Reviewer 1, we now refer to MINFLUX only in the **Discussion** section, and removed its description from the **Introduction**.

4. One of the main weaknesses of 2nd manuscript starts with the abstract which is mostly descriptive. It is not clear from reading the abstract what is the key 'take-home' biological findings. As the authors appear to intend both manuscripts as a technological demonstration rather than a full-fledged mechanistic dissection, I would suggest to revise according to #1 above and also rewrite the abstract accordingly. Alternatively, additional biological perturbations may be needed for 2nd manuscript to dissect what factors are regulating membrane partitioning in focal adhesions. However, given that there is a vast amount of data in these two manuscripts already, this reviewer is of the view that a revision as in #1 is probably more advisable.

As described in our response to item 1, we revised the Tools manuscript (first manuscript) as advised by Reviewer 1. Meanwhile, Reviewer 2 strongly recommended that we include more biological data about the focal adhesion in the Article manuscript (second manuscript). I discussed this with the editors, and they made it clear that we should add more biological data in the Article manuscript. Therefore, in addition to the revisions we made following your recommendations, for the Article manuscript, we have added some more very interesting and exciting cell biological data. They are described in our responses to the points raised by Reviewer 2 in the cover letter for the Article manuscript (second manuscript). Could you take a look at our responses there?

Reviewer #2 (Comments to the Authors (Required)):

This clearly presented, carefully prepared, and convincingly documented manuscript describes the development of an impressive ultrafast camera system for single-molecule imaging of fluorescently tagged molecules at or near the plasma membrane. The precision of 29-34 nm at 33 microseconds for multiple molecules represents a valuable advance for the study of dynamics of individual as well as groups of up to hundreds of labeled molecules without the need for the use of large gold probes. It can be used with TIRF, PALM (75 nm), and potentially STORM to characterize molecule translocation or immobilization spanning an entire cell. Interestingly, it uses a CMOS camera with a global shutter rather than an sCMOS camera. This valuable technology should substantially advance single-molecule tracking to explore a range of questions involving single- and collective-molecule dynamics. It now places the current technical challenge for innovation back to the developers of fluorophores. The only substantive reservation involves the availability of this technology.

Thank you very much for critically reading our manuscripts and for your kind and constructive comments.

The revisions made in the text are indicated by **yellow highlighting** (but small wording changes are not highlighted). The texts moved from the Article manuscript (with some necessary modifications) are shown by **cyan highlighting**. When the whole sections were moved or rewritten, to increase the readability, only the **section/figure titles** are highlighted in cyan or yellow, respectively. The figures moved from the original Article manuscript (**Figs. 6 and 7**) are surrounded by **blue rectangles**.

1. The only potentially major concern involves the availability of this new technological advance to cell biologists. Although JCB is logically the best place to publish this advance, the instructions for JCB Tools states: "Tools describe new methods ... of immediate value and broad utility to the cell biology community." The key question here is whether this clearly valuable, unique Tool will be of "broad utility" to other cell biology laboratories in terms of practical availability. The specific concern is whether this camera will be: 1)

commercially available, or 2) practical to construct by other laboratories based solely on the current information provided, or 3) available for use within the Kusumi laboratory by outside researchers. That is, is this camera only a one-laboratory instrument without availability to other labs in cell biology? For example, concerning point b), the specific components would need better description, including the straight optical-fiber bundle directly adhered to the phosphor screen, as well as how the components are actually assembled together. If any of these 3 types of availability can be provided, this concern would, in my opinion, be resolved.

Thank you very much for pointing out these issues. All three types of availability are now satisfied, as described on **p. 41**.

(1) This camera is commercially available by custom order from a company called Photron. Thus far, several people who attended my seminars and/or read our bioRxiv manuscripts have contacted me, and I have been referring them to Photron. Two cameras have already been sold to an Indian research institute and one has been ordered by a university in Singapore. These cameras have been purchased with the understanding that their initial projects will be conducted in collaboration with Fujiwara, Kusumi, and Photron, until this paper is published. After the publication of this manuscript, the camera can simply be custom ordered from Photron. A person in the Photron company management stated that they are interested in working with researchers in the cell biology field, and that at this point, it is OK with them if they do not lose too much money by the custom production of these cameras (they also said that since each camera is built and adjusted by their R&D engineers, they have difficulty estimating the personnel cost, as these excellent R&D engineers can be engaged in other projects that can give much quicker monetary returns).

(2) Similar cameras can be built by other laboratories and companies, based on what we describe in the Tools manuscript. The design idea of the ultrafast camera is new, but the technologies used to build the camera system are not new.

(3) These cameras are available to outside researchers in the Kusumi lab at OIST, the Fujiwara lab at Kyoto University, and the Suzuki lab at Gifu University (the labs of three co-authors of the manuscript). In fact, we are quite willing to share the ultrafast cameras.

2. In the spirit of Research Reproducibility, because this is a Tools manuscript, availability of code in GitHub (e.g., for single-molecule position determination) and deposit of an example or two of primary data in some other database (e.g., in Figshare) is strongly encouraged, rather than availability just "from the corresponding author upon reasonable request."

The codes are available at GitHub, as described in "Code Availability". Our method for obtaining the coordinates of the observed single molecules is basically the same as that described by Crocker and Grier (1996). The MATLAB codes based on this method, written by Blair and Dufresne (2009), are available at <https://site.physics.georgetown.edu/matlab/code.html>. Our codes for this purpose have been integrated into a much larger, complex software package and they cannot be extracted in a useful way. However, the entire software package is available from the corresponding author upon reasonable request (we will provide personal guidance on how to use it, as about half of its manual and comments are written in Japanese). This is explained on **p. 41**.

Minor points:

3. Has more than one color of fluorophore been imaged yet by using more than one camera? Doing so could provide valuable information about inter-molecular interactions, e.g., in focal adhesions.

The results of simultaneous two-color experiments were already shown in the original Article manuscript, and we have added more relevant data in its revision. Since the ultrafast camera can be used on single fluorescent-molecule imaging stations, just like any other cameras, it can naturally perform simultaneous two-color experiments.

Specifically, in the revised Article manuscript, the results of two-color observations are described in the following figures. **Fig. 5 C - G** and the entire **Fig. 6** show the results of simultaneous two-color data acquisitions of dSTORM (HMSiR-labeled Halo tags excited at 660 nm) and PALM (mEos3.2 at 561 nm), both at 1 kHz; **Fig. 9 A - D** and **E** show the results of simultaneous two-color data acquisitions of PALM of mEos3.2-paxillin (1 kHz at 561 nm; both **A - D** and **E**) and ultrafast SFMI integrin β 3 molecules (SeTau647-labeled ACP tags; 250 Hz at 660 nm for **A - D** and SeTau647-labeled Halo tags; 6 kHz at 660 nm for **E**).

4. Might suppressors of ROS help reduce laser-induced cytotoxicity?

They might, but we decided not to use them to avoid further complexities and side effects and because we think that the results are quite satisfactory without ROS suppressors (**Fig. 2 D** and **E**).

5. Please ensure that abbreviations are defined, e.g., TIR and MSK in Figure 3.

Done.

April 27, 2023

RE: JCB Manuscript #202110160R

Prof. Akihiro Kusumi
Okinawa Institute of Science and Technology Graduate University
Membrane Cooperativity Unit
Onna-son
Okinawa 904-0495
Japan

Dear Prof. Kusumi:

Thank you for submitting your revised manuscript entitled "Development of ultrafast camera-based single fluorescent-molecule imaging for cell biology". The reviewers all now support publication so we would be happy to publish your paper in JCB pending final revisions necessary to meet our formatting guidelines (see details below).

A. MANUSCRIPT ORGANIZATION AND FORMATTING:

- 1) Text limits: Character count for Tools is < 40,000, not including spaces. Count includes abstract, introduction, results, discussion, and acknowledgments. Count does not include title page, figure legends, materials and methods, references, tables, or supplemental legends.
- 2) Figures limits: Tools may have up to 10 main text figures.
- 3) Figure formatting: Scale bars must be present on all microscopy images, including inset magnifications. Molecular weight or nucleic acid size markers must be included on all gel electrophoresis.
- 4) Statistical analysis: Error bars on graphic representations of numerical data must be clearly described in the figure legend. The number of independent data points (n) represented in a graph must be indicated in the legend. Statistical methods should be explained in full in the materials and methods. For figures presenting pooled data the statistical measure should be defined in the figure legends. Please also be sure to indicate the statistical tests used in each of your experiments (either in the figure legend itself or in a separate methods section) as well as the parameters of the test (for example, if you ran a t-test, please indicate if it was one- or two-sided, etc.). Also, if you used parametric tests, please indicate if the data distribution was tested for normality (and if so, how). If not, you must state something to the effect that "Data distribution was assumed to be normal but this was not formally tested."
- 5) Abstract and title: The abstract should be no longer than 160 words and should communicate the significance of the paper for a general audience. The title should be less than 100 characters including spaces. Make the title concise but accessible to a general readership.
- 6) Materials and methods: Should be comprehensive and not simply reference a previous publication for details on how an experiment was performed. Please provide full descriptions in the text for readers who may not have access to referenced manuscripts.
- 7) Please be sure to provide the sequences for all of your primers/oligos and RNAi constructs in the materials and methods. You must also indicate in the methods the source, species, and catalog numbers (where appropriate) for all of your antibodies. Please also indicate the acquisition and quantification methods for immunoblotting/western blots.
- 8) Microscope image acquisition: The following information must be provided about the acquisition and processing of images:
 - a. Make and model of microscope
 - b. Type, magnification, and numerical aperture of the objective lenses
 - c. Temperature
 - d. Imaging medium
 - e. Fluorochromes
 - f. Camera make and model

g. Acquisition software

h. Any software used for image processing subsequent to data acquisition. Please include details and types of operations involved (e.g., type of deconvolution, 3D reconstitutions, surface or volume rendering, gamma adjustments, etc.).

9) References: There is no limit to the number of references cited in a manuscript. References should be cited parenthetically in the text by author and year of publication. Abbreviate the names of journals according to PubMed. Supplemental references are not permitted.

10) Supplemental materials: There are strict limits on the allowable amount of supplemental data. Tools may have up to 5 supplemental figures. Please also note that tables, like figures, should be provided as individual, editable files. A summary of all supplemental material should appear at the end of the Materials and methods section.

13) ORCID IDs: ORCID IDs are unique identifiers allowing researchers to create a record of their various scholarly contributions in a single place. At resubmission of your final files, please consider providing an ORCID ID for as many contributing authors as possible.

Please note that JCB now requires authors to submit Source Data used to generate figures containing gels and Western blots with all revised manuscripts. This Source Data consists of fully uncropped and unprocessed images for each gel/blot displayed in the main and supplemental figures. Since your paper includes cropped gel and/or blot images, please be sure to provide one Source Data file for each figure that contains gels and/or blots along with your revised manuscript files. File names for Source Data figures should be alphanumeric without any spaces or special characters (i.e., SourceDataF#, where F# refers to the associated main figure number or SourceDataFS# for those associated with Supplementary figures). The lanes of the gels/blots should be labeled as they are in the associated figure, the place where cropping was applied should be marked (with a box), and molecular weight/size standards should be labeled wherever possible.

Journal of Cell Biology now requires a data availability statement for all research article submissions. These statements will be published in the article directly above the Acknowledgments. The statement should address all data underlying the research presented in the manuscript. Please visit the JCB instructions for authors for guidelines and examples of statements at (<https://rupress.org/jcb/pages/editorial-policies#data-availability-statement>).

B. FINAL FILES:

**It is JCB policy that if requested, original data images must be made available to the editors. Failure to provide original images upon request will result in unavoidable delays in publication. Please ensure that you have access to all original data images prior

to final submission.**

Thank you for this interesting contribution, we look forward to publishing your paper in Journal of Cell Biology.

Sincerely,

Joerg Bewersdorf, PhD
Monitoring Editor

Andrea L. Marat, PhD
Senior Scientific Editor

Journal of Cell Biology

Reviewer #1 (Comments to the Authors (Required)):

In this revision, the author has revised and reorganized both their manuscripts substantially. This manuscript focus on the technical aspect of the camera design with detailed characterization of their performance in single-molecule imaging at the kHz regime of speed. The author systematically characterized performances of multiple fluorophores and identified Cy3 as the optimal fluorophore most suitable for high-speed single-molecule tracking and identified experimental condition amenable for live-cell analysis.

Using their camera system to probe diffusion of Cy3 conjugated lipid (DOPE) or membrane protein (Transferrin receptor) they clearly demonstrated that with kHz frame-rate the hop-diffusion behaviour of single-molecule can be visualized, which would otherwise be missed by the video-rate tracking. Additionally, the authors also developed detailed theoretical framework for analyzing hop diffusion data.

Finally, using their camera system they investigated apical and basal plasma membrane using both lipid DOPE or membrane protein (TfR and EGFR). Their analysis suggest that these membranes are likely compartmentalized to a similar extent, while compartment dwell time could vary (increased when EGF is added, consistent with liganded-mediated EGFR dimerization). Overall, this is a richly detailed, technically rigorous manuscript that describe an important technological advance in imaging capability. While the specific application demonstrated here give us new insight into plasma membrane organization, the new capability developed here as well as the associated theoretical framework will undoubtedly be useful in other areas of cell biology in future studies. For example, while I find it quite sensible (or even reassuring) that the apical and basal plasma membrane of an isolated cell are similarly organized, it will be interesting to probe the membrane organization in apicobasally polarized cell such as in mature epithelial models in future studies. In general, I am satisfied with this revision, and support its publication in Journal of Cell Biology.

Reviewer #2 (Comments to the Authors (Required)):

This resubmitted manuscript has been significantly improved by the revisions, and it now appears focused and well-organized. The comparison with MINFLUX is useful to retain from the point of view of informing the general cell biologist of the importance of this complementary, potentially broadly applicable technology. The original concerns about availability of equipment and code

have been resolved after revision. Overall, this technology is an impressive advance to the current state-of-the-art, and the authors should be congratulated. Acceptance for publication is now clearly appropriate.